# CLARITree: Cholesky and Lookahead Accelerations for Regression with Interpretable Piecewise Linear Trees

**Yixiao Wang** [* 1]  **Hayden McTavish** [* 1]  **Varun Babbar** [* 1]  **Margo Seltzer** [2]  **Cynthia Rudin** [1]

## Abstract

Regression trees are among the most interpretable yet expressive model classes in machine learning. Historically, greedy induction has been the dominant approach for constructing well-performing regression trees. While optimal methods based on dynamic programming and branch-and-bound exist, they are computationally prohibitive for general linear regression trees, despite often achieving substantially better performance than greedy approaches. Recent work has shown that specialized lookahead strategies can dramatically improve runtime while maintaining near-optimal performance, primarily in classification settings. In this work, we develop a novel algorithm for near-optimal, sparse, piecewise linear regression trees that combines a lookahead-style search strategy with efficient rank-one Cholesky updates of the Gram matrix. We demonstrate, both theoretically and empirically, that our method achieves a favorable trade-off between computational efficiency, predictive accuracy, and sparsity, and scales significantly better than the current state of the art. The code is available at https://github.com/Yixiao-Wang-Stats/CLARITree.

## 1. Introduction

Decision trees for regression date back to the early work of Morgan & Sonquist (1963) and were later popularized with CART and C4.5 (Breiman et al., 1984; Quinlan, 1993). They remain cornerstone models for interpretable learning (Rudin, 2019; Rudin et al., 2022; Blockeel et al., 2023), widely adopted for their simplicity, transparency, and ability to capture nonlinear relationships.

[*]Equal contribution  [1] Department of Computer Science, Duke University, Durham, USA  [2] Department of Computer Science, University of British Columbia, Vancouver, Canada . Correspondence to: Yixiao Wang <yixiao.wang@duke.edu>.

*Proceedings of the 43rd International Conference on Machine Learning*, Seoul, South Korea. PMLR 306, 2026. Copyright 2026 by the author(s).

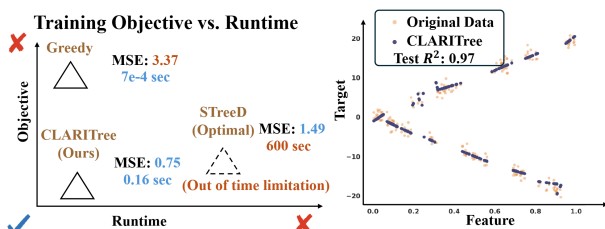

*Figure 1.* An illustration of CLARITree on a synthetic piecewise-linear dataset. Training objective versus runtime is shown. Greedy trees are fast but suboptimal, while optimal trees take too long to run. CLARITree provides a strong trade-off, achieving near-optimal performance with much lower runtime. The reported results include both train and test performance. **Test performance:** Greedy (MSE = 15.41, $R^2 = 0.88$); CLARITree (MSE = 4.03, $R^2 = 0.97$); STreeD (MSE = 13.72, $R^2 = 0.89$)

Greedy induction for regression trees has been extensively studied for both constant and linear models (Morgan & Sonquist, 1963; Quinlan, 1992; Wang & Witten, 1997; Breiman et al., 1984; Quinlan, 1993; Loh, 2002). While constant-leaf trees are computationally efficient, they lack linear expressiveness and therefore incur more model bias; conversely, linear-leaf trees improve modeling power but are less efficient to learn, requiring costly regressions at every node (Loh, 2014; Raymaekers et al., 2024). Greedy trees can deviate substantially from optimal solutions, with large gaps documented in both constant and linear settings (Zhang et al., 2023; Van Den Bos et al., 2024; van der Linden et al., 2024), underscoring the need for more principled approaches.

To address the issue of greedy induction's performance gap, recent work has used dynamic programming and branch-and-bound strategies to achieve provably optimal regression trees (Zhang et al., 2023; Van Den Bos et al., 2024), preceded by work on these techniques for classification trees (Hu et al., 2019; Lin et al., 2020; McTavish et al., 2022; Demirović et al., 2022; Aglin et al., 2020). Yet, the cost of solving linear regressions restricts these methods to constant predictors or highly constrained linear cases (e.g., single-feature), or leads to substantial scalability losses, leaving open the challenge of scalable algorithms that combine the efficiency of greedy induction with the accuracy of optimal search.

*Our work develops efficient near-optimal algorithms for general linear regression trees.* Empirically, the proposed method consistently outperforms greedy baselines and achieves performance close to optimal on small and medium-scale datasets, while remaining scalable and effective on large-scale problems. To address the cost of linear regressions and handle continuous features directly, we further use *rank-one Cholesky updates* to maintain regularized Gram factorizations while scanning thresholds, enabling numerically stable and efficient split evaluation for regression tree search.

Our work is motivated by a recent result of SPLIT (Babbar et al., 2025), which optimizes splits globally up to a lookahead depth and applies greedy induction thereafter. By combining this lookahead principle with efficient rank-one Cholesky updates for linear regression, we develop CLARITree (**C**holesky and **L**ookahead **A**ccelerations for **R**egression with **I**nterpretable **Tree**s), a scalable algorithm for learning interpretable piecewise linear regression trees.

Our contributions are as follows.

1. We introduce CLARITree, an efficient, near-optimal algorithm for learning sparse, piecewise linear regression trees that uses lookahead-style split optimization.

2. We make continuous-feature search computationally feasible and numerically stable by maintaining leaf regressors via rank-one Cholesky updates of the regularized Gram matrix, enabling fast, exact evaluation of many candidate splits without repeated refitting.

3. We provide theoretical analysis characterizing the runtime/accuracy trade-off of CLARITree and demonstrate, through extensive experiments, that our method achieves near-optimal accuracy while scaling substantially better than existing optimal baselines.

## 2. Related Work

**Greedy Regression Trees** Classical greedy regression trees can be divided into piecewise constant regression trees, which predict the sample mean in each leaf, and piecewise linear regression trees, which fit local regression models in the leaves. Traditional CART and C4.5 (Breiman et al., 1984; Quinlan, 1993) exemplify the former. Among linear regression trees, M5 (Quinlan, 1992) evaluates splits according to reduction in constant regression error but places linear regressors in the leaves. This hybrid strategy can induce a mismatch between split evaluation and final prediction, yielding suboptimal partitions (Malerba et al., 2004). GUIDE (Loh, 2002) instead applies statistical tests on residual patterns to select splits, both reducing variable-selection bias and improving the detection of informative variables. More recently, PILOT (Raymaekers et al., 2024) proposed

an efficient greedy algorithm for linear model trees: it restricts itself to simple linear fits, but adaptively selects among candidate models via a BIC criterion and maintains Gram matrices with rank-one updates during split evaluation. Unlike our work, PILOT focuses on very small regression models and recomputes and inverts the design matrices locally at each node, without maintaining shared state across the tree-building process. Our current work is scoped to standard greedy splitting criteria (i.e., reduction in MSE), allowing the framework to accommodate a range of greedy heuristics for tree computation (i.e., Balcan & Sharma (2024)).

**Optimal Decision and Regression Trees** There are many methods for finding optimal trees. These range across many techniques, including mixed-integer optimization (Bertsimas & Dunn, 2017) and SAT solvers (Verwer & Zhang, 2019). A recent literature review (Costa & Pedreira, 2023) suggested that the most promising approach for optimal trees has been tree-specific algorithms leveraging dynamic programming with branch and bound (Lin et al., 2020; Aglin et al., 2020; Demirović et al., 2022), which have been extended in more recent years (Sullivan et al., 2024; Chaouki et al., 2025; Zhang et al., 2023; van der Linden et al., 2024). Broadly speaking, these methods search through the space of decision trees while tracking lower and upper bounds at each split to prune the search space. To optimize regression trees, Zhang et al. (2023) leverage a novel k-means-based lower bound to prune search nodes. Van Den Bos et al. (2024) implement optimal algorithms for both constant and piecewise linear regression trees, employing lower bounds from (Zhang et al., 2023) to prune the search space, as well as specialized depth-2 solvers to speed up computation. Optimal tree methods have been extended to handle continuous features, either with heuristic preprocessing (McTavish et al., 2022) or specialized optimal methods (Mazumder et al., 2022; Briţa et al., 2025) but not while considering regression trees with linear functions in the leaves.

**Approximately optimal strategies** Because optimal tree construction methods can be slow, recent work has also focused on approximate strategies for decision tree construction, aiming to achieve the ideal balance between runtime and optimality. Top-$k$ (Blanc et al., 2023) provides a principled generalization of classical greedy decision tree algorithms by considering the top-$k$ candidate features at each split, yielding trees that perform better than greedy trees. DPDT (Kohler et al., 2025) further extends the splitting choices by introducing a split generating function that provides a richer set of candidate splits derived from deeper CART-based greedy trees. This approach identifies qualified splitting positions without enumerating all possible splits, thereby balancing the efficiency of greedy algorithms with the optimality of exact solvers. Several optimal meth-

ods have incorporated anytime behaviour (Demirović et al., 2023; Kiossou et al., 2025), allowing optimal methods to be terminated early if they exceed a compute budget while still affording a reasonable solution. Kiossou et al. (2024) make use of a greedy lookahead strategy, using a depth two optimal tree solver to optimize initial splits for a greedy heuristic. Babbar et al. (2025) introduced the SPLIT framework for classification trees, which selects split decisions using lookahead with greedy completions, potentially recursively. In contrast, our work is the first to make lookahead strategies practical for piecewise linear regression trees with continuous features, where exact, computationally expensive least-squares fitting and numerical stability pose fundamentally different algorithmic challenges.

## 3. Preliminaries and Notation

Let $\mathcal{F} = \mathcal{F}_c \cup \mathcal{F}_b$ denote the feature set with $|\mathcal{F}| = k$, where $\mathcal{F}_c$ and $\mathcal{F}_b$ are continuous and binary features, respectively. The training data is $D = \{(\boldsymbol{x}_i, y_i)\}_{i=1}^n$, where each feature vector $\boldsymbol{x}_i = (\boldsymbol{x}_i^c, \boldsymbol{x}_i^b)$ contains both continuous and binary components, and $y_i \in \mathbb{R}$ is the target. Let $[n] := \{1, \dots, n\}$ index the training examples. For any index set $\mathcal{I} \subseteq [n]$, let $D_\mathcal{I} := \{(\boldsymbol{x}_i, y_i)\}_{i \in \mathcal{I}}$ for the corresponding data subset, which we refer to as the node data when $\mathcal{I}$ represents the samples assigned to a tree node, with $\mathbf{X}_\mathcal{I} \in \mathbb{R}^{|\mathcal{I}| \times k}$ and $\boldsymbol{y}_\mathcal{I} \in \mathbb{R}^{|\mathcal{I}|}$ denoting its design matrix and target, respectively. For each feature $f \in \mathcal{F}$, let $\mathcal{P}_f$ denote the candidate threshold pool for feature $f$.

We define an *optimal linear sparse regression tree* as a tree $T_d \in \mathcal{T}$ of depth at most $d$, in the space of linear regression trees $\mathcal{T}$, that minimizes the sum of a prediction loss and a structural complexity penalty. Given a training dataset $D$ and depth budget $d$, each leaf $t$ is fitted with a ridge-regularized linear predictor with regularization parameter $\kappa > 0$, so that each sample receives the affine prediction $\widehat{y}_i = \beta_0 + \boldsymbol{x}_i^\top \boldsymbol{\beta}$, where the leaf coefficients are obtained by

$$\widehat{\beta}_0^{(t)}, \widehat{\boldsymbol{\beta}}^{(t)} = \underset{\beta_0, \boldsymbol{\beta}}{\operatorname{argmin}} \sum_{(\boldsymbol{x}_i, y_i) \in t} (y_i - \beta_0 - \boldsymbol{x}_i^\top \boldsymbol{\beta})^2 + \kappa \|\boldsymbol{\beta}\|_2^2. \tag{1}$$

The corresponding ridge-regularized prediction loss is denoted by $\mathcal{R}_\kappa(T_d, D)$. The overall objective is

$$\mathcal{L}^*(D, d, \lambda, \kappa) = \min_{T_d \in \mathcal{T}} \Big( \mathcal{R}_\kappa(T_d, D) + \lambda S(T_d) \Big), \tag{2}$$

where $S(T_d)$ is the number of leaves in $T_d$, $\lambda > 0$ controls the structural complexity of the resulting tree, and $\kappa > 0$ controls the complexity of the leaf-level linear models.

**Incremental Ridge Regression**  For ridge linear regression, solving the system from scratch incurs a cost of $\mathcal{O}(k^3)$

due to matrix factorization (e.g., Cholesky or QR decomposition). In recursive settings, such as regression trees, it is often desirable to update the solution incrementally as samples are added or removed, which substantially improves computational scalability. This can be efficiently achieved using Recursive Least Squares (RLS) algorithms (Potts, 2004), which maintain sufficient statistics and support $\mathcal{O}(k^2)$ updates per sample via the Sherman–Morrison formula (Sherman & Morrison, 1950). To improve numerical stability, we use Cholesky updates rather than inverse updates, as we detail below.

**Fast Rank-One Cholesky Updates**  We maintain numerical stability by using a Cholesky decomposition implementation. Consider a given ridge regression system $\mathbf{A}\boldsymbol{\beta} = \boldsymbol{b}$, where $\mathbf{A} = \mathbf{X}^\top \mathbf{X} + \kappa \mathbf{I}$. The regularized Gram matrix $\mathbf{A}$ is symmetric positive definite. Instead of forming $\mathbf{A}^{-1}$ explicitly, we factorize $\mathbf{A} = \mathbf{L}\mathbf{L}^\top$ using a pivot-free Cholesky decomposition (Benoit, 1924; Golub & Van Loan, 2013). The solution is then obtained by solving two triangular systems, $\mathbf{L}\boldsymbol{z} = \boldsymbol{b}$ and $\mathbf{L}^\top \boldsymbol{\beta} = \boldsymbol{z}$, a procedure that is efficient (Press et al., 1992) and numerically stable without the need for pivoting (Turing, 1948).

Since the RLS formulation requires frequent updates to this system, we efficiently handle rank-one modifications of the form

$$\mathbf{A}_{\text{new}} = \mathbf{A} \pm \boldsymbol{x}\boldsymbol{x}^\top,$$

by directly updating the Cholesky factor via a rank-one Cholesky update or downdate (Seeger, 2008), instead of recomputing the full factorization. This procedure takes $\mathcal{O}(k^2)$ time for a $k \times k$ system and is particularly useful when rows are incrementally added to or removed from $\mathbf{X}$. We adopt this technique to efficiently maintain the leaf statistics during the split process.

**Lookahead Completions**  When determining the next split to add to a partially constructed tree, CLARITree considers all possible next splits and selects the one that leads to the best tree when completed with a greedy subroutine. This strategy can be considered a type of lookahead (Babbar et al., 2025) and also corresponds to a rollout strategy for combinatorial optimization (Bertsekas et al., 1997) and the pilot method (Voß et al., 2005).

`CholeskyTree` denotes CLARITree's greedy completion subroutine (Algorithm 4) and also serves as a standalone baseline. While similar to classical model tree algorithms such as M5 (Quinlan, 1992), CholeskyTree differs fundamentally in that split selection is based on the downstream ridge-regularized linear regression objective rather than node means or variance reduction. Like CLARITree, CholeskyTree uses efficient rank-one Cholesky updates and downdates throughout the recursive tree construction, which is where its name comes from. To emphasize its role as the

greedy completion procedure inside CLARITree, we occasionally refer to it as Greedy CholeskyTree.

# 4. Algorithm Details

## 4.1. CLARITree Framework

This section describes how regression trees can be computed within our lookahead-based framework. This paradigm significantly improves scalability relative to globally optimal regression trees.

For the main paper, we focus on the simplified, polynomial-time one-step lookahead strategy, which is highly accurate, as shown in Section 6 (with some alternative settings of the algorithm discussed in Appendix F). More specifically, the algorithm explores all combinations of a single layer of splits, with Greedy CholeskyTree completion beyond this depth to find the best splitting position. For a dataset $D$, a given depth budget $d$, a structural complexity parameter $\lambda$, and a ridge regularization parameter $\kappa$. The objective is defined as:

$$\mathcal{L}_{\text{CLARITree}}(D, d, \lambda, \kappa) =$$
$$\begin{cases} \lambda + \text{LEAFOBJ}(D), & \text{if } d = 0, \\ \min\Big\{ \lambda + \text{LEAFOBJ}(D), \\ \quad \textit{Phase 1: Greedy CholeskyTree selection} \\ \quad (f^\star, \tau^\star) = \arg\min_{f,\tau} \big[ \mathcal{L}_g(D_{f \leq \tau}, d-1, \lambda, \kappa) \\ \quad\quad\quad + \mathcal{L}_g(D_{f > \tau}, d-1, \lambda, \kappa) \big], \\ \quad \textit{Phase 2: CLARITree completion} \\ \quad \mathcal{L}_{\text{CLARITree}}\big(D_{f^\star \leq \tau^\star}, d-1, \lambda, \kappa\big) \\ \quad + \mathcal{L}_{\text{CLARITree}}\big(D_{f^\star > \tau^\star}, d-1, \lambda, \kappa\big) \Big\}, & \text{if } d > 0. \end{cases}$$
(3)

where $D_{f \leq \tau}$ and $D_{f > \tau}$ denote the partitions induced by thresholding feature $f$ at value $\tau$, $\mathcal{L}_g(D, d-1, \lambda, \kappa)$ is the objective of a Greedy CholeskyTree of depth $d-1$ built on dataset $D$, and $\text{LEAFOBJ}(D)$ denotes the loss from fitting a linear model directly on $D$. The overall objective $\mathcal{L}^*(D, d, \lambda, \kappa)$ in (2) is approximated by $\mathcal{L}_{\text{CLARITree}}(D, d, \lambda, \kappa)$.

## 4.2. CLARITree Implementation

The framework above defines the recursive optimization objective, but an efficient implementation requires scalable split evaluation. We now present the CLARITree algorithm for constructing linear regression trees. The full procedure is summarized in Algorithm 1. A key component of the implementation is the use of rank-one Cholesky updates for efficient streaming split evaluation, as described in Algorithm 2.

---

**Algorithm 1** `CLARITree`: lookahead split selection with recursive refinement

**Require:** Node data $D_\mathcal{I}$; node sufficient statistics $(\mathbf{L}_\mathcal{I}, \boldsymbol{b}_\mathcal{I}, \|\boldsymbol{y}_\mathcal{I}\|^2)$; node-wise sorted lists $\Pi_\mathcal{I} = \{\pi_f(\mathcal{I})\}_{f \in \mathcal{F}}$; candidate threshold pool $\mathcal{P} = \{\mathcal{P}_f\}_{f \in \mathcal{F}}$; depth budget $d$

1: $\text{OBJ}_{leaf} \leftarrow \text{LEAFOBJ}(\mathbf{L}_\mathcal{I}, \boldsymbol{b}_\mathcal{I}, \|\boldsymbol{y}_\mathcal{I}\|^2)$
2: **if** $d = 0$ **then**
3: $\quad \boldsymbol{\beta} \leftarrow \texttt{SolveRidge}(\mathbf{L}_\mathcal{I}, \boldsymbol{b}_\mathcal{I})$
4: $\quad$ **return** $\text{Leaf}(\boldsymbol{\beta})$, $\text{OBJ}_{leaf}$
5: **end if**
6: $\text{OBJ}^*_{look} \leftarrow \infty, \quad (f^*, \tau^*) \leftarrow \texttt{None}$
7: $\mathcal{U} \leftarrow \texttt{EnumerateSplits}(D_\mathcal{I}, \Pi_\mathcal{I}, \mathcal{P}, \mathbf{L}_\mathcal{I}, \boldsymbol{b}_\mathcal{I},$
$\quad\quad \|\boldsymbol{y}_\mathcal{I}\|^2)$
8: **for** each tuple yielded by $\mathcal{U}$ **do**
9: $\quad$ Receive $(f, \tau, \mathcal{I}_L, \mathcal{I}_R, \mathbf{L}_L, \boldsymbol{b}_L, \|\boldsymbol{y}_{\mathcal{I}_L}\|^2$
$\quad\quad \mathbf{L}_R, \boldsymbol{b}_R, \|\boldsymbol{y}_{\mathcal{I}_R}\|^2)$
10: $\quad$ Construct child sorted lists $\Pi_{\mathcal{I}_L}, \Pi_{\mathcal{I}_R}$
11: $\quad$ Construct child data views $D_{\mathcal{I}_L}, D_{\mathcal{I}_R}$
12: $\quad$ **if** $d = 1$ **then**
13: $\quad\quad \text{OBJ}_{cand} \leftarrow \text{LEAFOBJ}(\mathbf{L}_L, \boldsymbol{b}_L, \|\boldsymbol{y}_{\mathcal{I}_L}\|^2) +$
$\quad \text{LEAFOBJ}(\mathbf{L}_R, \boldsymbol{b}_R, \|\boldsymbol{y}_{\mathcal{I}_R}\|^2)$
14: $\quad$ **else** ▷ Greedy completion of the remaining subtree.
15: $\quad\quad (\widetilde{T}_L, \widetilde{\text{OBJ}}_L) \leftarrow$
$\quad \texttt{CholeskyTree}(\mathbf{L}_L, \boldsymbol{b}_L, D_{\mathcal{I}_L}, \Pi_{\mathcal{I}_L}, \mathcal{P}, d-1)$
16: $\quad\quad (\widetilde{T}_R, \widetilde{\text{OBJ}}_R) \leftarrow$
$\quad \texttt{CholeskyTree}(\mathbf{L}_R, \boldsymbol{b}_R, D_{\mathcal{I}_R}, \Pi_{\mathcal{I}_R}, \mathcal{P}, d-1)$
17: $\quad\quad \text{OBJ}_{cand} \leftarrow \widetilde{\text{OBJ}}_L + \widetilde{\text{OBJ}}_R$
18: $\quad$ **end if**
19: $\quad$ **if** $\text{OBJ}_{cand} < \text{OBJ}^*_{look}$ **then**
20: $\quad\quad \text{OBJ}^*_{look} \leftarrow \text{OBJ}_{cand}, \quad (f^*, \tau^*) \leftarrow (f, \tau)$
21: $\quad\quad$ Store the corresponding child information for the current best split: $(\mathbf{L}_L^{best}, \boldsymbol{b}_L^{best}, D_{\mathcal{I}_L}^{best}, \Pi_{\mathcal{I}_L}^{best})$ and $(\mathbf{L}_R^{best}, \boldsymbol{b}_R^{best}, D_{\mathcal{I}_R}^{best}, \Pi_{\mathcal{I}_R}^{best})$
22: $\quad$ **end if**
23: **end for**
24: **if** $(f^*, \tau^*) = \texttt{None}$ **then**
25: $\quad \boldsymbol{\beta} \leftarrow \texttt{SolveRidge}(\mathbf{L}_\mathcal{I}, \boldsymbol{b}_\mathcal{I})$
26: $\quad$ **return** $\text{Leaf}(\boldsymbol{\beta})$, $\text{OBJ}_{leaf}$
27: **end if**
28: $(T_L, \text{OBJ}_L) \leftarrow \texttt{CLARITree}(\mathbf{L}_L^{best}, \boldsymbol{b}_L^{best}, D_{\mathcal{I}_L}^{best}, \Pi_{\mathcal{I}_L}^{best},$
$\quad \mathcal{P}, d-1)$
29: $(T_R, \text{OBJ}_R) \leftarrow \texttt{CLARITree}(\mathbf{L}_R^{best}, \boldsymbol{b}_R^{best}, D_{\mathcal{I}_R}^{best}, \Pi_{\mathcal{I}_R}^{best},$
$\quad \mathcal{P}, d-1)$
30: $\text{OBJ}_{subtree} \leftarrow \text{OBJ}_L + \text{OBJ}_R$
31: **if** $\text{OBJ}_{subtree} < \text{OBJ}_{leaf}$ **then**
32: $\quad$ **return** $\text{Node}(f^*, \tau^*, T_L, T_R)$, $\text{OBJ}_{subtree}$
33: **else**
34: $\quad \boldsymbol{\beta} \leftarrow \texttt{SolveRidge}(\mathbf{L}_\mathcal{I}, \boldsymbol{b}_\mathcal{I})$
35: $\quad$ **return** $\text{Leaf}(\boldsymbol{\beta})$, $\text{OBJ}_{leaf}$ ▷ Prune if the refined subtree does not improve the leaf.
36: **end if**

---

For notational simplicity, the treatment of the intercept is deferred to Appendix A.4. The main paper also presents a simplified version of the algorithms that omits the full treatment of the structural complexity parameter $\lambda$ and ridge regularization parameter $\kappa$. Complete pseudocode and additional technical details deferred to the Appendix C.

---

**Algorithm 2** `EnumerateSplits`: streamed split enumeration with Cholesky updates

---

**Require:** Node data $D_{\mathcal{I}}$; node-wise sorted lists $\Pi_{\mathcal{I}} = \{\pi_f(\mathcal{I})\}_{f \in \mathcal{F}}$; candidate threshold pool $\mathcal{P} = \{\mathcal{P}_f\}_{f \in \mathcal{F}}$; parent sufficient statistics $(\mathbf{L}_{\mathcal{I}}, \boldsymbol{b}_{\mathcal{I}}, \|\boldsymbol{y}_{\mathcal{I}}\|^2)$

 1: **for** each split feature $f$ **do**
 2:     Initialize the left child as empty: $\mathbf{L}_L \leftarrow \mathrm{chol}(\kappa I)$, $\boldsymbol{b}_L \leftarrow 0, \|\boldsymbol{y}_{\mathcal{I}_L}\|^2 \leftarrow 0, \mathcal{I}_L \leftarrow \emptyset$
 3:     Initialize the right child as the parent: $\mathbf{L}_R \leftarrow \mathbf{L}_{\mathcal{I}}$, $\boldsymbol{b}_R \leftarrow \boldsymbol{b}_{\mathcal{I}}, \|\boldsymbol{y}_{\mathcal{I}_R}\|^2 \leftarrow \|\boldsymbol{y}_{\mathcal{I}}\|^2$
 4:     $q \leftarrow 1$   ▷ Pointer into the sorted threshold set $\mathcal{P}_f$.
 5:     **for** $r = 1$ **to** $|\pi_f(\mathcal{I})| - 1$ **do**
 6:         $i \leftarrow \pi_f(\mathcal{I})[r]$   ▷ Move sample $i$ from the right child to the left child.
 7:         $\boldsymbol{b}_L \leftarrow \boldsymbol{b}_L + \boldsymbol{x}_i y_i, \qquad \boldsymbol{b}_R \leftarrow \boldsymbol{b}_R - \boldsymbol{x}_i y_i$
 8:         $\|\boldsymbol{y}_{\mathcal{I}_L}\|^2 \leftarrow \|\boldsymbol{y}_{\mathcal{I}_L}\|^2 + y_i^2, \qquad \|\boldsymbol{y}_{\mathcal{I}_R}\|^2 \leftarrow \|\boldsymbol{y}_{\mathcal{I}_R}\|^2 - y_i^2$
 9:         $\mathbf{L}_L \leftarrow \mathrm{cholupdate}(\mathbf{L}_L, \boldsymbol{x}_i, +1)$
10:         $\mathbf{L}_R \leftarrow \mathrm{cholupdate}(\mathbf{L}_R, \boldsymbol{x}_i, -1)$
11:         $v_{cur} \leftarrow x_{if}, \quad v_{next} \leftarrow x_{\pi_f(\mathcal{I})[r+1], f}$
12:         **if** $v_{cur} = v_{next}$ **then**
13:             **continue** ▷ No threshold can split identical feature values.
14:         **end if**
15:         **while** $q \leq |\mathcal{P}_f|$ **and** $\mathcal{P}_f[q] \leq v_{cur}$ **do**
16:             $q \leftarrow q + 1$
17:         **end while**
18:         **while** $q \leq |\mathcal{P}_f|$ **and** $\mathcal{P}_f[q] < v_{next}$ **do**   ▷ All thresholds in this interval reuse the same sufficient statistics.
19:             $\tau \leftarrow \mathcal{P}_f[q]$
20:             **yield** $(f, \tau, \mathcal{I}_L, \mathcal{I}_R, \mathbf{L}_L, \boldsymbol{b}_L, \|\boldsymbol{y}_{\mathcal{I}_L}\|^2, \mathbf{L}_R, \boldsymbol{b}_R, \|\boldsymbol{y}_{\mathcal{I}_R}\|^2)$   ▷ A stream that yields tuples for Algorithm 1's iterative loop
21:             $q \leftarrow q + 1$
22:         **end while**
23:     **end for**
24: **end for**

---

**Recursive CLARITree Construction.** Algorithm 1 implements the recursive CLARITree construction procedure. Line 1 evaluates the objective obtained by terminating the current node and fitting a leaf predictor directly. Lines 2–5 return this leaf solution when the depth budget is exhausted. If the depth budget is exhausted, the algorithm immediately returns this leaf solution.

Line 6 initializes the incumbent lookahead objective, so all feasible candidate splits are evaluated before a split is selected. lines 7–23 enumerate candidate splits using the `EnumerateSplits` subroutine and evaluate each split via greedy CholeskyTree completion. The split with the smallest completed objective is selected.

Finally, lines 24–36 recursively refine the left and right child nodes using CLARITree itself. The resulting refined subtree objective is then compared against the direct leaf objective, and the subtree is pruned whenever recursive refinement does not improve the leaf solution.

**Streaming Split Enumeration with Rank-One Cholesky Updates** Algorithm 2 implements the streaming split-enumeration procedure used throughout CLARITree. For each feature, samples are first traversed in sorted order, and candidate thresholds are evaluated incrementally as the threshold moves from left to right.

Lines 2–3 initialize the sufficient statistics and Cholesky factors associated with the left and right partitions. Lines 5–23 then stream through the sorted feature values and incrementally update the corresponding ridge-regression systems as samples are transferred between child nodes during the threshold sweep, rather than recomputing them from scratch.

More specifically, the main bottleneck in CLARITree is evaluating splits, since each threshold requires updating both child regressors. Rather than recomputing from scratch, we maintain the Cholesky factorization of the regularized Gram matrix $\mathbf{A} = \mathbf{X}^\top \mathbf{X} + \kappa \mathbf{I}$ together with the moment vector $\boldsymbol{b} = \mathbf{X}^\top \boldsymbol{y}$. As thresholds are scanned, samples move between child nodes; each move triggers a rank-one Cholesky update to $\mathbf{A}$ and $\boldsymbol{b}$, allowing both sides to be updated in $\mathcal{O}(k^2)$ time (see Remark 4.1). The ridge loss is then obtained directly from the Cholesky factor, without solving for coefficients explicitly (see Remark 4.2). During the sweep, we also maintain $\|\boldsymbol{y}_{\mathcal{I}_L}\|^2$ and $\|\boldsymbol{y}_{\mathcal{I}_R}\|^2$, the running sums of squared targets on the left and right subsets, which provide the $\|\boldsymbol{y}\|^2$ term in the ridge loss, where $\mathcal{I}_L$ and $\mathcal{I}_R$ are the current left and right child node index sets.

Building on these results, the `EnumerateSplits` subroutine (Algorithm 2) called in line 7 of Algorithm 1 presents our split-enumeration procedure. Beyond removing the main computational bottleneck, these incremental updates also allow us to directly handle continuous features without binarizing the dataset $D$, thereby reducing the complexity of a full threshold scan by an additional factor of $\mathcal{O}(n)$ (see Theorem 5.5).

Together, these techniques make CLARITree a practical and scalable linear extension of our framework. In addition to computational efficiency, we also analyze the numerical sta-

bility of the proposed rank-one Cholesky updates under our setting; see Appendix A.5 for details. Empirical results in Section 6 further demonstrate the effectiveness and stability of the approach.

**Remark** 4.1 (Rank-One Update for Cholesky). *Let* $\mathbf{A} = \mathbf{L}\mathbf{L}^\top$ *be the Cholesky factorization of a positive definite matrix* $\mathbf{A}$*. For any rank-one update or downdate* $\mathbf{A}' = \mathbf{A} \pm \boldsymbol{x}\boldsymbol{x}^\top$*, the updated Cholesky factor* $\mathbf{L}'$ *can be computed in* $\mathcal{O}(k^2)$ *time.*

**Remark** 4.2 (Ridge Loss from Cholesky). *Let* $\mathbf{A} = \mathbf{X}^\top\mathbf{X} + \kappa\mathbf{I} = \mathbf{L}\mathbf{L}^\top$ *and* $\boldsymbol{b} = \mathbf{X}^\top\boldsymbol{y}$*. Then*

$$\min_{\boldsymbol{\beta}} \|\mathbf{X}\boldsymbol{\beta} - \boldsymbol{y}\|^2 + \kappa\|\boldsymbol{\beta}\|^2 = \|\boldsymbol{y}\|^2 - \|\mathbf{L}^{-1}\boldsymbol{b}\|^2,$$

*so the loss can be evaluated in* $\mathcal{O}(k^2)$ *time using only the Cholesky factors.*

Proofs of these remarks are deferred to Appendix A.1.

**Special Case: Constant-LeafObj Variant of CLARITree.**
Setting the leaf coefficients in Equation (1) to $\boldsymbol{\beta} = \mathbf{0}$ recovers the standard *constant* regression tree model. Therefore, constant regression trees are a strict special case of our linear-leaf framework and inherit the same split-search and implementation benefits. We report additional algorithmic details, pseudocode, theorems, multi-step further variants, and experimental results for this setting in Appendix F.

## 5. Theoretical Analysis

We now establish the theoretical analysis of our algorithms. We first analyze the runtime and space complexity of CLARITree. Then, we show that our methods achieve lower objective values than Greedy CholeskyTree and can yield arbitrarily large improvements in MSE in some data distributions. $T$ denotes the number of thresholds per feature ($1 \leq T \leq n$). Finally, we show that rank-one Cholesky updates efficiently handle continuous feature updates.

**Complexity Analysis** We first analyze the runtime for CLARITree:

**Theorem 5.1** (Runtime for CLARITree). *Including a one-time presort of all features, the total runtime for CLARITree is* $\mathcal{O}\left(kn\log n + d^2nk^4T\right)$.

At each depth $d'$, CLARITree enumerates $\mathcal{O}(kT)$ candidate thresholds. Each candidate is evaluated by greedily completing the remaining tree of depth $d - d'$, which costs $\mathcal{O}((d - d')nk^3)$, with the rank-one Cholesky update reducing the cost by an $\mathcal{O}(k)$ factor. Summing over all depths $d' = 1, \ldots, d$ yields a total runtime of $\mathcal{O}(d^2nk^4T)$. We next analyze the space complexity for CLARITree:

**Theorem 5.2** (Space Complexity for CLARITree). *In typical regimes where* $n \gg dk$*, CLARITree requires no addi-*

*Table 1.* Overview of linear regression tree methods compared in this work.

| Method | Type | Reference |
|---|---|---|
| CholeskyTree | Heuristic | see Algorithm 4 |
| GUIDE | Heuristic | Loh (2002) |
| PILOT | Heuristic | Raymaekers et al. (2024) |
| M5 | Heuristic | Quinlan (1992) |
| STreeD-S | Optimal DP | Van Den Bos et al. (2024) |
| STreeD | Optimal DP | Van Den Bos et al. (2024) |

*tional asymptotic memory beyond storing the input data, and its space complexity during training is* $\mathcal{O}(nk)$*.*

The detailed proofs of both the runtime and space complexity results are deferred to Appendix B.1.

**Accuracy Analysis** Because Greedy CholeskyTree solutions are explored during the construction of CLARITree, we have:

**Theorem 5.3** (CLARITree Dominates Greedy). *CLARITree's returned tree always has objective* $\leq$ *that of Greedy CholeskyTree.*

Moreover, the improvement can be arbitrarily large:

**Theorem 5.4** (Arbitrary MSE Gap between CLARITree and Greedy). *For every depth budget* $d \geq 2$ *and* $\varepsilon \in (0, 1/2)$*, there exist data distributions such that* $\mathrm{MSE}_{Greedy}/\mathrm{MSE}_{CLARITree} \geq \frac{1}{4\varepsilon}$*.*

The construction of these distributions and the proof of the theorems are deferred to Appendix B.2.

**Efficient Split Evaluation for Continuous Features** We show that our split-evaluation routine can operate directly on continuous features. For comparison, we consider a binarization approach in which each continuous feature gives rise to $T$ binary split predicates, corresponding to its candidate thresholds $\tau$.

**Theorem 5.5** (Continuous Features vs. Binarization). *Compared to binarization-based approaches that enumerate* $T$ *thresholds per feature, operating directly on continuous features reduces the split-evaluation complexity by a factor of* $T$ *(and by* $T^3$ *compared to fully binarized regression).*

The proof is deferred to Appendix A.2.

## 6. Experiments

We now compare CLARITree to state-of-the-art optimal and greedy tree methods. Following prior work (Van Den Bos et al., 2024; Loh, 2002), we distinguish between split features and leaf-regression features. Continuous features are used both for splitting and in leaf regressors, whereas binary

*Table 2.* Out-of-sample performance comparison of piecewise linear regression tree methods (Depth = 4, Thresholds = 20). We report Test $R^2$ (mean $\pm$ std) for each dataset; the corresponding Test MSE ratio (relative to CLARITree) is shown in parentheses. Best results per dataset are highlighted in bold, and the second-best results are underlined. Dataset statistics include the number of instances $|D|$, original features $|\mathcal{F}|$, binary features $|\mathcal{F}_b|$, continuous features $|\mathcal{F}_c|$, and the size of the binarized feature set $|\mathcal{F}_{20}|$, where $\mathcal{F}_{20}$ denotes the feature set obtained by binarizing each continuous feature using 20 quantile-based thresholds. An asterisk ($*$) indicates that the selected configuration exceeded the training time limit (600s). Scientific notation is expressed using $e$ (e.g., $7.2e3 = 7.2 \times 10^3$). MSE ratios are omitted when the CLARITree MSE is numerically zero or nearly zero, rendering the ratio unstable.

| | | | | | | Methods | | | | | | |
|---|---|---|---|---|---|---|---|---|---|---|---|---|
| Dataset | $|D|$ | $|\mathcal{F}|$ | $|\mathcal{F}_b|$ | $|\mathcal{F}_c|$ | $|\mathcal{F}_{20}|$ | CLARITree | STreeD | STreeD-S | GUIDE | CholeskyTree | PILOT | M5 |
| *Small / Medium-scale datasets* | | | | | | | | | | | | |
| Airfoil | 1503 | 5 | 0 | 5 | 64 | 0.88 ± 0.01 (1.00) | **0.89 ± 0.01 (0.92)** | 0.76 ± 0.02 (2.00) | 0.84 ± 0.04 (1.33) | 0.87 ± 0.01 (1.08) | 0.51 ± 0.03 (4.08) | 0.54 ± 0.04 (3.83) |
| Auction | 2043 | 7 | 2 | 5 | 31 | **0.94 ± 0.00 (1.00)** | **0.94 ± 0.01 (1.00)** | 0.92 ± 0.03 (1.33) | 0.93 ± 0.01 (1.17) | 0.92 ± 0.03 (1.33) | 0.86 ± 0.03 (2.33) | 0.87 ± 0.02 (2.17) |
| Auto MPG | 392 | 7 | 0 | 7 | 98 | 0.84 ± 0.03 (1.00) | **0.85 ± 0.05 (0.94)** | 0.80 ± 0.05 (1.25) | 0.84 ± 0.04 (1.00) | 0.83 ± 0.03 (1.06) | 0.80 ± 0.03 (1.25) | 0.78 ± 0.04 (1.37) |
| Energy (Cooling) | 768 | 8 | 1 | 7 | 43 | 0.97 ± 0.01 (1.00) | **0.97 ± 0.00 (1.00)** | 0.97 ± 0.01 (1.00) | $-2.1e2 \pm 4.4e2$ (7.2e3) | **0.97 ± 0.00 (1.00)** | 0.89 ± 0.02 (3.67) | 0.95 ± 0.01 (1.67) |
| Energy (Heating) | 768 | 8 | 1 | 7 | 43 | **1.00 ± 0.00** | **1.00 ± 0.00** | **1.00 ± 0.00** | **1.00 ± 0.00** | **1.00 ± 0.00** | 0.92 ± 0.02 | 0.97 ± 0.00 |
| Insurance | 1338 | 9 | 6 | 3 | 51 | **0.86 ± 0.03 (1.00)** | **0.86 ± 0.03 (1.00)** | 0.85 ± 0.04 (1.07) | **0.86 ± 0.03 (1.00)** | **0.86 ± 0.03 (1.00)** | 0.83 ± 0.04 (1.21) | 0.84 ± 0.04 (1.14) |
| Optical Net | 630 | 7 | 1 | 6 | 93 | **0.92 ± 0.03 (1.00)** | 0.88 ± 0.04 (1.50) | 0.86 ± 0.01 (1.75) | 0.80 ± 0.10 (2.50) | 0.91 ± 0.03 (1.13) | 0.71 ± 0.17 (3.63) | 0.74 ± 0.03 (3.25) |
| Real Estate | 414 | 6 | 0 | 6 | 101 | **0.64 ± 0.09 (1.00)** | **0.64 ± 0.09 (1.00)** | 0.55 ± 0.09 (1.25) | 0.61 ± 0.06 (1.08) | **0.64 ± 0.09 (1.00)** | 0.55 ± 0.07 (1.25) | 0.53 ± 0.07 (1.31) |
| Servo | 167 | 2 | 0 | 2 | 7 | 0.51 ± 0.12 (1.00) | 0.51 ± 0.14 (1.00) | 0.48 ± 0.20 (1.06) | 0.49 ± 0.20 (1.04) | 0.51 ± 0.12 (1.00) | **0.52 ± 0.11 (0.98)** | **0.52 ± 0.11 (0.98)** |
| Synch | 557 | 4 | 0 | 4 | 80 | **1.00 ± 0.00** | **1.00 ± 0.00** | **1.00 ± 0.00** | **1.00 ± 0.00** | **1.00 ± 0.00** | **1.00 ± 0.00** | 0.99 ± 0.00 |
| Yacht | 308 | 6 | 0 | 6 | 58 | **1.00 ± 0.00** | **1.00 ± 0.00** | 0.99 ± 0.01 | **1.00 ± 0.00** | **1.00 ± 0.00** | 0.87 ± 0.02 | 0.99 ± 0.01 |
| *Large-scale datasets* | | | | | | | | | | | | |
| California Housing | 20433 | 13 | 5 | 8 | 165 | **0.75 ± 0.01 (1.00)** | 0.70 ± 0.01 (1.20)* | 0.66 ± 0.01 (1.36)* | 0.73 ± 0.01 (1.08) | 0.73 ± 0.01 (1.08) | 0.64 ± 0.01 (1.44) | 0.60 ± 0.01 (1.60) |
| Seoul Bike | 8760 | 9 | 0 | 9 | 131 | **0.72 ± 0.02 (1.00)** | 0.69 ± 0.02 (1.11)* | 0.63 ± 0.01 (1.32)* | 0.68 ± 0.03 (1.14) | 0.71 ± 0.02 (1.04) | 0.54 ± 0.01 (1.64) | 0.60 ± 0.02 (1.43) |
| Temperature (Max) | 7590 | 21 | 0 | 21 | 359 | **0.88 ± 0.01 (1.00)** | 0.82 ± 0.01 (1.50)* | 0.76 ± 0.01 (2.00)* | 0.84 ± 0.01 (1.33) | 0.84 ± 0.01 (1.33) | 0.78 ± 0.02 (1.83) | 0.72 ± 0.01 (2.33) |
| Temperature (Min) | 7590 | 21 | 0 | 21 | 359 | **0.89 ± 0.01 (1.00)** | 0.85 ± 0.01 (1.36)* | 0.83 ± 0.01 (1.55)* | 0.85 ± 0.01 (1.36) | 0.88 ± 0.01 (1.09) | 0.84 ± 0.01 (1.45) | 0.79 ± 0.01 (1.91) |
| Walmart | 6435 | 5 | 1 | 4 | 81 | **0.22 ± 0.02 (1.00)** | 0.21 ± 0.01 (1.01)* | 0.18 ± 0.01 (1.05)* | 0.18 ± 0.02 (1.05) | 0.19 ± 0.01 (1.04) | 0.02 ± 0.01 (1.26) | 0.10 ± 0.01 (1.15) |

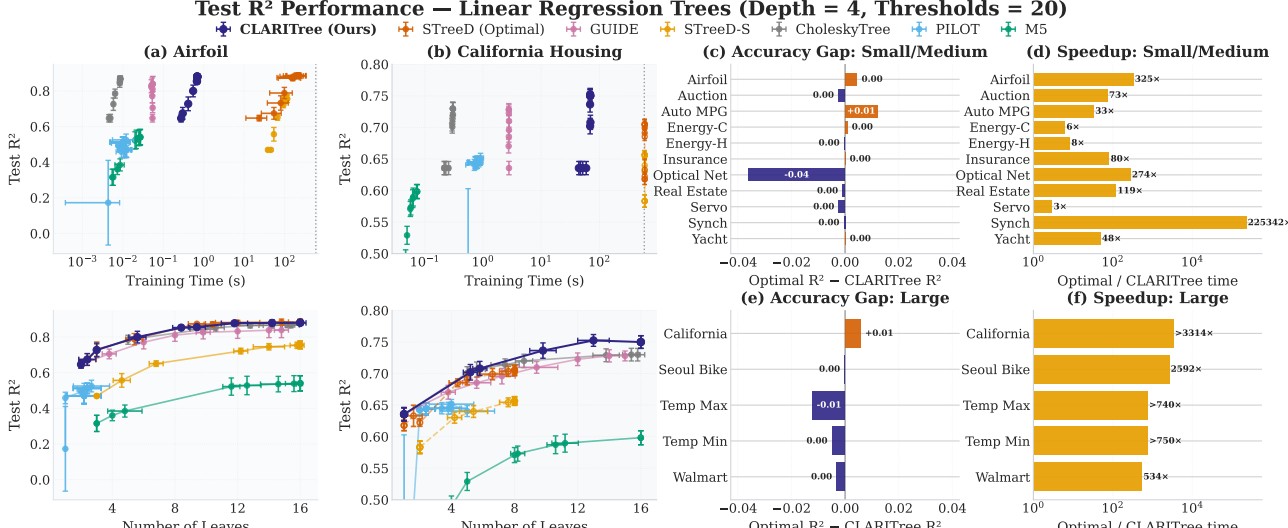

*Figure 2.* **Comparison of accuracy, complexity, and efficiency across representative datasets.** Panels (a)–(b) show the trade-off between test $R^2$ and training time (top) as well as test $R^2$ and the number of leaves (bottom) on two representative datasets. Our proposed **CLARITree** (blue) consistently achieves strong accuracy–sparsity trade-offs while remaining highly efficient, often approaching or closely matching the optimal **StreeD** (orange) and **StreeD-S** (yellow) solutions at substantially lower computational cost. Panels (c)–(f) further summarize the accuracy gap relative to the optimal solver and the corresponding speedup factors across small/medium and large-scale datasets. **GUIDE** (pink), **PILOT** (sky blue), **M5** (green), and **Greedy CholeskyTree** (gray) are included for reference. Dashed lines and hollow markers indicate runs that reached the prescribed time limit. The vertical dashed line in the top row denotes the default time limit of 10 minutes. For the large-scale experiments in Panels (e)–(f), we further increased the time limit for the optimal solver to 64 hours to evaluate whether **CLARITree** remains near-optimal under substantially larger computational budgets.

or one-hot encoded features are used for splitting only in our experiments. Ordinal or discretized non-one-hot variables are treated as continuous features.

## 6.1. Experimental Setup

**Datasets** We evaluate our methods on a total of 16 datasets, including datasets from the UCI Machine Learning Repository (Dua & Graff, 2017), the Medical Cost Personal dataset from Choi (2018), and two publicly available regres-

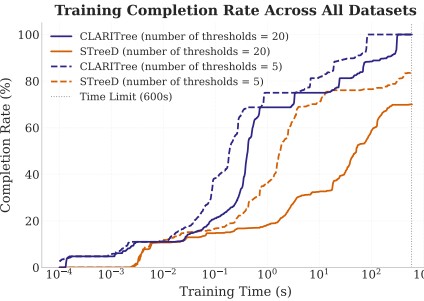

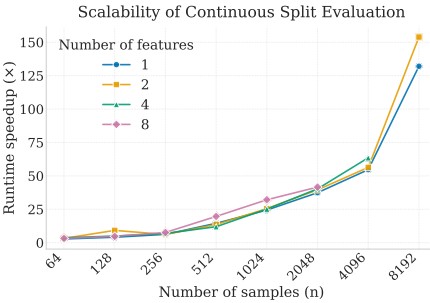

*Figure 3.* (a) **Training completion rate under a 10-minute budget.** Empirical completion curves aggregated across all datasets. The dashed vertical line marks the $600\,\mathrm{s}$ time limit used in our default protocol. (b) **Scalability of continuous split evaluation.** Ablation test on synthetic data showing speedup for CLARITree using rank-one updates relative to CLARITree without rank-one updates.

sion benchmarks hosted on Kaggle (Nugent, 2017; H, 2021), with a summary shown in Table 3 in Appendix D.

**Methods**   We compare CLARITree with six representative linear regression tree methods, summarized in Table 1. These include four greedy approaches and two optimal dynamic programming–based methods. As is done by both GUIDE and STreeD, we use binary features only for splitting and not as regressors in leaf models in experiments, which does not affect the complexity analysis described in Appendix B. STreeD-S (Van Den Bos et al., 2024) is an optimal regression-tree method that restricts each leaf to a univariate linear model, trading expressiveness for tractable optimization.

**Experimental Protocol**   All experiments are evaluated using five independent outer $80/20$ train–test splits. To vary structural complexity, we sweep over a range of leaf-penalty regularization parameters, thereby obtaining models of different sparsity and corresponding $R^2$ values. For each penalty and each split, hyperparameters for the leaf-level linear regressions are selected via 3-fold cross-validation on the training portion, using a grid search over the regularization constant. Note that GUIDE provides no leaf-level regression regularization parameters, as its node-wise linear

models are unregularized. We therefore control sparsity by sweeping over the maximum number of nodes. Similarly, PILOT does not offer explicit leaf-level regularization; instead, its complexity is governed by varying the minimum number of samples required in internal and leaf nodes. M5 provides no explicit control over the number of candidate thresholds per feature, and therefore always uses its default split enumeration strategy.

All methods are given a 10-minute training budget (the default timeout for the slowest method, STreeD). We set the maximum number of thresholds to 20, noting that some features have fewer than 20 available thresholds. To further evaluate scalability and near-optimality, we additionally run STreeD with an extended 64-hour time budget on large-scale datasets, allowing the optimal solver sufficient time to approach its best attainable solutions while providing a stress-test comparison against our method; these results are reported later in Figure 2. Further, we also report results using 5 thresholds and full threshold enumeration to provide an ablation study on the effect of the threshold budget and to better understand the relationship between predictive performance and the number of candidate thresholds. Further experimental details, including preprocessing steps, are presented in Appendix D, and full threshold results are shown in Appendix E.

### 6.2. Experimental Results

We summarize out-of-sample performance in Table 2. For each dataset, we sweep the leaf penalty to obtain models at different sparsity levels and report the best test $R^2$ (mean $\pm$ std over 5 folds) across the sparsity levels for each method. We additionally report the test MSE ratio relative to CLARITree to make accuracy differences comparable across datasets. Across the benchmark suite, CLARITree achieves the best or ties for best test $R^2$ on nearly all datasets (with the exception of Servo, a very small and simple dataset where all methods exhibit high std and unstable performance), while remaining within the 10-minute default time limit.

To characterize the trade-offs among sparsity, runtime, accuracy, and scalability in large, high-dimensional settings, Figure 2 presents both representative dataset trajectories and aggregate benchmark comparisons. Panels (a)–(b) plot two representative datasets (one medium-scale and one large-scale). The top row shows test $R^2$ versus training time, while the bottom row shows test $R^2$ versus the number of leaves (i.e., sparsity). CLARITree traces a consistently favorable Pareto frontier: for a given runtime budget, it attains higher test $R^2$, and for a given sparsity level, it achieves accuracy comparable to STreeD while outperforming GUIDE, PILOT, and M5 in both accuracy and stability. Panels (c)–(f) further summarize the aggregate accuracy gap relative to

the optimal solver and the corresponding speedup factors across the benchmark suite for both small/medium-scale and large-scale datasets.

Complete experimental results and additional ablations are presented in Appendix E.

### 6.2.1. HOW DOES CLARITREE COMPARE TO STATE-OF-THE-ART PIECEWISE LINEAR REGRESSION TREES

On large-scale datasets, Table 2 shows that CLARITree consistently delivers the strongest out-of-sample performance under the default 10-minute budget. On small/medium-scale datasets where optimal solvers are feasible, CLARITree remains highly competitive, often matching the best test $R^2$ within standard deviation. Beyond accuracy, we evaluate practical solvability within a 600-second time budget, showing the proportion of problems solved by our method and the SOTA solver at each moment. Figure 3 (a) reports the training completion rate aggregated across all datasets. While CLARITree reliably completes the target training runs within 600 seconds, the optimal solver STreeD frequently exceeds the time limit and completes only about 60% of runs, even with 20-threshold binarization, improving to only around 80% even under the more favorable 5-threshold setting. This directly highlights the scalability bottleneck of exact search for large datasets.

Figure 2 further illustrates that CLARITree navigates the trade-off between runtime and accuracy competitively with an optimal method. Crucially, on large-scale datasets such as California Housing and Temperature Max/Min, the optimal solver frequently exhausts the 10-minute limit and can only search to shallower trees (depth 2–3), whereas CLARITree reliably completes the target depth-4 model. Even when extending the runtime budget to 64 hours, STreeD still fails to certify optimal solutions on datasets such as California Housing and Temperature Max/Min. This gap directly reveals the scalability advantage of our approach: it achieves near-optimal accuracy when exact search is tractable and remains effective on large datasets where exact methods cannot instantiate the desired model class within practical budgets.

Overall, CLARITree is never more than approximately $0.01$ below the optimal solver in observed test $R^2$, while achieving over two orders of magnitude speedup on most datasets relative to the optimal solver.

### 6.2.2. SCALABILITY OF CONTINUOUS FEATURE SPLITS

Figure 3(b) compares CLARITree against an otherwise identical baseline that does not use rank-one Cholesky updates. The experiment uses synthetic datasets with controllable sample size and feature dimension, as described in Appendix D.3. As the sample size $n$ increases, CLARITree achieves substantial speedups that grow steadily, reaching improvements of several orders of magnitude across all tested settings of the number of features ($= 1, 2, 4, 8$). These results empirically validate the improvement predicted by Theorem 5.5 and confirm the scalability advantage of handling continuous features directly.

## 7. Conclusion

We presented a family of algorithms for piecewise regression trees. These algorithms enable users to efficiently find trees near global optima, improving the accessibility and scalability of accurate, glass-box models. Future work could include extending our algorithms to find scalable approximations of a set of near-optimal regression trees (perhaps approximating the Rashomon set of these trees, see Xin et al., 2022). One could also incorporate the accelerated depth-two solver from the MurTree family of approaches (Demirović et al., 2022; Van Den Bos et al., 2024) to decide the final splits of the tree, potentially improving quality at some additional computational cost.

## Acknowledgment

This material is based upon work supported by the National Institutes of Health/NIDA grant number R01DA054994. We acknowledge the support of the Natural Sciences and Engineering Research Council of Canada (NSERC). Nous remercions le Conseil de recherches en sciences naturelles et en génie du Canada (CRSNG) de son soutien.

## Impact Statement

This paper advances the field of interpretable machine learning, which is essential for (and central to) trustworthy AI.

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

**Roadmap.** In Appendix A, we introduce Cholesky rank-one updates and downdates, explain why we focus directly on continuous features for regression problems, and describe the implementation of the intercept. In Appendix B, we analyze the algorithmic complexity and accuracy. In Appendix C, we present pseudocode for the key subroutines. In Appendix D, we summarize the experimental setup, including data preparation, platforms, and hardware. In Appendix E, we report extensive experimental results in detail. Finally, in Appendix F, we introduce a representative variant of CLARITree and show that, even with constant leaves, the method maintains strong performance and scalability.

# A. Preliminaries on Incremental Regression Updates

Before presenting the full complexity analysis, we introduce two technical ingredients that underlie our algorithms: efficient Cholesky rank-one updates/downdates and direct handling of continuous features without one-hot expansion.

## A.1. Rank-One Cholesky Updates and Downdates

Given a ridge regression system

$$\mathbf{A} = \mathbf{X}^\top \mathbf{X} + \kappa \mathbf{I},$$

with Cholesky factorization $\mathbf{A} = \mathbf{L}\mathbf{L}^\top$, adding or removing one row $x$ corresponds to the rank-one modification

$$\mathbf{A}' = \mathbf{A} \pm \boldsymbol{x}\boldsymbol{x}^\top.$$

Instead of recomputing the factorization, we update $\mathbf{L}$ via a rank-one Cholesky update/downdate in $\mathcal{O}(k^2)$ time (Remark A.1). This reduces the cost of evaluating splits from $\mathcal{O}(k^3)$ per node to $\mathcal{O}(k^2)$ per moved sample.

*Remark* A.1 (Update Recursive Cholesky Ridge, see Gill et al. 1974; Seeger 2008, recalling Remark 4.1). Let $\{(\boldsymbol{x}_i, y_i)\}_{i=1}^t$ be a data stream with $x_i \in \mathbb{R}^k$ and ridge parameter $\kappa > 0$. Define

$$\mathbf{A}_t := \kappa \mathbf{I} + \sum_{i=1}^t \boldsymbol{x}_i \boldsymbol{x}_i^\top, \quad \boldsymbol{b}_t := \sum_{i=1}^t \boldsymbol{x}_i y_i, \quad \boldsymbol{\beta}_t := \mathbf{A}_t^{-1} \boldsymbol{b}_t.$$

Suppose $\mathbf{A}_{t-1} = \mathbf{L}_{t-1}\mathbf{L}_{t-1}^\top$ is available. Then

$$\mathbf{L}_t \leftarrow \mathsf{cholupdate}(\mathbf{L}_{t-1}, \boldsymbol{x}_t, +1), \qquad \boldsymbol{b}_t = \boldsymbol{b}_{t-1} + \boldsymbol{x}_t y_t,$$

and $\boldsymbol{\beta}_t$ is obtained by two triangular solves with $\mathbf{L}_t$. Each update costs $\mathcal{O}(k^2)$ time.

*Proof Sketch.* We have $\mathbf{A}_t = \mathbf{A}_{t-1} + \boldsymbol{x}_t \boldsymbol{x}_t^\top = \mathbf{L}_{t-1}(\mathbf{I} + \boldsymbol{u}\boldsymbol{u}^\top)\mathbf{L}_{t-1}^\top$, where $\boldsymbol{u} = \mathbf{L}_{t-1}^{-1}\boldsymbol{x}_t$. Thus $\mathbf{L}_t = \mathbf{L}_{t-1}\mathbf{R}$ with $\mathbf{R}\mathbf{R}^\top = \mathbf{I} + \boldsymbol{u}\boldsymbol{u}^\top$. The factor $\mathbf{R}$ is computed stably by $\mathsf{cholupdate}$, which applies a sequence of Givens-type rotations to maintain triangular structure; this routine is implemented in numerical libraries such as MATLAB's `cholupdate` and Eigen's `LLT::rankUpdate`. The moment vector $\boldsymbol{b}_t$ is updated additively, and $\boldsymbol{\beta}_t$ follows by solving $\mathbf{L}_t\mathbf{L}_t^\top \boldsymbol{\beta}_t = \boldsymbol{b}_t$ via forward and backward substitution. The overall complexity is $\mathcal{O}(k^2)$ per update (Gill et al., 1974). □

Having established that the Cholesky factor $\mathbf{L}$ can be updated incrementally, we next show that the ridge regression loss itself can be evaluated directly from $\mathbf{L}$ without explicitly computing the coefficients. Thus, the evaluation also takes only $k^2$ time.

*Remark* A.2 (Cholesky-based Ridge Regression Loss Evaluation, recalling Remark 4.2). Following the notation in Remark A.1, let $\mathbf{A} = \mathbf{X}^\top \mathbf{X} + \kappa \mathbf{I} = \mathbf{L}\mathbf{L}^\top$ be the regularized Gram matrix with Cholesky factor $\mathbf{L}$, and let $\boldsymbol{b} = \mathbf{X}^\top \boldsymbol{y}$. Then the minimum value of the ridge regression objective

$$\mathcal{L}(\mathbf{X}, \boldsymbol{y}) = \min_{\boldsymbol{\beta} \in \mathbb{R}^k} \|\mathbf{X}\boldsymbol{\beta} - \boldsymbol{y}\|^2 + \kappa \|\boldsymbol{\beta}\|^2$$

is given by

$$\mathcal{L}(\mathbf{X}, \boldsymbol{y}) = \|\boldsymbol{y}\|^2 - \|\mathbf{L}^{-1}\boldsymbol{b}\|^2.$$

*Proof.* The ridge regression solution is

$$\boldsymbol{\beta}^* = (\mathbf{X}^\top \mathbf{X} + \kappa \mathbf{I})^{-1}\mathbf{X}^\top \boldsymbol{y} = \mathbf{A}^{-1}\boldsymbol{b}.$$

The corresponding loss value is

$$\mathcal{L}(\mathbf{X}, \boldsymbol{y}) = \|\boldsymbol{y}\|^2 - 2\boldsymbol{b}^\top \boldsymbol{\beta}^* + (\boldsymbol{\beta}^*)^\top \mathbf{A} \boldsymbol{\beta}^* = \|\boldsymbol{y}\|^2 - \boldsymbol{b}^\top \mathbf{A}^{-1} \boldsymbol{b}.$$

Given the Cholesky decomposition $\mathbf{A} = \mathbf{L}\mathbf{L}^\top$, we have $\mathbf{A}^{-1} = \mathbf{L}^{-\top}\mathbf{L}^{-1}$, so

$$\boldsymbol{b}^\top \mathbf{A}^{-1} \boldsymbol{b} = \boldsymbol{b}^\top \mathbf{L}^{-\top}\mathbf{L}^{-1}\boldsymbol{b} = \|\mathbf{L}^{-1}\boldsymbol{b}\|^2,$$

which gives the desired result. □

### A.2. Why Handle Continuous Features Directly Instead of via Binarization

A naive binarization based on binary indicator predicates (e.g., features of the form $f < \tau$) over all candidate thresholds would inflate the feature dimension from $k$ to $kT$, where $T$ denotes the number of candidate split thresholds per feature. We now formalize the complexity gap between continuous regression with Cholesky rank-one updates and downdates, binarized predicates with regression on the original feature space, and fully binarized regression.

**Theorem A.3** (Complexity of Continuous Features vs. Binarization). *Let $n$ be the number of samples, $k$ the number of original features, and $T$ the number of candidate thresholds per continuous feature. Then:*

1. *Continuous regression with rank-one Cholesky updates and downdates costs $\mathcal{O}(nk^3)$.*

2. *Binarized predicates with regression on the original $k$ features cost $\mathcal{O}(nk^3 T)$.*

3. *Fully binarized regression on $kT$ features costs $\mathcal{O}(nk^3 T^3)$.*

*Proof.* For case 1, scanning all thresholds on one continuous feature using Cholesky rank-one updates and downdates requires $\mathcal{O}(nk^2)$ time, and across $k$ features the total cost is $\mathcal{O}(nk^3)$.

For case 2, if each threshold is treated as a separate binarized predicate, we obtain $kT$ predicates. Each predicate requires $\mathcal{O}(nk^2)$ work, while leaf regressions still depend on the original $k$ features, yielding a total cost of $\mathcal{O}(nk^3 T)$.

For case 3, if the design matrix itself is expanded to $kT$ binarized features and regression is performed directly in this enlarged space, each update costs $\mathcal{O}((kT)^2)$. With $kT$ such features, the total cost scales as $\mathcal{O}(n(kT)^3) = \mathcal{O}(nk^3 T^3)$.

Thus, binarization introduces multiplicative factors in $T$, whereas direct handling of continuous features via rank-one updates remains $\mathcal{O}(nk^3)$. True binary features ($T = 1$) incur no additional overhead. □

*Remark* A.4. Preprocessing costs differ between continuous features and binarization.

For **continuous features**, each of the $k$ features requires a one-time presort of its $n$ sample values to determine the admissible thresholds. In the worst case where $T \approx n$ distinct values are present, this costs $\mathcal{O}(kn \log n)$. If each feature has only $T$ distinct values, the same result can be achieved in $\mathcal{O}(kn + kT \log T)$ time by first aggregating identical values. After this presort step, all subsequent threshold evaluations are performed by linear sweeps with rank-one updates, and no further sorting is required.

For **binarization**, no sorting is required. Instead, constructing the $kT$ binarized indicator features costs $\mathcal{O}(nkT)$.

In both cases, these preprocessing costs are asymptotically dominated by the regression complexities in Theorem A.3, and are therefore omitted from the main runtime bounds.

### A.3. Incremental versus Bulk Ridge Solvers

We now compare two Cholesky-based strategies for solving ridge regression. When the number of thresholds is small (for example, when a feature is binary or degenerates to a binary case), it may be more efficient to form the Gram matrix once and perform a direct Cholesky factorization, rather than updating the factorization incrementally via rank-one updates. We formalize this comparison below.

**Theorem A.5** (Incremental vs. Bulk Ridge Solver). *Let $\mathbf{X} \in \mathbb{R}^{n \times k}$, $\boldsymbol{y} \in \mathbb{R}^n$, and $\kappa > 0$. Define the ridge Gram matrix $\mathbf{G} = \kappa \mathbf{I}_k + \mathbf{X}^\top \mathbf{X}$ and let $\widehat{\boldsymbol{\beta}} = \arg\min_{\boldsymbol{\beta}} \|\mathbf{X}\boldsymbol{\beta} - \boldsymbol{y}\|_2^2 + \kappa \|\boldsymbol{\beta}\|_2^2$. Consider two computational strategies:*

***(i) Incremental (rank-1 update).*** *Starting from* $\mathbf{L}_0 = \sqrt{\kappa}\mathbf{I}_k$, *apply a sequence of rank-one Cholesky rank-one updates/downdates* $\mathbf{L}_i\mathbf{L}_i^\top = \mathbf{L}_{i-1}\mathbf{L}_{i-1}^\top + \boldsymbol{x}_i\boldsymbol{x}_i^\top$, *followed by a single triangular solve to obtain* $\widehat{\boldsymbol{\beta}}$.

***(ii) Bulk (GEMM+Cholesky).*** *First compute G using a Level-3 BLAS operation (e.g.,* `syrk`/`gemm`*), then perform a blocked Cholesky factorization* $\mathbf{G} = \mathbf{L}\mathbf{L}^\top$, *and finally solve* $\mathbf{L}\mathbf{L}^\top\widehat{\boldsymbol{\beta}} = \mathbf{X}^\top\boldsymbol{y}$.

*Their respective arithmetic complexities are:*

$$\text{FLOPs}_{inc} = \tfrac{3}{2}nk^2 + \mathcal{O}(nk), \qquad \textit{(Gill et al., 1974)},$$

$$\text{FLOPs}_{bulk} = 2nk^2 + \tfrac{1}{3}k^3 + \mathcal{O}(nk^2), \qquad \textit{(Golub \& Van Loan, 2013, §4.1.2)}.$$

*Hence,*

$$\text{FLOPs}_{inc} < \text{FLOPs}_{bulk} \tag{*}$$

*Proof.* Subtracting the leading-order terms gives $2nk^2 + \tfrac{1}{3}k^3 - \tfrac{3}{2}nk^2 = \tfrac{1}{2}nk^2 + \tfrac{1}{3}k^3$. This expression is non-negative. $\square$

*Remark* A.6 (Arithmetic Intensity and Runtime Implications). Rank-one updates operate as Level-2 BLAS kernels with arithmetic intensity $\text{AI} \approx 1$, making them memory-bound. In contrast, the bulk route uses Level-3 BLAS with intensity $\text{AI} \approx k \gg 1$, making them compute-bound.

Under the Roofline model, the effective runtime satisfies

$$T_{\text{inc}} \approx \frac{\tfrac{3}{2}nk^2}{\mathcal{B}}, \qquad T_{\text{bulk}} \approx \frac{2nk^2 + \tfrac{1}{3}k^3}{\mathcal{P}},$$

where $\mathcal{B}$ is memory bandwidth (bytes/sec) and $\mathcal{P}$ is peak fused multiply–add throughput (FLOPs/sec). Because $\mathcal{P}/\mathcal{B} \gg 1$ on modern CPUs/GPUs, the bulk approach can outperform the incremental one in wall-clock time, even when it performs more total FLOPs. This effect is especially prominent when $k$ is large and $\mathbf{X}$ fits in cache-optimized blocks.

## A.4. Handling Intercepts in Cholesky Rank-One Updates

**Theorem A.7** (Ridge with an unpenalized intercept: Cholesky loss and updates). *Let* $\mathbf{X} \in \mathbb{R}^{n \times k}$, $\boldsymbol{y} \in \mathbb{R}^n$, $\mathbf{1} \in \mathbb{R}^n$ *be the all-ones vector, and* $\kappa > 0$. *Consider the ridge objective with an* unpenalized *intercept*

$$\min_{\beta_0 \in \mathbb{R}, \boldsymbol{\beta} \in \mathbb{R}^k} \|\boldsymbol{y} - \beta_0 \mathbf{1} - \mathbf{X}\boldsymbol{\beta}\|_2^2 + \kappa\|\boldsymbol{\beta}\|_2^2. \tag{A1}$$

*Define the augmented design and (block-diagonal) regularizer*

$$\mathbf{Z} := [\mathbf{1}, \mathbf{X}] \in \mathbb{R}^{n \times (k+1)}, \qquad \Lambda := \text{diag}(0, \kappa, \ldots, \kappa) \in \mathbb{R}^{(k+1) \times (k+1)}.$$

*Let*

$$\mathbf{H} := \mathbf{Z}^\top\mathbf{Z} + \Lambda, \qquad \boldsymbol{t} := \mathbf{Z}^\top\boldsymbol{y}.$$

*Then:*

1. $\mathbf{H}$ *is positive definite for* $n \geq 1$, *hence admits a Cholesky factorization* $\mathbf{H} = \mathbf{L}\mathbf{L}^\top$.

2. *The minimizer equals* $\boldsymbol{\theta}^* = [\beta_0^*; \boldsymbol{\beta}^*] = \mathbf{H}^{-1}\boldsymbol{t}$, *and the minimum value is*

$$\boldsymbol{y}^\top\boldsymbol{y} - \|\mathbf{L}^{-1}\boldsymbol{t}\|_2^2. \tag{A2}$$

3. *(Incremental updates.) For adding/removing a sample* $(\boldsymbol{x}, y) \in \mathbb{R}^k \times \mathbb{R}$, *let* $\widetilde{\boldsymbol{x}} := \begin{bmatrix} 1 \\ \boldsymbol{x} \end{bmatrix}$. *Then*

$$\mathbf{H} \leftarrow \mathbf{H} \pm \widetilde{\boldsymbol{x}}\widetilde{\boldsymbol{x}}^\top, \qquad \boldsymbol{t} \leftarrow \boldsymbol{t} \pm y\widetilde{\boldsymbol{x}},$$

*so the new optimum can be evaluated by a rank-one Cholesky update/downdate of* $\mathbf{L}$ *in* $O((k+1)^2)$ *time, followed by (A2).*

*Proof.* Writing $\boldsymbol{\theta} = [\beta_0; \boldsymbol{\beta}] \in \mathbb{R}^{k+1}$, the objective (A1) is

$$\|\boldsymbol{y} - \mathbf{Z}\boldsymbol{\theta}\|_2^2 + \boldsymbol{\theta}^\top \Lambda \boldsymbol{\theta} = \boldsymbol{y}^\top \boldsymbol{y} - 2\boldsymbol{\theta}^\top \mathbf{Z}^\top \boldsymbol{y} + \boldsymbol{\theta}^\top (\mathbf{Z}^\top \mathbf{Z} + \Lambda)\boldsymbol{\theta} = \boldsymbol{y}^\top \boldsymbol{y} - 2\boldsymbol{\theta}^\top \boldsymbol{t} + \boldsymbol{\theta}^\top \mathbf{H}\boldsymbol{\theta}.$$

We first prove that $\mathbf{H}$ is positive definite. For any nonzero vector $\boldsymbol{\alpha}$,

$$\boldsymbol{\alpha}^\top \mathbf{H}\boldsymbol{\alpha} = \boldsymbol{\alpha}^\top \mathbf{Z}^\top \mathbf{Z}\boldsymbol{\alpha} + \boldsymbol{\alpha}^\top \Lambda \boldsymbol{\alpha}.$$

The second term equals zero only when $\boldsymbol{\alpha} = (a, 0, \ldots, 0)$. In this case, the first term becomes $a^2 n$, which is still positive. Thus $\mathbf{H}$ is positive definite and a Cholesky factor exists.

Since $\mathbf{H}$ is symmetric, the unique stationary point satisfies $\mathbf{H}\boldsymbol{\theta} = \boldsymbol{t}$, hence $\boldsymbol{\theta}^* = \mathbf{H}^{-1}\boldsymbol{t}$. Completing the square (or substituting $\boldsymbol{\theta}^*$) yields that the minimum value is

$$\boldsymbol{y}^\top \boldsymbol{y} - \boldsymbol{t}^\top \mathbf{H}^{-1} \boldsymbol{t}.$$

With $\mathbf{H} = \mathbf{L}\mathbf{L}^\top$, $\mathbf{H}^{-1} = \mathbf{L}^{-\top}\mathbf{L}^{-1}$ and thus $\boldsymbol{t}^\top \mathbf{H}^{-1} \boldsymbol{t} = \|\mathbf{L}^{-1}\boldsymbol{t}\|_2^2$, proving (A2).

For the incremental statement, note that appending/removing one row $(\widetilde{\boldsymbol{x}}^\top, y)$ changes $\mathbf{Z}^\top \mathbf{Z}$ by $\pm \widetilde{\boldsymbol{x}}\widetilde{\boldsymbol{x}}^\top$ and $\mathbf{Z}^\top \boldsymbol{y}$ by $\pm y\widetilde{\boldsymbol{x}}$, leaving $\Lambda$ unchanged. Rank-one Cholesky update/downdate on $\mathbf{H}$ gives the new $\mathbf{L}$ in $O((k+1)^2)$, after which the minimum is evaluated by (A2). □

*Remark* A.8 (Engineering note). In code, handling an unpenalized intercept reduces to two surgical changes: (i) augment features with a column of ones, i.e. use $\mathbf{Z} = [\mathbf{1}, \mathbf{X}]$ everywhere (updates, solves, prediction); (ii) replace $\kappa I$ by $\Lambda = \mathrm{diag}(0, \kappa, \ldots, \kappa)$. When initializing a child's factorization on an empty set, $\Lambda$ alone is not strictly positive definite due to the leading zero; for numerical robustness use a tiny floor, e.g. $\Lambda_\varepsilon = \mathrm{diag}(\varepsilon, \kappa, \ldots, \kappa)$ with $\varepsilon = 10^{-12}$. With this, every rank-one update uses $\widetilde{\boldsymbol{x}} = \begin{bmatrix} 1 \\ \boldsymbol{x} \end{bmatrix}$ and every loss is computed by $\|\boldsymbol{y}\|_2^2 - \|\mathbf{L}^{-1}\boldsymbol{t}\|_2^2$ with $\boldsymbol{t} = \mathbf{Z}^\top \boldsymbol{y}$.

## A.5. Numerical Stability of Rank-One Updates and Downdates

To further address the numerical stability of rank-one Cholesky update/downdate, we analyze why these operations remain stable.

In practice, we include a nonzero regularization term $\kappa$ for numerical stability. Empirically, Cholesky downdates never resulted in non-positive definite matrices in our experiments; even $\kappa = 10^{-12}$ yields no instability. In addition, we normalize the design matrix, which ensures that the condition number of

$$\mathbf{A} = \mathbf{X}^\top \mathbf{X} + \kappa \mathbf{I}$$

remains bounded.

To be more specific, we present a theoretical analysis of the error accumulation in Cholesky rank-one updates and quantify the resulting error bound for the linear system.

Throughout, we use the spectral norm $\|\cdot\|_2$ for matrices and the Euclidean norm for vectors.

**Theorem A.9** (Error of Cholesky Rank-One Update). *Let*

$$\mathbf{A}_t = \mathbf{A}_{t-1} + \boldsymbol{x}_t \boldsymbol{x}_t^\top, \quad \mathbf{A}_0 = \kappa \mathbf{I}, \quad \mathbf{A}_t \succ 0,$$

*and let $\mathbf{A}_t = \mathbf{L}_t \mathbf{L}_t^\top$ denote its Cholesky factorization.*

*We denote by $\widetilde{\mathbf{L}}_t$ the numerically computed Cholesky factor obtained via rank-one updates:*

$$\widetilde{\mathbf{L}}_t = \mathrm{cholupdate}(\widetilde{\mathbf{L}}_{t-1}, \boldsymbol{x}_t),$$

*where* $\mathrm{cholupdate}(\cdot, \boldsymbol{x}_t)$ *performs a rank-one Cholesky update corresponding to adding $\boldsymbol{x}_t \boldsymbol{x}_t^\top$. Then*

$$\widetilde{\mathbf{L}}_t \widetilde{\mathbf{L}}_t^\top = \mathbf{A}_t + \mathbf{E}_t,$$

*with*

$$\|\mathbf{E}_t\| \leq (1 + cu)\|\mathbf{E}_{t-1}\| + cu(\|\mathbf{A}_{t-1}\| + \|\boldsymbol{x}_t\|^2).$$

*Furthermore,*

$$\|\mathbf{E}_t\| = \mathcal{O}\left(tu\max_{r \leq t}\|\mathbf{A}_r\|\right),$$

*where $u$ is the machine varepsilon.*

*Proof.* Cholesky rank-one update can be viewed as a sequence of Givens rotations, as shown in Seeger (2008):

$$\mathbf{L}_t = \mathbf{Q}_k\cdots\mathbf{Q}_1\mathbf{X}, \quad \mathbf{X} = [\mathbf{L}_{t-1}, \boldsymbol{x}_t].$$

Givens rotations to be backward stable. Each computed transformation satisfies

$$\mathrm{fl}(\mathbf{Q}_i\mathbf{A}) = (\mathbf{Q}_i + \Delta\mathbf{Q}_i)\mathbf{A}, \quad \|\Delta\mathbf{Q}_i\| \leq cu.$$

Assume that

$$\mathbf{A}_{t-1} = \widetilde{\mathbf{L}}_{t-1}\widetilde{\mathbf{L}}_{t-1}^\top - \mathbf{E}_{t-1}.$$

Then the computed update is

$$\widetilde{\mathbf{L}}_t = (\mathbf{Q}_k + \Delta\mathbf{Q}_k)\cdots(\mathbf{Q}_1 + \Delta\mathbf{Q}_1)[\widetilde{\mathbf{L}}_{t-1}, \boldsymbol{x}_t].$$

Expanding and neglecting $O(u^2)$ terms,

$$\widetilde{\mathbf{L}}_t = \mathbf{Q}_k\cdots\mathbf{Q}_1[\widetilde{\mathbf{L}}_{t-1}, \boldsymbol{x}_t] + \Delta\mathbf{L}_t,$$

with

$$\|\Delta\mathbf{L}_t\| \leq cu\|[\widetilde{\mathbf{L}}_{t-1}, \boldsymbol{x}_t]\| \leq cu(\|\widetilde{\mathbf{L}}_{t-1}\| + \|\boldsymbol{x}_t\|).$$

Define

$$\widetilde{\mathbf{L}}_t = \widehat{\mathbf{L}}_t + \Delta\mathbf{L}_t,$$

where

$$\widehat{\mathbf{L}}_t = \mathbf{Q}_p\cdots\mathbf{Q}_1[\widetilde{\mathbf{L}}_{t-1}, \boldsymbol{x}_t].$$

Then

$$\widehat{\mathbf{L}}_t\widehat{\mathbf{L}}_t^\top = \widetilde{\mathbf{L}}_{t-1}\widetilde{\mathbf{L}}_{t-1}^\top + \boldsymbol{x}_t\boldsymbol{x}_t^\top = \mathbf{A}_{t-1} + \mathbf{E}_{t-1} + \boldsymbol{x}_t\boldsymbol{x}_t^\top = \mathbf{A}_t + \mathbf{E}_{t-1}.$$

Thus,

$$\widetilde{\mathbf{L}}_t\widetilde{\mathbf{L}}_t^\top = (\widehat{\mathbf{L}}_t + \Delta\mathbf{L}_t)(\widehat{\mathbf{L}}_t + \Delta\mathbf{L}_t)^\top = \mathbf{A}_t + \mathbf{E}_t,$$

where

$$\mathbf{E}_t = \widehat{\mathbf{L}}_t\Delta\mathbf{L}_t^\top + \Delta\mathbf{L}_t\widehat{\mathbf{L}}_t^\top + \Delta\mathbf{L}_t\Delta\mathbf{L}_t^\top + \mathbf{E}_{t-1}.$$

Taking norms,

$$\|\mathbf{E}_t\| \leq 2\|\widehat{\mathbf{L}}_t\|\|\Delta\mathbf{L}_t\| + \|\Delta\mathbf{L}_t\|^2 + \|\mathbf{E}_{t-1}\|.$$

Substituting the bound on $\|\Delta\mathbf{L}_t\|$ and neglecting $O(u^2)$ term,

$$\|\mathbf{E}_t\| \leq cu\|\widehat{\mathbf{L}}_t\|(\|\widetilde{\mathbf{L}}_{t-1}\| + \|\boldsymbol{x}_t\|) + \|\mathbf{E}_{t-1}\|.$$

Now,

$$\|\widehat{\mathbf{L}}_t\|^2 = \|\mathbf{A}_t + \mathbf{E}_{t-1}\|, \quad \|\widetilde{\mathbf{L}}_{t-1}\|^2 \leq \|\mathbf{A}_{t-1}\| + \|\mathbf{E}_{t-1}\|.$$

Using $\mathbf{A}_t = \mathbf{A}_{t-1} + \boldsymbol{x}_t \boldsymbol{x}_t^\top$, together with

$$\sqrt{a+b} \leq \sqrt{a} + \sqrt{b}, \quad ab \leq \frac{a^2 + b^2}{2},$$

we obtain

$$\|\mathbf{E}_t\| \leq cu\|\mathbf{A}_t + \mathbf{E}_{t-1}\|^{1/2} \left(\|\mathbf{A}_{t-1}\|^{1/2} + \|\mathbf{E}_{t-1}\|^{1/2} + \|\boldsymbol{x}_t\|\right) + \|\mathbf{E}_{t-1}\|.$$

Applying the inequalities again,

$$\|\mathbf{E}_t\| \leq cu \left(\|\mathbf{A}_t + \mathbf{E}_{t-1}\| + \|\mathbf{A}_{t-1}\| + \|\mathbf{E}_{t-1}\| + \|\boldsymbol{x}_t\|^2\right) + \|\mathbf{E}_{t-1}\|.$$

Then

$$\|\mathbf{E}_t\| \leq cu \left(\|\mathbf{A}_t\| + \|\mathbf{E}_{t-1}\| + \|\mathbf{A}_{t-1}\| + \|\mathbf{E}_{t-1}\| + \|\boldsymbol{x}_t\|^2\right) + \|\mathbf{E}_{t-1}\|.$$

Thus,

$$\|\mathbf{E}_t\| \leq cu \left(\|\mathbf{A}_{t-1}\| + \|\mathbf{E}_{t-1}\| + \|\boldsymbol{x}_t\|^2\right) + \|\mathbf{E}_{t-1}\|.$$

Therefore,

$$\|\mathbf{E}_t\| \leq (1 + cu)\|\mathbf{E}_{t-1}\| + cu(\|\mathbf{A}_{t-1}\| + \|\boldsymbol{x}_t\|^2).$$

Unrolling the recursion,

$$\|\mathbf{E}_t\| \leq (1 + cu)^t \|\mathbf{E}_0\| + cu \sum_{r=1}^{t} (1 + cu)^{t-r}(\|\mathbf{A}_{r-1}\| + \|\boldsymbol{x}_r\|^2).$$

Since $\mathbf{A}_0 = \kappa\mathbf{I}$ is diagonal, its Cholesky factorization introduces only $\mathbf{E}_0 = \mathcal{O}(u)$ error. Thus, as $u$ is extremely small, we conclude

$$\|\mathbf{E}_t\| = \mathcal{O}\left(tu \max_{r \leq t} \|\mathbf{A}_r\|\right).$$

$\square$

**Theorem A.10** (Error Bound for the Linear Coefficients). *Let*

$$\mathbf{A}_t = \mathbf{X}_t^\top \mathbf{X}_t + \kappa\mathbf{I}, \qquad \boldsymbol{b}_t = \mathbf{X}_t^\top \boldsymbol{y}_t,$$

*and define the exact and computed coefficients by*

$$\boldsymbol{\beta}_t = \mathbf{A}_t^{-1} \boldsymbol{b}_t, \qquad \widehat{\boldsymbol{\beta}}_t = (\mathbf{A}_t + \mathbf{E}_t)^{-1} \boldsymbol{b}_t,$$

*where the perturbation matrix $\mathbf{E}_t$ satisfies*

$$\|\mathbf{E}_t\|_2 = O\left(tu \max_{r \leq t} \|\mathbf{A}_r\|_2\right).$$

*Assume*

$$\|\mathbf{A}_t^{-1}\|_2 \|\mathbf{E}_t\|_2 < 1.$$

*Then*

$$\frac{\|\widehat{\boldsymbol{\beta}}_t - \boldsymbol{\beta}_t\|_2}{\|\boldsymbol{\beta}_t\|_2} = O(tu\,\kappa_2(\mathbf{A}_t)),$$

*where $\kappa_2(\cdot)$ is the conditional number. Moreover, if the columns of $\mathbf{X}_t$ are normalized so that*

$$\kappa_2(\mathbf{A}_t) = O(k/\kappa),$$

*then*

$$\frac{\|\widehat{\boldsymbol{\beta}}_t - \boldsymbol{\beta}_t\|_2}{\|\boldsymbol{\beta}_t\|_2} = O\left(\frac{tuk}{\kappa}\right).$$

*Finally, since the Cholesky factorization is recomputed independently at each node, coefficient errors do not accumulate along the tree. Hence the worst-case coefficient error is bounded by*

$$O\left(\frac{unk}{\kappa}\right).$$

*Proof.* Let

$$\mathbf{A} = \mathbf{A}_t, \qquad \Delta\mathbf{A} = \mathbf{E}_t, \qquad \boldsymbol{b} = \boldsymbol{b}_t, \qquad \Delta\boldsymbol{b} = \mathbf{0}.$$

Then the exact and computed coefficients satisfy

$$\mathbf{A}\boldsymbol{\beta}_t = \boldsymbol{b}_t, \qquad (\mathbf{A} + \mathbf{E}_t)\widehat{\boldsymbol{\beta}}_t = \boldsymbol{b}_t.$$

We apply Theorem 7.2 with $\varepsilon$ such that

$$\|\Delta\mathbf{A}\|_2 \le \varepsilon\|\mathbf{A}\|_2, \qquad \|\Delta\boldsymbol{b}\|_2 \le \varepsilon\|0\|_2.$$

Since $\Delta\boldsymbol{b} = \mathbf{0}$, the second inequality is trivial, and we may take

$$\varepsilon = \frac{\|\mathbf{E}_t\|_2}{\|\mathbf{A}_t\|_2}.$$

Therefore, adding we know that $\varepsilon\kappa_2(\mathbf{A}_t) = \|\mathbf{A}_t^{-1}\|_2\|\mathbf{E}_t\|_2 < 1$, by Theorem 7.2 in Higham (2002),

$$\frac{\|\widehat{\boldsymbol{\beta}}_t - \boldsymbol{\beta}_t\|_2}{\|\boldsymbol{\beta}_t\|_2} \le \frac{\varepsilon}{1 - \varepsilon\|\mathbf{A}_t^{-1}\|_2\|\mathbf{A}_t\|_2}\left(\frac{\|\mathbf{A}_t^{-1}\|_2\|0\|_2}{\|\boldsymbol{\beta}_t\|_2} + \|\mathbf{A}_t^{-1}\|_2\|\mathbf{A}_t\|_2\right).$$

Now note that

$$\|\mathbf{A}_t^{-1}\|_2\|\mathbf{A}_t\|_2 = \kappa_2(\mathbf{A}_t).$$

Hence,

$$\frac{\|\widehat{\boldsymbol{\beta}}_t - \boldsymbol{\beta}_t\|_2}{\|\boldsymbol{\beta}_t\|_2} = \mathcal{O}\left(\varepsilon\kappa_2(\mathbf{A}_t)\right).$$

Using the previous theorem,

$$\|\mathbf{E}_t\|_2 = \mathcal{O}\left(tu\max_{r\le t}\|\mathbf{A}_r\|_2\right).$$

Since $\mathbf{A}_r$ and $\mathbf{A}_t$ are of the same order along the update path, we obtain

$$\varepsilon = \frac{\|\mathbf{E}_t\|_2}{\|\mathbf{A}_t\|_2} = \mathcal{O}(tu).$$

Substituting this into the bound above yields

$$\frac{\|\widehat{\boldsymbol{\beta}}_t - \boldsymbol{\beta}_t\|_2}{\|\boldsymbol{\beta}_t\|_2} = \mathcal{O}\left(tu\,\kappa_2(\mathbf{A}_t)\right).$$

If the columns of $\mathbf{X}_t$ are normalized, then

$$\|\mathbf{X}_t^\top\mathbf{X}_t\|_2 = \mathcal{O}(k),$$

and since

$$\mathbf{A}_t = \mathbf{X}_t^\top\mathbf{X}_t + \kappa\mathbf{I},$$

its smallest eigenvalue is at least $\kappa$. Therefore,

$$\kappa_2(\mathbf{A}_t) = \frac{\lambda_{\max}(\mathbf{A}_t)}{\lambda_{\min}(\mathbf{A}_t)} = O(k/\kappa).$$

Consequently,

$$\frac{\|\widehat{\boldsymbol{\beta}}_t - \boldsymbol{\beta}_t\|_2}{\|\boldsymbol{\beta}_t\|_2} = O\left(\frac{tuk}{\kappa}\right).$$

Finally, the Cholesky factorization is recomputed from scratch at each tree node rather than passed recursively down the tree. Hence numerical errors in the coefficients do not accumulate across nodes. Since the number of iterations is bounded by the sample size $n$, the worst-case bound is

$$O\left(\frac{unk}{\kappa}\right).$$

$\square$

*Remark* A.11. The condition $\|\mathbf{A}_t^{-1}\|_2\|\mathbf{E}_t\|_2 < 1$ is reasonable. In our setting, we have

$$\|\mathbf{A}_t^{-1}\|_2\|\mathbf{E}_t\|_2 = \mathcal{O}\left(tu\kappa_2(\mathbf{A}_t)\right) = \mathcal{O}\left(\frac{tuk}{\kappa}\right) \ll 1,$$

since $u$ is the machine precision.

# B. Detailed Theoretical Analysis

This section provides detailed proofs and supporting arguments for the theoretical results in the main paper. We first analyze the computational complexity of CLARITree. We then establish its performance improvement over Greedy CholeskyTree and show that, under suitable data distributions, this improvement can be arbitrarily large, yielding an arbitrary MSE gap.

## B.1. Complexity Analysis

We first present both runtime and space complexity and emphasize scalability. As noted in Section 6, binary features are used only for splitting and do not enter the regression model. Consequently, the effective dimensionality in all regression and complexity analyses is given by the number of continuous features. For notational simplicity, we denote this number by $k$ throughout and treat all features as continuous; the regression case with binary splitting is recovered by setting $T = 1$.

**Lemma B.1** (Complexity for Greedy CholeskyTree). *Let $n$ be the number of training examples, $k$ the number of continuous features, $T$ the number of thresholds per feature, and $d$ the total tree depth. Including a one-time global presort of cost $\mathcal{O}(kn \log n)$, the total runtime of Greedy CholeskyTree is*

$$\mathcal{O}\left(kn \log n + dnk^3\right).$$

*Proof.* For the first level, the computational cost comes from scanning all thresholds and fitting the corresponding linear regressions to evaluate the MSE. For each feature, computing all the linear regressions across its thresholds requires $\mathcal{O}(nk^2)$ operations, and evaluating the resulting MSEs is also bounded by $\mathcal{O}(nk^2)$. Therefore, across all $k$ features, the total complexity at the first level is $\mathcal{O}(nk^3)$.

At a given level, the worst-case scenario occurs when all nodes are active, that is, every node is waiting to be split, and no branch has stopped early. In this case, the total number of training instances across all subproblems at that level remains $n$. For each subproblem containing $n_{\text{sub}}$ samples, the computational cost of evaluating all possible splits is $\mathcal{O}(n_{\text{sub}}k^3)$. Summing over all active nodes gives a total cost of $\mathcal{O}(nk^3)$ for that level. Repeating this process over $d$ levels yields an overall worst-case complexity of $\mathcal{O}(dnk^3)$.

including the one-time global presort of cost $\mathcal{O}(kn \log n)$, we obtain the final runtime bound

$$\mathcal{O}\left(kn \log n + dnk^3\right).$$

$\square$

**Theorem B.2** (Runtime for CLARITree, recalling Theorem 5.1). *Let $n$ be the number of training examples, $k$ the number of continuous features, $T$ the number of thresholds per feature, and $d$ the total tree depth. Including a one-time global presort of cost $\mathcal{O}(kn \log n)$, the total runtime for CLARITree is*

$$\mathcal{O}\left(kn \log n + d^2 nk^4 T\right).$$

*Proof.* At each depth, the cost naturally decomposes into two parts: (i) the cost of enumerating and scoring all candidate splits, and (ii) the cost of greedily completing the remaining $(d - t)$ levels for each candidate.

At the root ($d' = 1$), there are $kT$ split candidates $(f_j, \tau)$, representing feature–threshold pairs. However, we evaluate all thresholds of a given feature in a single linear sweep using rank-one Cholesky updates and downdates, so one sweep per feature suffices to score all its thresholds. Moving one instance triggers both a left and a right update, resulting in a per-feature cost of $2nk^2$. Consequently, across $k$ features, the total root enumeration cost is $\mathcal{O}(2nk^3)$.

Conditioned on any fixed root candidate $(f_j, \tau)$, the remaining $(d - 1)$ levels are built greedily. Based on Lemma B.1, the complexity of building Greedy CholeskyTree is bounded by $\mathcal{O}\left((d - 1)nk^3\right)$. Since there are $kT$ possible root candidates, the contribution of the Greedy CholeskyTree is $\mathcal{O}\left((d - 1)nk^3(kT)\right)$. Combining both parts yields:

$$\mathcal{O}(2nk^3) + \mathcal{O}\left((d - 1)nk^3(kT)\right).$$

At the second layer ($d' = 2$), the two children contain $\alpha n$ and $(1 - \alpha)n$ samples, which again sum to $n$. Hence, the cost of enumerating all features is still $\mathcal{O}(2nk^3)$. For each candidate we must greedily complete $(d - 2)$ layers, leading to

$$\mathcal{O}(2nk^3) + \mathcal{O}\left((d - 2)nk^3(kT)\right).$$

Since this argument is linear in $n$, the same reasoning applies at every depth. Recursively, at depth $d'$ the cost is

$$\mathcal{O}(2nk^3) + \mathcal{O}\big((d - d')nk^3(kT)\big).$$

Summing over all depths $d' = 1, \ldots, d$, the total runtime is

$$\sum_{d'=1}^{d} \big[\mathcal{O}(2nk^3) + \mathcal{O}\big((d - d')nk^3(kT)\big)\big] = \mathcal{O}\left(2dnk^3 + \tfrac{d(d-1)}{2}nk^4T\right).$$

Since the quadratic term dominates, and including the one-time global presort of cost $\mathcal{O}(kn \log n)$, we obtain the final runtime bound

$$\mathcal{O}\left(kn \log n + d^2 nk^4 T\right).$$

$\square$

**Theorem B.3** (Space Complexity for CLARITree, recalling Theorem 5.2). *Let $n$ be the number of training examples, $k$ the number of continuous features, and $d$ the maximum tree depth. In typical regimes where $n \gg dk$, CLARITree requires no additional asymptotic memory beyond storing the input data, and its space complexity during training is $\mathcal{O}(nk)$.*

*Proof.* The design matrix $\mathbf{X}$ is stored using $\mathcal{O}(nk)$ memory, while the response vector $\boldsymbol{y}$ requires $\mathcal{O}(n)$. Each Cholesky factor requires $\mathcal{O}(k^2)$ space. Since the algorithm proceeds via depth-first search, at any given time there are at most $d$ Cholesky factors simultaneously active on the recursion stack, resulting in a total of $\mathcal{O}(dk^2)$ working memory. In typical regimes, this quantity is easily dominated by $\mathcal{O}(nk)$.

Each node stores a regression coefficient vector of size $\mathcal{O}(k)$. The total number of leaves is at most $2^d$, which is further bounded by $n$, since each leaf must contain at least one instance. Therefore, the total memory required to store all regression coefficients and intercepts is bounded by $\mathcal{O}(nk)$.

Therefore, the space complexity at any time is $\mathcal{O}(nk)$, and no additional asymptotic memory beyond the input storage is required. $\square$

## B.2. Accuracy Analysis

In this section, we first establish that CLARITree outperforms Greedy CholeskyTree. We further show that there exists a data distribution under which the MSE gap between CLARITree and Greedy CholeskyTree can be made arbitrarily large, in the sense that

$$\frac{\text{MSE}_{\text{Greedy}}}{\text{MSE}_{\text{CLARITree}}} \geq \frac{1}{4\varepsilon},$$

for any $\varepsilon \in (0, 1/2)$.

### B.2.1. CLARITREE DOMINATES GREEDY TREES

Before presenting the construction, we first prove a simple observation: CLARITree consistently outperforms Greedy CholeskyTree.

**Theorem B.4** (CLARITree Dominates Greedy, recalling Theorem 5.3). *For any dataset, CLARITree's returned tree always has objective $\leq$ that of Greedy CholeskyTree.*

*Proof.* Fix a depth budget $d$. For any node with data subset $D$ and remaining depth $t \leq d$, let $C_t(D)$ denote the minimum objective value attainable by CLARITree on $(D, t)$, and let $G_t(D)$ denote the objective value attained by Greedy CholeskyTree on $(D, t)$. We prove by induction on $t$ that

$$C_t(D) \leq G_t(D) \quad \text{for all } D.$$

**Base case ($t = 0$).** Both methods return the same leaf model (since no further splits are allowed), hence $C_0(D) = G_0(D)$.

**Inductive step.** Assume that $C_{t-1}(\cdot) \leq G_{t-1}(\cdot)$ holds for depth $t-1$. Consider running Greedy CholeskyTree on $(D, t)$ and let it choose a split $(f^\star, \tau^\star)$ at the root, yielding left and right subsets $D_L^\star$ and $D_R^\star$. By the definition of CholeskyTree,

$$G_t(D) = G_{t-1}(D_L^\star) + G_{t-1}(D_R^\star).$$

CLARITree evaluates all candidate splits, which include $(f^\star, \tau^\star)$. Suppose CLARITree selects the split $(f^{\star\star}, \tau^{\star\star})$, inducing subsets $D_L^{\star\star}$ and $D_R^{\star\star}$. Then we have

$$\begin{aligned}
C_t(D) &= C_{t-1}(D_L^{\star\star}) + C_{t-1}(D_R^{\star\star}) \\
&\leq G_{t-1}(D_L^{\star\star}) + G_{t-1}(D_R^{\star\star}) \\
&\leq G_{t-1}(D_L^\star) + G_{t-1}(D_R^\star) \\
&= G_t(D).
\end{aligned}$$

The first equality follows from the definition of CLARITree. The second inequality follows from the induction hypothesis. The third inequality holds because CLARITree selects the split $(f^{\star\star}, \tau^{\star\star})$ that minimizes the objective under full Greedy completion, whereas $(f^\star, \tau^\star)$ is chosen to be optimal only for the one-step Greedy split. Consequently, the Greedy-completed tree induced by $(f^{\star\star}, \tau^{\star\star})$ cannot be worse than that induced by $(f^\star, \tau^\star)$.

Finally, we have $C_d(D_{\text{root}}) \leq G_d(D_{\text{root}})$ for the full dataset. $\qquad \square$

### B.2.2. AN ARBITRARY MSE GAP FOR CLARITREE

In this section, we show that for CLARITree, there exist data distributions on which its performance can be made arbitrarily better than that of a Greedy CholeskyTree. We focus on continuous features and targets evaluated under the MSE.

**Data Generating Process (DGP)** Fix a depth budget $d \geq 2$ and choose an integer $U > d$. Let the feature vector be

$$X = (g, h, M, z) \in \{-1, 1\}^{1+1+2U} \times \mathbb{R},$$

where all binary coordinates are independent Rademacher variables, $X_i \sim \text{Rad}(\frac{1}{2})$. Specifically,

$$g := X_1, \qquad h := X_2, \qquad M := (X_3, \dots, X_{2U+2}),$$

and the continuous feature $z \sim \mathcal{N}(0, 1)$ is independent of $(g, h, M)$.

We further decompose the nuisance block $M$ into $U$ independent pairs:

$$M = \{(m_{j1}, m_{j2})\}_{j=1}^U, \qquad (m_{j1}, m_{j2}) := (X_{2j+1}, X_{2j+2}).$$

Let $B \sim \text{Ber}(\varepsilon)$, where $\varepsilon > 0$, and $J \sim \text{Unif}\{1, \dots, U\}$, independent of all features. Define the response $Y$ by the mixture

$$Y = \begin{cases} ghz, & \text{if } B = 0, \\ m_{J1} m_{J2}, & \text{if } B = 1. \end{cases}$$

**Lemma B.5.** *Let $(Y, X)$ with $X = (g, h, M, z)$ follow the DGP definition, where $M$ are referred to as* nuisance variables. *Fix a $t$-step path $\{m_{k,i}\}_{k \leq U, i \in \{1,2\}}$, and let $A$ be any event measurable with respect to the corresponding $\sigma$-field,*

$$A \in \sigma(\{m_{k,i}\}_{k=1,\dots,t,\, i \in \{1,2\}}).$$

*Under the linear regression setup, splitting on any of $g$, $h$, or $z$ yields zero gain, whereas splitting on a new coordinate $m_{j,1}$ always yields a positive gain.*

*Proof.* **Split on $g$ or $h$.** We only discuss splitting on $g$, since the argument for $h$ is identical. Let $s \in \{-1, +1\}$ and define $A_s := A \cap \{g = s\}$. Recall that $Y = (1 - B)T + BN$ with $T = ghz$ and $N = m_{J1} m_{J2}$. In node $A_s$, the linear leaf model class consists of functions:

$$f(X) = \beta_0 + \beta_g s + \beta_h h + \beta_z z + \boldsymbol{\beta}_M^\top M,$$

where $M$ collects all nuisance features.

For any such $f$,

$$\mathbb{E}[(Y - f(X))^2 \mid A_s] = \mathbb{E}[Y^2 \mid A_s] + \mathbb{E}[f(X)^2 \mid A_s] - 2\mathbb{E}[Yf(X) \mid A_s].$$

The term $\mathbb{E}[Y^2 \mid A_s]$ does not depend on $s$, since $T^2 = z^2$ and $s^2 = 1$. Moreover, $\beta_g s$ is constant on $A_s$ and can be absorbed into the intercept, so the attainable minimum of $\mathbb{E}[f(X)^2 \mid A_s]$ is independent of $s$.

For the cross term, we decompose:

$$\mathbb{E}[Yf(X) \mid A_s] = \mathbb{E}[(1 - B)Tf(X) \mid A_s] + \mathbb{E}[BNf(X) \mid A_s].$$

Since $A_s \in \sigma(M, g)$ and $(h, z) \perp (M, g)$, we have

$$\mathbb{E}[Tf(X) \mid A_s] = 0$$

for any linear $f$, by expanding $f$ and using $\mathbb{E}[h] = \mathbb{E}[z] = 0$ together with independence. Hence

$$\mathbb{E}[(1 - B)Tf(X) \mid A_s] = 0.$$

We next show that the population-optimal coefficient on $g$ is zero. Consider the parent node $A \in \sigma(M)$, so that $g \perp A$. Let $f_A^\star$ be the $L_2(A)$-projection of $Y$ onto the linear span of $(1, g, h, z, M)$. By the normal equations, the optimal coefficient $\beta_g^\star$ satisfies

$$\mathbb{E}[(Y - f_A^\star(X)) g \mid A] = 0.$$

Now $\mathbb{E}[g \mid A] = 0$ and $g$ is independent of $(h, z, M, B, J)$ given $A$. Moreover,

$$\mathbb{E}[Yg \mid A] = \mathbb{E}[(1 - B) ghz \cdot g \mid A] + \mathbb{E}[BN \cdot g \mid A] = \mathbb{E}[(1 - B) hz \mid A] + \mathbb{E}[BN \mid A]\mathbb{E}[g \mid A] = 0,$$

since $\mathbb{E}[h] = \mathbb{E}[z] = 0$ and $BN \perp g$. Also $\mathbb{E}[hg \mid A] = \mathbb{E}[zg \mid A] = 0$ and $\mathbb{E}[M_i g \mid A] = 0$ for each coordinate $M_i$. Therefore, the normal equation for $g$ reduces to

$$\beta_g^\star \mathbb{E}[g^2 \mid A] = 0,$$

and hence $\beta_g^\star = 0$ because $\mathbb{E}[g^2 \mid A] = 1$.

Consequently, adding $g$ as a regressor does not reduce the MSE in node $A$. Conditioning on $g = s$ only reparametrizes the intercept in the child node $A_s$, and therefore the minimal conditional MSE is identical in $A$ and $A_s$:

$$\min_f \mathbb{E}[(Y - f(X))^2 \mid A_s] = \min_f \mathbb{E}[(Y - f(X))^2 \mid A].$$

Thus, splitting on $g$ (or $h$) yields zero gain.

**Split on $z$.** Fix a node event $A \in \sigma(M)$ and a threshold $\tau \in \mathbb{R}$, and define

$$A_\tau := A \cap \{z \leq \tau\}, \qquad A_\tau^c := A \cap \{z > \tau\}.$$

Recall that

$$Y = (1 - B)ghz + BN,$$

and consider linear leaf predictors of the form

$$f(X) = \beta_0 + \beta_g g + \beta_h h + \boldsymbol{\beta}_M^\top M + \beta_z z.$$

We first determine the population-optimal predictor in node $A$. Let $W := (1, g, h, M)$ and write $z = \widetilde{z} + \mathbb{E}[z \mid A]$ with $\widetilde{z} := z - \mathbb{E}[z \mid A]$. Since $1 \in W$, any predictor can be written as $f(X) = \theta^\top W + \beta_z \widetilde{z}$ after absorbing the constant $\beta_z \mathbb{E}[z \mid A]$ into the intercept.

For $b \in \mathbb{R}$, define
$$\mathcal{L}_A(b) := \mathbb{E}\big[(Y - \theta^\top W - b\widetilde{z})^2 \mid A\big].$$

Expanding,
$$\mathcal{L}_A(b) = \mathcal{L}_A(0) - 2b\,\mathbb{E}[(Y - \theta^\top W)\widetilde{z} \mid A] + b^2 \mathbb{E}[\widetilde{z}^2 \mid A].$$

Since $A \in \sigma(M)$ and $z \perp (g, h, M, B, J)$, conditioning on $A$ preserves independence between $\widetilde{z}$ and $(g, h, M, B, J)$. Hence $\mathbb{E}[(\theta^\top W)\widetilde{z} \mid A] = 0$. Moreover,
$$\mathbb{E}[Y\widetilde{z} \mid A] = \mathbb{E}[(1 - B)ghz\,\widetilde{z} \mid A] + \mathbb{E}[BN\,\widetilde{z} \mid A].$$

The second term is zero since $BN$ is independent of $z$. For the first term, $(1 - B) \perp (g, h, z)$ and $(g, h) \perp z$ under $A$, and $\mathbb{E}[gh] = 0$, implying $\mathbb{E}[(1 - B)ghz\,\widetilde{z} \mid A] = 0$. Therefore $\mathbb{E}[(Y - \theta^\top W)\widetilde{z} \mid A] = 0$ and
$$\mathcal{L}_A(b) = \mathcal{L}_A(0) + b^2 \mathbb{E}[\widetilde{z}^2 \mid A].$$

Since $\mathbb{E}[\widetilde{z}^2 \mid A] > 0$, the minimum is attained at $b = 0$, so the optimal predictor in node $A$ satisfies $\beta_z = 0$.

We now repeat the same calculation in node $A_\tau$. Write $z = \widetilde{z}_\tau + \mathbb{E}[z \mid A_\tau]$ with $\widetilde{z}_\tau := z - \mathbb{E}[z \mid A_\tau]$ and parameterize $f(X) = \theta^\top W + \beta_z \widetilde{z}_\tau$. Defining
$$\mathcal{L}_{A_\tau}(b) := \mathbb{E}\big[(Y - \theta^\top W - b\widetilde{z}_\tau)^2 \mid A_\tau\big].$$

The same expansion applies. Since $A_\tau = A \cap \{z \leq \tau\}$ and $z \perp (g, h, M, B, J)$, conditioning on $A_\tau$ still preserves independence between $\widetilde{z}_\tau$ and $(g, h, M, B, J)$. Using again $\mathbb{E}[gh] = 0$, we obtain $\mathbb{E}[(Y - \theta^\top W)\widetilde{z}_\tau \mid A_\tau] = 0$, and hence
$$\mathcal{L}_{A_\tau}(b) = \mathcal{L}_{A_\tau}(0) + b^2 \mathbb{E}[\widetilde{z}_\tau^2 \mid A_\tau].$$

Since $\mathbb{E}[\widetilde{z}_\tau^2 \mid A_\tau] > 0$, the minimum is attained at $b = 0$, so the optimal predictor in node $A_\tau$ also satisfies $\beta_z = 0$. An identical argument applies to $A_\tau^c$.

Consequently, in all three nodes, the population-optimal predictor can be chosen in the form $f(X) \in \operatorname{span}\{1, g, h, M\}$. For a node $S \in \{A, A_\tau, A_\tau^c\}$, let $\mathcal{V}_S := \operatorname{span}\{1, g, h, M\} \subset L_2(S)$ and denote by $f_S^\star$ the $L_2(S)$-projection of $Y$ onto $\mathcal{V}_S$. The orthogonality condition uniquely characterizes such a projection $\mathbb{E}[(Y - f_S^\star)v \mid S] = 0$ for all $v \in \mathcal{V}_S$.

Let $f^\star := f_A^\star$. For any $v \in \mathcal{V}_{A_\tau}$,
$$\mathbb{E}[(Y - f^\star)v \mid A_\tau] = \mathbb{E}[(1 - B)ghz\,v \mid A_\tau] + \mathbb{E}[(BN - f^\star)v \mid A_\tau].$$

The first term vanishes since $(g, h) \perp z$ under $A_\tau$ and $\mathbb{E}[gh] = 0$. The second term equals $\mathbb{E}[(BN - f^\star)v \mid A]$ because $BN$, $f^\star$, and $v$ depend only on $(g, h, M, B, J)$ and are independent of $z$. This term is zero by the defining orthogonality of $f^\star$ in node $A$. Hence $\mathbb{E}[(Y - f^\star)v \mid A_\tau] = 0$ for all $v \in \mathcal{V}_{A_\tau}$, and by uniqueness of the projection, $f_{A_\tau}^\star = f_A^\star$ as functions. The same argument applies to $A_\tau^c$.

Thus, since $f_A^\star = f_{A_\tau}^\star = f_{A_\tau^c}^\star$ as functions, by the law of total expectation,
$$\mathbb{E}[(Y - f_A^\star(X))^2 \mid A] = \mathbb{P}(A_\tau \mid A)\,\mathbb{E}[(Y - f_A^\star(X))^2 \mid A_\tau] + \mathbb{P}(A_\tau^c \mid A)\,\mathbb{E}[(Y - f_A^\star(X))^2 \mid A_\tau^c]$$
$$= \mathbb{P}(A_\tau \mid A)\,\mathbb{E}[(Y - f_{A_\tau}^\star(X))^2 \mid A_\tau] + \mathbb{P}(A_\tau^c \mid A)\,\mathbb{E}[(Y - f_{A_\tau^c}^\star(X))^2 \mid A_\tau^c].$$

Therefore, splitting on $z$ does not change the population-optimal linear leaf predictor and yields zero one-step gain.

**split on $M$.** We next consider splitting on $m_{j1} = s$, where the index $j$ has not been split on previously. Let $A$ denote the current node event, which depends only on nuisance variables.

Let
$$f_A \in \arg\min_{f \in \mathcal{L}} \mathbb{E}[(Y - f(X))^2 \mid A]$$

denote the population-optimal linear predictor in the parent node $A$.

Since $A$ does not condition on $m_{j2}$ and $m_{j2}$ is independent of $(g, h, z, B, J, \{m_{k,b}\}_{(k,b) \neq (j,2)})$ with $\mathbb{E}[m_{j2}] = 0$, we have
$$\mathbb{E}[m_{j2} \mid A] = 0, \qquad \mathbb{E}[Y\,m_{j2} \mid A] = 0, \qquad \mathbb{E}[m_{j2}\,X_{\text{rest}} \mid A] = 0,$$

where $X_{\text{rest}}$ collects all regressors except $m_{j2}$. Hence the normal equation for the coefficient of $m_{j2}$ implies that $f_A$ has zero coefficient on $m_{j2}$, i.e., $f_A$ does not depend on $m_{j2}$.

Now consider splitting on $m_{j1}$ and the child node $A_s := A \cap \{m_{j1} = s\}$. Define a refined predictor in $A_s$ by

$$f_{A_s}(X) := f_A(X) + a_s\, m_{j2}.$$

Then, expanding the square and using $\mathbb{E}[m_{j2}^2 \mid A_s] = 1$,

$$\mathbb{E}[(Y - f_{A_s}(X))^2 \mid A_s] = \mathbb{E}[(Y - f_A(X))^2 \mid A_s] + a_s^2 - 2a_s\, \mathbb{E}[(Y - f_A(X))m_{j2} \mid A_s].$$

Since $f_A$ does not depend on $m_{j2}$ and $m_{j2}$ is independent of the other regressors given $A_s$, we have $\mathbb{E}[f_A(X)m_{j2} \mid A_s] = 0$, and thus

$$\mathbb{E}[(Y - f_A(X))m_{j2} \mid A_s] = \mathbb{E}[Y m_{j2} \mid A_s].$$

Next, we compute the cross term:

$$\mathbb{E}[Y\, m_{j2} \mid A_s] = \mathbb{E}[(1 - B)T\, m_{j2} \mid A_s] + \mathbb{E}[BN\, m_{j2} \mid A_s]. \tag{4}$$

The first term is zero since $(g, h, z)$ is independent of $(M, B, J)$ and hence of $A_s$, and moreover $T = ghz$ is independent of $m_{j2}$ with $\mathbb{E}[T] = 0$:

$$\mathbb{E}[(1 - B)T\, m_{j2} \mid A_s] = \mathbb{E}[1 - B] \cdot \mathbb{E}[T] \cdot \mathbb{E}[m_{j2} \mid A_s] = 0.$$

For the second term, using $B \perp (M, J)$ and $A_s \in \sigma(M)$,

$$\mathbb{E}[BN\, m_{j2} \mid A_s] = \mathbb{E}[B] \cdot \mathbb{E}[N\, m_{j2} \mid A_s] = \varepsilon \cdot \mathbb{E}[N\, m_{j2} \mid A_s].$$

Finally, since $N = m_{J,1}m_{J,2}$ and $J$ is uniform on $\{1, \dots, U\}$,

$$\mathbb{E}[N\, m_{j2} \mid A_s] = \frac{1}{U} \sum_{k=1}^{U} \mathbb{E}[m_{k,1}m_{k,2}m_{j2} \mid A_s]$$

$$= \frac{1}{U}\mathbb{E}[m_{j,1}m_{j,2}^2 \mid A_s] + \frac{1}{U} \sum_{k \neq j} \mathbb{E}[m_{k,1}m_{k,2}m_{j2} \mid A_s]. \tag{5}$$

For $k \neq j$, $(m_{k,1}, m_{k,2})$ is independent of $(m_{j1}, m_{j2})$, while $m_{j2} \notin A_s$ hence the $\mathbb{E}[m_{j2} \mid A_s] = 0$ and thus each summand is zero. For $k = j$, on $A_s$ we have $m_{j1} = s$ and $m_{j2}^2 = 1$, so

$$\mathbb{E}[m_{j,1}m_{j,2}^2 \mid A_s] = \mathbb{E}[m_{j1} \mid A_s] = s.$$

Therefore,

$$\mathbb{E}[N\, m_{j2} \mid A_s] = \frac{s}{U}, \qquad \text{and hence} \qquad \mathbb{E}[Y\, m_{j2} \mid A_s] = \varepsilon \cdot \frac{s}{U}.$$

Therefore,

$$\mathbb{E}[(Y - f_{A_s}(X))^2 \mid A_s] = \mathbb{E}[(Y - f_A(X))^2 \mid A_s] + a_s^2 - 2a_s \cdot \frac{\varepsilon s}{U}.$$

Minimizing over $a_s \in \mathbb{R}$ yields $a_s^* = \varepsilon s/U$ and

$$\inf_{a_s} \mathbb{E}[(Y - f_A(X) - a_s m_{j2})^2 \mid A_s] = \mathbb{E}[(Y - f_A(X))^2 \mid A_s] - \frac{\varepsilon^2}{U^2}.$$

Since $\inf_f \mathbb{E}[(Y - f(X))^2 \mid A_s] \leq \inf_{a_s} \mathbb{E}[(Y - f_A(X) - a_s m_{j2})^2 \mid A_s]$, we obtain

$$\inf_f \mathbb{E}[(Y - f(X))^2 \mid A_s] \leq \mathbb{E}[(Y - f_A(X))^2 \mid A_s] - \frac{\varepsilon^2}{U^2}.$$

Averaging over $s \in \{\pm 1\}$ and using the law of total expectation,

$$\sum_{s=\pm 1} \mathbb{P}(m_{j1} = s \mid A) \inf_f \mathbb{E}[(Y - f(X))^2 \mid A_s] \leq \mathbb{E}[(Y - f_A(X))^2 \mid A] - \frac{\varepsilon^2}{U^2}$$

Hence, splitting on $m_{j1}$ yields a strictly positive one-step gain of at least $\varepsilon^2/U^2$. $\qquad \square$

**Theorem B.6** (Arbitrary MSE Gap between CLARITree and Greedy, recalling Theorem 5.4). *Fix any depth $d \geq 2$ and any $\varepsilon \in (0, 1/2)$. Let $(Y, X)$ with $X = (g, h, M, z)$ follow the DGP definition, where $M$ are referred to as* nuisance variables. *Then*

$$\text{MSE}_{\text{Greedy}} \geq 1 - \varepsilon, \qquad \text{MSE}_{\text{CLARITree}} \leq 2\varepsilon,$$

*and hence*

$$\frac{\text{MSE}_{\text{Greedy}}}{\text{MSE}_{\text{CLARITree}}} \geq \frac{1 - \varepsilon}{2\varepsilon} > \frac{1}{4\varepsilon},$$

*which can be made arbitrarily large as $\varepsilon \to 0$.*

*Proof.* **Greedy CholeskyTree never splits on $g$, $h$ or $z$.** By the Lemma B.5, at any node event $A \in \sigma(M)$ the population one-step gain from splitting on $g$, $h$ or $z$ is zero, while there exists at least on some $m_{j,1}$ or $m_{j,2}$ not previously split with strictly positive one-step gain. Since $U > d$, along the first $d$ splits, there always remains an unresolved nuisance pair, so the greedy rule selects nuisance variables at every split. Consequently, every leaf of the depth-$d$ Greedy CholeskyTree corresponds to some event $A \in \sigma(M)$ and does not condition on $g$ or $h$.

**Lower bound for Greedy MSE.** Fix any such leaf event $A \in \sigma(M)$. Because $(g, h, z)$ is independent of $A$, the conditional distribution of $(g, h, z)$ given $A$ is unchanged. Let $\widehat{f}_A$ denote the population OLS predictor in that leaf, i.e.,

$$\widehat{f}_A \in \arg\min_f \mathbb{E}[(Y - f(X))^2 \mid A],$$

where $\mathcal{L}$ is the linear leaf model class. On the main component $B = 0$, we have $Y = T = ghz$. $T$ is $L_2(A)$-orthogonal to the linear span of the available regressors in the leaf, hence for any linear $f$,

$$\mathbb{E}[(T - f(X))^2 \mid A] \geq \mathbb{E}[T^2 \mid A] = 1.$$

In particular, this holds for $f = \widehat{f}_A$. Therefore,

$$\mathbb{E}[(Y - \widehat{f}_A(X))^2 \mid A] \geq \mathbb{P}(B = 0) \cdot \mathbb{E}[(T - \widehat{f}_A(X))^2 \mid A, B = 0] \geq (1 - \varepsilon) \cdot 1.$$

Averaging over all leaves yields

$$\text{MSE}_{\text{Greedy}} = \mathbb{E}\big[(Y - \widehat{f}_{\text{Greedy}}(X))^2\big] \geq 1 - \varepsilon.$$

**Upper bound for CLARITree MSE.**

To make the objective notation precise, for any event $A$ with $\mathbb{P}(A) > 0$, let $D_A$ denote the data distribution restricted to $A$. We define the weighted leaf risk by

$$\text{LeafRisk}(D_A) := \mathbb{P}(A) \inf_{\ell \in \mathcal{L}} \mathbb{E}\left[(Y - \ell(X))^2 \mid A\right],$$

where $\mathcal{L}$ is the class of linear leaf predictors. Thus $\text{LeafRisk}(D_A)$ already includes the probability mass of node $A$. Under this convention, the objective of a tree is the sum of LeafRisk over its leaves.

We define the objective of the fixed depth-two candidate that first splits on $g$ and then splits on $h$ in both children as

$$\Phi_{g,h}(D) := \sum_{s,t \in \{-1,1\}} \text{LeafRisk}(D_{g=s,h=t}).$$

We first record a simple property of the Greedy CholeskyTree subroutine. For any node distribution $D_A$, any remaining depth $t \geq 1$, and any feasible one-step split $q$ producing children $D_{A_L(q)}$ and $D_{A_R(q)}$, we have

$$G_t(D_A) \leq \text{LeafRisk}(D_{A_L(q)}) + \text{LeafRisk}(D_{A_R(q)}).$$

Indeed, suppose Greedy CholeskyTree selects the one-step split $(f^\star, \tau^\star)$, producing children $D_{A_L^\star}$ and $D_{A_R^\star}$. By the pruning rule in Algorithm 4, recursive growth cannot return an objective larger than the immediate two-leaf objective of the selected split. Hence

$$G_t(D_A) \leq \text{LeafRisk}(D_{A_L^\star}) + \text{LeafRisk}(D_{A_R^\star}).$$

Moreover, since $(f^\star, \tau^\star)$ is chosen to minimize the one-step child-leaf objective, for any feasible split $q$,

$$\text{LeafRisk}(D_{A_L^\star}) + \text{LeafRisk}(D_{A_R^\star}) \leq \text{LeafRisk}(D_{A_L(q)}) + \text{LeafRisk}(D_{A_R(q)}).$$

Combining the two inequalities gives the claim.

Now following the notation in Theorem B.4, let $D_L^{\star\star}$ and $D_R^{\star\star}$ denote the two children of the root split selected by CLARITree. Since CLARITree chooses the root split minimizing the Greedy-completed objective, while splitting on $g$ is one feasible root candidate, we have

$$
\begin{aligned}
C_d(D) &= C_{d-1}(D_L^{\star\star}) + C_{d-1}(D_R^{\star\star}) \\
&\leq G_{d-1}(D_L^{\star\star}) + G_{d-1}(D_R^{\star\star}) \\
&\leq G_{d-1}(D_{g=-1}) + G_{d-1}(D_{g=1}) \\
&\leq \sum_{s \in \{-1,1\}} [\text{LeafRisk}(D_{g=s,h=-1}) + \text{LeafRisk}(D_{g=s,h=1})] \\
&= \Phi_{g,h}(D).
\end{aligned}
$$

The first inequality follows from Theorem B.4. The second inequality follows from the definition of CLARITree: the split selected by CLARITree minimizes the objective under Greedy CholeskyTree completion, whereas the root split on $g$ is only one feasible candidate. The third inequality applies the Greedy property above to each child node $D_{g=s}$, using the feasible split on $h$.

It remains to bound $\Phi_{g,h}(D)$. Conditional on $g = s$ and $h = t$, the main component satisfies

$$T = ghz = stz = cz,$$

where $c = st \in \{-1, 1\}$. Hence the linear predictor $\widehat{f}(X) = cz$ represents the main component exactly on the event $B = 0$. Therefore, for every $s, t \in \{-1, 1\}$,

$$
\begin{aligned}
\inf_{\ell \in \mathcal{L}} \mathbb{E}\left[(Y - \ell(X))^2 \mid g = s, h = t\right] &\leq \mathbb{E}\left[(Y - cz)^2 \mid g = s, h = t\right] \\
&= \mathbb{E}\left[B(N - cz)^2 \mid g = s, h = t\right] \\
&= \varepsilon\, \mathbb{E}\left[(N - cz)^2 \mid g = s, h = t\right] \\
&= 2\varepsilon,
\end{aligned}
$$

where $N = m_{J1}m_{J2}$, $B^2 = B$, $N^2 = 1$, $z \sim N(0, 1)$, and $N$ is independent of $z$. Thus

$$\text{LeafRisk}(D_{g=s,h=t}) \leq 2\varepsilon\, \mathbb{P}(g = s, h = t).$$

Summing over the four leaves gives

$$\Phi_{g,h}(D) \leq 2\varepsilon \sum_{s,t \in \{-1,1\}} \mathbb{P}(g = s, h = t) = 2\varepsilon.$$

Consequently,

$$C_d(D) \leq \Phi_{g,h}(D) \leq 2\varepsilon.$$

Combining the bounds and note that when $\varepsilon \in (0, 1/2)$,

$$\frac{1 - \varepsilon}{2\varepsilon} > \frac{1}{4\varepsilon},$$

completes the proof. $\square$

To further validate the theoretical gap in a simple setting, we additionally conducted a synthetic experiment with $U = 8$, tree depth $d = 4$, and sample size $n = 1000$. We evaluated a sequence of $\varepsilon$ values and compared the empirical performance gap between the greedy regression tree and **CLARITree**. The corresponding results are shown in Figure 4, where the empirical behavior closely matches the theoretical prediction as $\varepsilon \to 0$.

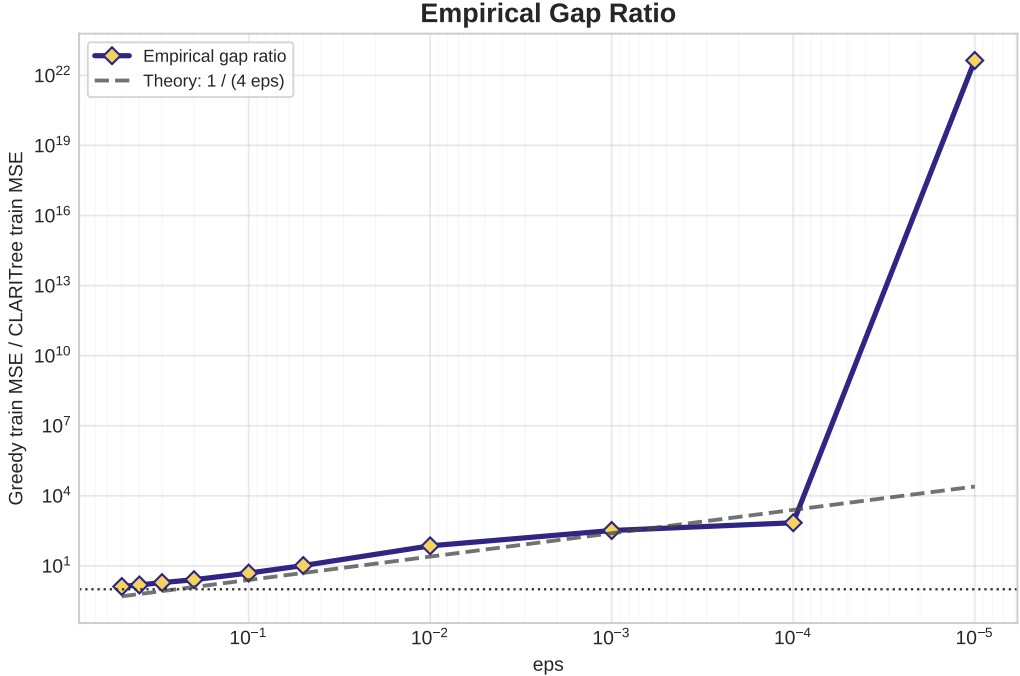

*Figure 4.* Empirical performance gap under the synthetic DGP for varying $\varepsilon$ with $U = 8$, depth $= 4$, and $n = 1000$. As $\varepsilon$ decreases, the gap between greedy regression trees and **CLARITree** becomes increasingly pronounced.

## C. Full Algorithms

This appendix presents the complete pseudocode for the proposed methods and supporting routines used throughout the implementation.

We first describe the preprocessing and initialization procedure in Algorithm 3, followed by the construction algorithms for `CholeskyTree` and `CLARITree`.

We then provide the streamed split enumeration routine together with auxiliary procedures for threshold construction, sorted-index maintenance, ridge sufficient statistics, and Cholesky-based linear algebra operations.

**Treatment of categorical features.** Categorical variables may be handled in two different ways:

(a) One-hot encoded categorical variables are treated as binary splitting features;

(b) Ordinal or discretized categorical variables may instead be represented as discrete-valued continuous variables.

In the current implementation, one-hot encoded categorical variables are used only for split decisions and are excluded from the leaf regression models, while ordinal or discretized categorical variables are treated identically to continuous features.

---

**Algorithm 3** `FitTree`: preprocessing and initialization for `CholeskyTree` or `CLARITree`

---

**Require:** Data matrix $\mathbf{X} \in \mathbb{R}^{n \times k}$; response $\boldsymbol{y} \in \mathbb{R}^n$; categorical feature set $\mathcal{G}$; ridge parameter $\kappa$; leaf penalty $\lambda$; maximum depth $d$; number of thresholds $T$; threshold strategy `strategy`; minimum leaf size parameter $m_{\min}$; method `Method` $\in \{$`CholeskyTree`, `CLARITree`$\}$

1: Ensure an intercept column is present in $\mathbf{X}$ as column $0$

2: Detect binary features $\mathcal{B}$ and continuous features $\mathcal{C}$

3: $m_{\min}^{resolved} \leftarrow \begin{cases} m_{\min}, & m_{\min} > 0, \\ \max\{1, 5|\mathcal{C}|\}, & m_{\min} = 0 \end{cases}$

4: $\bar{y} \leftarrow n^{-1} \sum_{i=1}^n y_i, \quad \widetilde{\boldsymbol{y}} \leftarrow \boldsymbol{y} - \bar{y}\mathbf{1}$           ▷ Center response unless it is already centered.

5: $\lambda_s \leftarrow \lambda \sum_{i=1}^n (y_i - \bar{y})^2, \quad \kappa_s \leftarrow n\kappa$           ▷ Use the scaled parameters from the implementation.

6: Standardize each continuous column $c \in \mathcal{C}$ using training mean and standard deviation

7: Define the ridge-regression row
$$\boldsymbol{\phi}_i^\top = \left(1, \widetilde{x}_{i,c_1}, \ldots, \widetilde{x}_{i,c_{|\mathcal{C}|}}\right) \in \mathbb{R}^{1+|\mathcal{C}|}$$

          ▷ Leaves regress only on intercept and continuous features.

8: $\mathbf{A}_\mathcal{I} \leftarrow \sum_{i=1}^n \boldsymbol{\phi}_i \boldsymbol{\phi}_i^\top + \kappa_s I$

9: Set the intercept ridge entry to a negligible value           ▷ The intercept is effectively unpenalized.

10: $\mathbf{L}_\mathcal{I} \leftarrow$ `chol`$(\mathbf{A}_\mathcal{I})$

11: $\boldsymbol{b}_\mathcal{I} \leftarrow \sum_{i=1}^n \boldsymbol{\phi}_i \widetilde{y}_i, \quad \|\boldsymbol{y}_\mathcal{I}\|^2 \leftarrow \sum_{i=1}^n \widetilde{y}_i^2$

12: $\text{OBJ}_{init} \leftarrow$ `LossFromCholesky`$(\mathbf{L}_\mathcal{I}, \boldsymbol{b}_\mathcal{I}, \|\boldsymbol{y}_\mathcal{I}\|^2) + \lambda_s$

13: Build global sorted index lists $\Pi = \{\pi_f(\mathcal{I})\}_{f=1}^k$           ▷ Each list is sorted by the original feature value $x_{if}$.

14: $\mathcal{P} \leftarrow$ `BuildThresholdPool`$(\mathbf{X}, \Pi, \mathcal{B}, \mathcal{C}, T, $`strategy`$)$

15: **if** `Method` $=$ `CholeskyTree` **then**

16:      $(T, \text{OBJ}) \leftarrow$ `CholeskyTree`$(\mathbf{L}_\mathcal{I}, \boldsymbol{b}_\mathcal{I}, D_\mathcal{I}, \Pi, \mathcal{P}, \mathcal{B}, \mathcal{C}, \kappa_s, \lambda_s, m_{\min}^{resolved}, d, \|\boldsymbol{y}_\mathcal{I}\|^2, \text{OBJ}_{init})$

17: **else**

18:      $(T, \text{OBJ}) \leftarrow$ `CLARITree`$(\mathbf{L}_\mathcal{I}, \boldsymbol{b}_\mathcal{I}, D_\mathcal{I}, \Pi, \mathcal{P}, \mathcal{B}, \mathcal{C}, \kappa_s, \lambda_s, m_{\min}^{resolved}, d, \|\boldsymbol{y}_\mathcal{I}\|^2, \text{OBJ}_{init})$

19: **end if**

20: Fit final ridge coefficients in every leaf of $T$      ▷ Convert coefficients back to the original feature scale for prediction.

21: **return** $T$, OBJ

---

---

**Algorithm 4** `CholeskyTree`: greedy linear regression tree with streamed split enumeration

---

**Require:** Node Cholesky factor $\mathbf{L}_{\mathcal{I}}$; node vector $\boldsymbol{b}_{\mathcal{I}}$; node data $D_{\mathcal{I}}$; node-wise sorted lists $\Pi_{\mathcal{I}}$; threshold pool $\mathcal{P}$; binary features $\mathcal{B}$; continuous features $\mathcal{C}$; scaled ridge parameter $\kappa_s$; scaled leaf penalty $\lambda_s$; resolved minimum leaf size $m_{\min}^{resolved}$; depth budget $d$; node response norm $\|\boldsymbol{y}_{\mathcal{I}}\|^2$; initial leaf objective $\text{OBJ}_{init}$

1: $n_{\mathcal{I}} \leftarrow |\mathcal{I}|$
2: **if** $d = 0$ **or** $\text{OBJ}_{init} \le 2\lambda_s$ **or** $n_{\mathcal{I}} < 2m_{\min}^{resolved}$ **then**
3:     $\boldsymbol{\beta} \leftarrow \texttt{SolveRidge}(\mathbf{L}_{\mathcal{I}}, \boldsymbol{b}_{\mathcal{I}})$
4:     **return** $\texttt{Leaf}(\boldsymbol{\beta}, \text{OBJ}_{init})$, $\text{OBJ}_{init}$
5: **end if**
6: $\text{OBJ}_{split}^* \leftarrow \infty, \quad f^*, \tau^* \leftarrow \texttt{None}$                   ▷ Avoid early stopping based on one-layer gain.
7: $\mathcal{U} \leftarrow \texttt{EnumerateSplits}(D_{\mathcal{I}}, \Pi_{\mathcal{I}}, \mathcal{P}, \mathcal{B}, \mathcal{C}, \kappa_s, \lambda_s, m_{\min}^{resolved}, \mathbf{L}_{\mathcal{I}}, \boldsymbol{b}_{\mathcal{I}}, \|\boldsymbol{y}_{\mathcal{I}}\|^2)$
8: **for** each tuple yielded by $\mathcal{U}$ **do**
9:     $(f, \tau, \mathcal{I}_L, \text{OBJLEAF}_L, \text{OBJLEAF}_R, \mathbf{L}_L, \mathbf{L}_R, \boldsymbol{b}_L, \boldsymbol{b}_R, \|\boldsymbol{y}_{\mathcal{I}_L}\|^2, \|\boldsymbol{y}_{\mathcal{I}_R}\|^2)$         ▷ Variable list from $\mathcal{U}$.
10:     $\text{OBJ}_{split} \leftarrow \text{OBJLEAF}_L + \text{OBJLEAF}_R$
11:     **if** $\text{OBJ}_{split} < \text{OBJ}_{split}^*$ **then**
12:         $\text{OBJ}_{split}^* \leftarrow \text{OBJ}_{split}, \quad f^*, \tau^* \leftarrow f, \tau$
13:         Store $(\mathcal{I}_L, \mathbf{L}_L, \mathbf{L}_R, \boldsymbol{b}_L, \boldsymbol{b}_R, \text{OBJLEAF}_L, \text{OBJLEAF}_R, \|\boldsymbol{y}_{\mathcal{I}_L}\|^2, \|\boldsymbol{y}_{\mathcal{I}_R}\|^2)$ as best
14:     **end if**
15: **end for**
16: **if** $f^* = \texttt{None}$ **then**
17:     $\boldsymbol{\beta} \leftarrow \texttt{SolveRidge}(\mathbf{L}_{\mathcal{I}}, \boldsymbol{b}_{\mathcal{I}})$
18:     **return** $\texttt{Leaf}(\boldsymbol{\beta}, \text{OBJ}_{init})$, $\text{OBJ}_{init}$
19: **end if**
20: **if** $d = 1$ **then**
21:     **if** $\text{OBJ}_{split}^* \ge \text{OBJ}_{init}$ **then**
22:         $\boldsymbol{\beta} \leftarrow \texttt{SolveRidge}(\mathbf{L}_{\mathcal{I}}, \boldsymbol{b}_{\mathcal{I}})$
23:         **return** $\texttt{Leaf}(\boldsymbol{\beta}, \text{OBJ}_{init})$, $\text{OBJ}_{init}$     ▷ At the last level, prune if the best split does not improve the leaf.
24:     **else**
25:         Build leaf children with objectives $\text{OBJLEAF}_L^{best}$ and $\text{OBJLEAF}_R^{best}$
26:         **return** $\texttt{Node}(f^*, \tau^*, T_L, T_R)$, $\text{OBJ}_{split}^*$
27:     **end if**
28: **end if**
29: $\Pi_{\mathcal{I}_L^{best}}, \Pi_{\mathcal{I}_R^{best}} \leftarrow \texttt{SplitSortedIndices}(\Pi_{\mathcal{I}}, \mathcal{I}_L^{best})$
30: $D_{\mathcal{I}_L^{best}}, D_{\mathcal{I}_R^{best}} \leftarrow \texttt{RestrictData}(D_{\mathcal{I}}, \mathcal{I}_L^{best})$                   ▷ View via indices.
31: $(T_L, \text{OBJ}_L) \quad\quad\quad \leftarrow \quad\quad\quad \texttt{CholeskyTree}(\mathbf{L}_L^{best}, \boldsymbol{b}_L^{best}, D_{\mathcal{I}_L^{best}}, \Pi_{\mathcal{I}_L^{best}}, \mathcal{P}, \mathcal{B}, \mathcal{C}, \kappa_s, \lambda_s, m_{\min}^{resolved}, d - 1, \|\boldsymbol{y}_{\mathcal{I}_L^{best}}\|^2, \text{OBJLEAF}_L^{best})$
32: $(T_R, \text{OBJ}_R) \quad\quad\quad \leftarrow \quad\quad\quad \texttt{CholeskyTree}(\mathbf{L}_R^{best}, \boldsymbol{b}_R^{best}, D_{\mathcal{I}_R^{best}}, \Pi_{\mathcal{I}_R^{best}}, \mathcal{P}, \mathcal{B}, \mathcal{C}, \kappa_s, \lambda_s, m_{\min}^{resolved}, d - 1, \|\boldsymbol{y}_{\mathcal{I}_R^{best}}\|^2, \text{OBJLEAF}_R^{best})$
33: $\text{OBJ}_{subtree} \leftarrow \text{OBJ}_L + \text{OBJ}_R$
34: **if** $\text{OBJ}_{subtree} < \text{OBJ}_{init}$ **then**
35:     **return** $\texttt{Node}(f^*, \tau^*, T_L, T_R)$, $\text{OBJ}_{subtree}$
36: **else**
37:     $\boldsymbol{\beta} \leftarrow \texttt{SolveRidge}(\mathbf{L}_{\mathcal{I}}, \boldsymbol{b}_{\mathcal{I}})$
38:     **return** $\texttt{Leaf}(\boldsymbol{\beta}, \text{OBJ}_{init})$, $\text{OBJ}_{init}$     ▷ Prune if recursive growth does not improve the leaf.
39: **end if**

---

---

**Algorithm 5** `CLARITree`: Cholesky and lookahead-accelerated linear regression tree

---

**Require:** Node Cholesky factor $\mathbf{L}_{\mathcal{I}}$; node vector $\boldsymbol{b}_{\mathcal{I}}$; node data $D_{\mathcal{I}}$; node-wise sorted lists $\Pi_{\mathcal{I}}$; threshold pool $\mathcal{P}$; binary features $\mathcal{B}$; continuous features $\mathcal{C}$; scaled ridge parameter $\kappa_s$; scaled leaf penalty $\lambda_s$; resolved minimum leaf size $m_{\min}^{resolved}$; depth budget $d$; node response norm $\|\boldsymbol{y}_{\mathcal{I}}\|^2$; initial leaf objective $\text{OBJ}_{init}$

1: $n_{\mathcal{I}} \leftarrow |\mathcal{I}|$
2: **if** $d = 0$ **or** $\text{OBJ}_{init} \leq 2\lambda_s$ **or** $n_{\mathcal{I}} < 2m_{\min}^{resolved}$ **then**
3:     $\boldsymbol{\beta} \leftarrow \text{SolveRidge}(\mathbf{L}_{\mathcal{I}}, \boldsymbol{b}_{\mathcal{I}})$
4:     **return** $\text{Leaf}(\boldsymbol{\beta}, \text{OBJ}_{init})$, $\text{OBJ}_{init}$
5: **end if**
6: $\text{OBJ}_{look}^* \leftarrow \infty, \quad f^*, \tau^* \leftarrow \text{None}$          ▷ Select the split using greedy completions.
7: $\mathcal{U} \leftarrow \text{EnumerateSplits}(D_{\mathcal{I}}, \Pi_{\mathcal{I}}, \mathcal{P}, \mathcal{B}, \mathcal{C}, \kappa_s, \lambda_s, m_{\min}^{resolved}, \mathbf{L}_{\mathcal{I}}, \boldsymbol{b}_{\mathcal{I}}, \|\boldsymbol{y}_{\mathcal{I}}\|^2)$
8: **for** each tuple yielded by $\mathcal{U}$ **do**
9:     $(f, \tau, \mathcal{I}_L, \text{OBJLEAF}_L, \text{OBJLEAF}_R, \mathbf{L}_L, \mathbf{L}_R, \boldsymbol{b}_L, \boldsymbol{b}_R, \|\boldsymbol{y}_{\mathcal{I}_L}\|^2, \|\boldsymbol{y}_{\mathcal{I}_R}\|^2)$      ▷ One candidate split.
10:     **if** $d = 1$ **then**
11:         $\text{OBJ}_{cand} \leftarrow \text{OBJLEAF}_L + \text{OBJLEAF}_R$      ▷ No remaining depth for lookahead.
12:     **else**      ▷ Greedy CholeskyTree completion of the remaining subtree.
13:         $\Pi_{\mathcal{I}_L}, \Pi_{\mathcal{I}_R} \leftarrow \text{SplitSortedIndices}(\Pi_{\mathcal{I}}, \mathcal{I}_L)$
14:         $D_{\mathcal{I}_L}, D_{\mathcal{I}_R} \leftarrow \text{RestrictData}(D_{\mathcal{I}}, \mathcal{I}_L)$      ▷ Create temporary child views for greedy completion.
15:         $(\widetilde{T}_L, \widetilde{\text{OBJ}}_L) \leftarrow \text{CholeskyTree}(\mathbf{L}_L, \boldsymbol{b}_L, D_{\mathcal{I}_L}, \Pi_{\mathcal{I}_L}, \mathcal{P}, \mathcal{B}, \mathcal{C}, \kappa_s, \lambda_s, m_{\min}^{resolved}, d-1, \|\boldsymbol{y}_{\mathcal{I}_L}\|^2, \text{OBJLEAF}_L)$
16:         $(\widetilde{T}_R, \widetilde{\text{OBJ}}_R) \leftarrow \text{CholeskyTree}(\mathbf{L}_R, \boldsymbol{b}_R, D_{\mathcal{I}_R}, \Pi_{\mathcal{I}_R}, \mathcal{P}, \mathcal{B}, \mathcal{C}, \kappa_s, \lambda_s, m_{\min}^{resolved}, d-1, \|\boldsymbol{y}_{\mathcal{I}_R}\|^2, \text{OBJLEAF}_R)$
17:         $\text{OBJ}_{cand} \leftarrow \widetilde{\text{OBJ}}_L + \widetilde{\text{OBJ}}_R$
18:     **end if**
19:     **if** $\text{OBJ}_{cand} < \text{OBJ}_{look}^*$ **then**
20:         $\text{OBJ}_{look}^* \leftarrow \text{OBJ}_{cand}, \quad f^*, \tau^* \leftarrow f, \tau$
21:         Store $(\mathcal{I}_L, \mathbf{L}_L, \mathbf{L}_R, \boldsymbol{b}_L, \boldsymbol{b}_R, \text{OBJLEAF}_L, \text{OBJLEAF}_R, \|\boldsymbol{y}_{\mathcal{I}_L}\|^2, \|\boldsymbol{y}_{\mathcal{I}_R}\|^2)$ as best
22:     **end if**
23: **end for**
24: **if** $f^* = \text{None}$ **then**
25:     $\boldsymbol{\beta} \leftarrow \text{SolveRidge}(\mathbf{L}_{\mathcal{I}}, \boldsymbol{b}_{\mathcal{I}})$
26:     **return** $\text{Leaf}(\boldsymbol{\beta}, \text{OBJ}_{init})$, $\text{OBJ}_{init}$
27: **end if**
28: **if** $d = 1$ **then**
29:     **if** $\text{OBJ}_{look}^* \geq \text{OBJ}_{init}$ **then**
30:         $\boldsymbol{\beta} \leftarrow \text{SolveRidge}(\mathbf{L}_{\mathcal{I}}, \boldsymbol{b}_{\mathcal{I}})$
31:         **return** $\text{Leaf}(\boldsymbol{\beta}, \text{OBJ}_{init})$, $\text{OBJ}_{init}$
32:     **else**
33:         Build leaf children with objectives $\text{OBJLEAF}_L^{best}$ and $\text{OBJLEAF}_R^{best}$
34:         **return** $\text{Node}(f^*, \tau^*, T_L, T_R)$, $\text{OBJ}_{look}^*$
35:     **end if**
36: **end if**
37: $\Pi_{\mathcal{I}_L^{best}}, \Pi_{\mathcal{I}_R^{best}} \leftarrow \text{SplitSortedIndices}(\Pi_{\mathcal{I}}, \mathcal{I}_L^{best})$
38: $D_{\mathcal{I}_L^{best}}, D_{\mathcal{I}_R^{best}} \leftarrow \text{RestrictData}(D_{\mathcal{I}}, \mathcal{I}_L^{best})$
39: $(T_L, \text{OBJ}_L) \leftarrow \text{CLARITree}(\mathbf{L}_L^{best}, \boldsymbol{b}_L^{best}, D_{\mathcal{I}_L^{best}}, \Pi_{\mathcal{I}_L^{best}}, \mathcal{P}, \mathcal{B}, \mathcal{C}, \kappa_s, \lambda_s, m_{\min}^{resolved}, d-1, \|\boldsymbol{y}_{\mathcal{I}_L^{best}}\|^2, \text{OBJLEAF}_L^{best})$
40: $(T_R, \text{OBJ}_R) \leftarrow \text{CLARITree}(\mathbf{L}_R^{best}, \boldsymbol{b}_R^{best}, D_{\mathcal{I}_R^{best}}, \Pi_{\mathcal{I}_R^{best}}, \mathcal{P}, \mathcal{B}, \mathcal{C}, \kappa_s, \lambda_s, m_{\min}^{resolved}, d-1, \|\boldsymbol{y}_{\mathcal{I}_R^{best}}\|^2, \text{OBJLEAF}_R^{best})$
     ▷ Replace greedy lookahead completions with CLARITree recursion.
41: $\text{OBJ}_{final} \leftarrow \text{OBJ}_L + \text{OBJ}_R$
42: **if** $\text{OBJ}_{final} < \text{OBJ}_{init}$ **then**
43:     **return** $\text{Node}(f^*, \tau^*, T_L, T_R)$, $\text{OBJ}_{final}$
44: **else**
45:     $\boldsymbol{\beta} \leftarrow \text{SolveRidge}(\mathbf{L}_{\mathcal{I}}, \boldsymbol{b}_{\mathcal{I}})$
46:     **return** $\text{Leaf}(\boldsymbol{\beta}, \text{OBJ}_{init})$, $\text{OBJ}_{init}$      ▷ Prune if the final CLARITree recursion does not improve the leaf.
47: **end if**

---

---

**Algorithm 6** `EnumerateSplits`: streamed split enumeration using threshold pools and Cholesky updates

---

**Require:** Node data $D_\mathcal{I}$; node-wise sorted lists $\Pi_\mathcal{I}$; threshold pool $\mathcal{P} = \{\mathcal{P}_f\}_f$; binary features $\mathcal{B}$; continuous features $\mathcal{C}$; scaled ridge parameter $\kappa_s$; scaled leaf penalty $\lambda_s$; resolved minimum leaf size $m_{\min}^{resolved}$; parent Cholesky factor $\mathbf{L}_\mathcal{I}$; parent vector $\boldsymbol{b}_\mathcal{I}$; parent response norm $\|\boldsymbol{y}_\mathcal{I}\|^2$

1: **for** $f = 1$ **to** $k$ **do**
2:     **if** $f \in \mathcal{B}$ **then**
3:         $\mathcal{I}_L \leftarrow \{i \in \mathcal{I} : x_{if} \leq 0.5\}, \quad \mathcal{I}_R \leftarrow \mathcal{I} \setminus \mathcal{I}_L$
4:         **if** $|\mathcal{I}_L| < m_{\min}^{resolved}$ **or** $|\mathcal{I}_R| < m_{\min}^{resolved}$ **then**
5:             **continue**
6:         **end if**
7:         $(\mathbf{L}_L, \boldsymbol{b}_L, \|\boldsymbol{y}_{\mathcal{I}_L}\|^2) \leftarrow \texttt{RecomputeStatsFromRows}(\mathcal{I}_L, \kappa_s)$
8:         $(\mathbf{L}_R, \boldsymbol{b}_R, \|\boldsymbol{y}_{\mathcal{I}_R}\|^2) \leftarrow \texttt{RecomputeStatsFromRows}(\mathcal{I}_R, \kappa_s)$
9:         $\text{OBJLEAF}_L \leftarrow \texttt{LossFromCholesky}(\mathbf{L}_L, \boldsymbol{b}_L, \|\boldsymbol{y}_{\mathcal{I}_L}\|^2) + \lambda_s$
10:       $\text{OBJLEAF}_R \leftarrow \texttt{LossFromCholesky}(\mathbf{L}_R, \boldsymbol{b}_R, \|\boldsymbol{y}_{\mathcal{I}_R}\|^2) + \lambda_s$
11:       **yield** $(f, 0.5, \mathcal{I}_L, \text{OBJLEAF}_L, \text{OBJLEAF}_R, \mathbf{L}_L, \mathbf{L}_R, \boldsymbol{b}_L, \boldsymbol{b}_R, \|\boldsymbol{y}_{\mathcal{I}_L}\|^2, \|\boldsymbol{y}_{\mathcal{I}_R}\|^2)$   ▷ Binary features use the fixed split threshold 0.5.
12:       **continue**
13:     **end if**
14:     **if** $\mathcal{P}_f = \emptyset$ **then**
15:         **continue**
16:     **end if**
17:     $\mathbf{A}_L \leftarrow \kappa_s I$ and set the intercept ridge entry to a negligible value
18:     $\mathbf{L}_L \leftarrow \texttt{chol}(\mathbf{A}_L), \quad \boldsymbol{b}_L \leftarrow \boldsymbol{0}, \quad \|\boldsymbol{y}_{\mathcal{I}_L}\|^2 \leftarrow 0, \quad \mathcal{I}_L \leftarrow \emptyset$
19:     $\mathbf{L}_R \leftarrow \mathbf{L}_\mathcal{I}, \quad \boldsymbol{b}_R \leftarrow \boldsymbol{b}_\mathcal{I}, \quad \|\boldsymbol{y}_{\mathcal{I}_R}\|^2 \leftarrow \|\boldsymbol{y}_\mathcal{I}\|^2$
20:     $q \leftarrow 1$     ▷ Pointer into the sorted threshold pool $\mathcal{P}_f$.
21:     **for** $r = 1$ **to** $|\pi_f(\mathcal{I})|$ **do**
22:         $i \leftarrow \pi_f(\mathcal{I})[r]$     ▷ Move sample $i$ from the right child to the left child.
23:         $\mathcal{I}_L \leftarrow \mathcal{I}_L \cup \{i\}$
24:         $\boldsymbol{b}_L \leftarrow \boldsymbol{b}_L + \boldsymbol{\phi}_i \widetilde{y}_i, \quad \boldsymbol{b}_R \leftarrow \boldsymbol{b}_R - \boldsymbol{\phi}_i \widetilde{y}_i$
25:         $\|\boldsymbol{y}_{\mathcal{I}_L}\|^2 \leftarrow \|\boldsymbol{y}_{\mathcal{I}_L}\|^2 + \widetilde{y}_i^2, \quad \|\boldsymbol{y}_{\mathcal{I}_R}\|^2 \leftarrow \|\boldsymbol{y}_{\mathcal{I}_R}\|^2 - \widetilde{y}_i^2$
26:         $\mathbf{L}_L \leftarrow \texttt{cholupdate}(\mathbf{L}_L, \boldsymbol{\phi}_i, +1), \quad \mathbf{L}_R \leftarrow \texttt{cholupdate}(\mathbf{L}_R, \boldsymbol{\phi}_i, -1)$
27:         **if** $r = |\pi_f(\mathcal{I})|$ **then**
28:             **continue**     ▷ The last prefix cannot define a valid split.
29:         **end if**
30:         $v_{cur} \leftarrow x_{if}, \quad v_{next} \leftarrow x_{\pi_f(\mathcal{I})[r+1], f}$
31:         **if** $v_{cur} = v_{next}$ **then**
32:             **continue**     ▷ No threshold can split identical feature values.
33:         **end if**
34:         $n_L \leftarrow |\mathcal{I}_L|, \quad n_R \leftarrow |\mathcal{I}| - n_L$
35:         **if** $n_L < m_{\min}^{resolved}$ **or** $n_R < m_{\min}^{resolved}$ **then**
36:             **continue**
37:         **end if**
38:         **while** $q \leq |\mathcal{P}_f|$ **and** $\mathcal{P}_f[q] < v_{cur}$ **do**
39:             $q \leftarrow q + 1$
40:         **end while**
41:         $q' \leftarrow q$
42:         **while** $q' \leq |\mathcal{P}_f|$ **and** $\mathcal{P}_f[q'] < v_{next}$ **do**
43:             $\tau \leftarrow \mathcal{P}_f[q']$
44:             $\text{OBJLEAF}_L \leftarrow \texttt{LossFromCholesky}(\mathbf{L}_L, \boldsymbol{b}_L, \|\boldsymbol{y}_{\mathcal{I}_L}\|^2) + \lambda_s$
45:             $\text{OBJLEAF}_R \leftarrow \texttt{LossFromCholesky}(\mathbf{L}_R, \boldsymbol{b}_R, \|\boldsymbol{y}_{\mathcal{I}_R}\|^2) + \lambda_s$
46:             **yield** $(f, \tau, \mathcal{I}_L, \text{OBJLEAF}_L, \text{OBJLEAF}_R, \mathbf{L}_L, \mathbf{L}_R, \boldsymbol{b}_L, \boldsymbol{b}_R, \|\boldsymbol{y}_{\mathcal{I}_L}\|^2, \|\boldsymbol{y}_{\mathcal{I}_R}\|^2)$ ▷ All thresholds in this interval reuse the same sufficient statistics.
47:             $q' \leftarrow q' + 1$
48:         **end while**
49:         $q \leftarrow q'$
50:     **end for**
51: **end for**

---

**Algorithm 7** `BuildThresholdPool`: global candidate thresholds used at every node

---

**Require:** Data matrix $\mathbf{X}$; global sorted lists $\Pi$; binary features $\mathcal{B}$; continuous features $\mathcal{C}$; number of thresholds $T$; threshold strategy `strategy`

1: Initialize $\mathcal{P}_f \leftarrow \emptyset$ for every feature $f$
2: **for** $f = 1$ **to** $k$ **do**
3:      **if** $f \in \mathcal{B}$ **then**
4:          $\mathcal{P}_f \leftarrow \{0.5\}$
5:          **continue**
6:      **end if**
7:      Let $(v_1, \ldots, v_n)$ be the feature-$f$ values sorted according to $\pi_f(\mathcal{I})$
8:      **if** $T \leq 0$ **or** $v_1 = v_n$ **then**
9:          **continue**
10:      **end if**
11:      $(u_1, \ldots, u_s) \leftarrow$ `DeduplicateSortedValues`$(v_1, \ldots, v_n)$
12:      **if** $s \leq T + 1$ **then**
13:          $\mathcal{P}_f \leftarrow \{(u_j + u_{j+1})/2 : j = 1, \ldots, s - 1\}$          ▷ Use all adjacent midpoints if there are few unique values.
14:      **else if** `strategy` = `uniform` **then**
15:          $\mathcal{P}_f \leftarrow$ `UniformThresholds`$(v_1, v_n, T)$
16:      **else if** `strategy` = `quantile` **then**
17:          Let $\widehat{Q}_f(\alpha)$ denote the empirical $\alpha$-quantile of the sorted feature-$f$ values $(v_1, \ldots, v_n)$
18:          $\mathcal{P}_f \leftarrow \{\widehat{Q}_f(k/(T+1)) : k = 1, \ldots, T\}$          ▷ Empirical quantiles of the sorted feature values.
19:      **else**
20:          Raise error: unknown threshold strategy
21:      **end if**
22:      Remove duplicate values from $\mathcal{P}_f$ while preserving sorted order
23: **end for**
24: **return** $\mathcal{P} = \{\mathcal{P}_f\}_{f=1}^k$

---

---

**Algorithm 8** `SplitSortedIndices`: stable filtering of node-wise sorted lists

---

**Require:** Node-wise sorted lists $\Pi_{\mathcal{I}} = \{\pi_f(\mathcal{I})\}_f$; left index set $\mathcal{I}_L$

1: Build membership mask `inLeft`$[i] \leftarrow \mathbf{1}\{i \in \mathcal{I}_L\}$
2: **for** each feature $f$ **do**
3:      Initialize empty lists $\pi_f(\mathcal{I}_L)$ and $\pi_f(\mathcal{I}_R)$
4:      **for** each row index $r$ in $\pi_f(\mathcal{I})$, in order **do**
5:          **if** `inLeft`$[r] = 1$ **then**
6:              Append $r$ to $\pi_f(\mathcal{I}_L)$
7:          **else**
8:              Append $r$ to $\pi_f(\mathcal{I}_R)$
9:          **end if**
10:      **end for**
11: **end for**
12: **return** $\Pi_{\mathcal{I}_L}, \Pi_{\mathcal{I}_R}$

---

---

**Algorithm 9** `RecomputeStatsFromRows`: compute ridge sufficient statistics for a row set

---

**Require:** Row set $\mathcal{S}$; scaled ridge parameter $\kappa_s$
 1: $\mathbf{A} \leftarrow \kappa_s I$ and set the intercept ridge entry to a negligible value
 2: $\boldsymbol{b} \leftarrow \mathbf{0}, \quad \|\boldsymbol{y}_\mathcal{S}\|^2 \leftarrow 0$
 3: **for** each $i \in \mathcal{S}$ **do**
 4: $\quad \mathbf{A} \leftarrow \mathbf{A} + \boldsymbol{\phi}_i \boldsymbol{\phi}_i^\top$
 5: $\quad \boldsymbol{b} \leftarrow \boldsymbol{b} + \boldsymbol{\phi}_i \widetilde{y}_i$
 6: $\quad \|\boldsymbol{y}_\mathcal{S}\|^2 \leftarrow \|\boldsymbol{y}_\mathcal{S}\|^2 + \widetilde{y}_i^2$
 7: **end for**
 8: $\mathbf{L} \leftarrow \texttt{chol}(\mathbf{A})$
 9: **return** $\mathbf{L}, \boldsymbol{b}, \|\boldsymbol{y}_\mathcal{S}\|^2$

---

**Algorithm 10** `RestrictData`: create child-node views via index sets

---

**Require:** Node data $D_\mathcal{I} = \{(\boldsymbol{x}_i, y_i)\}_{i \in \mathcal{I}}$; left index set $\mathcal{I}_L$
 1: $\mathcal{I}_R \leftarrow \mathcal{I} \setminus \mathcal{I}_L$
 2: Define $D_{\mathcal{I}_L}$ as a view of $D_\mathcal{I}$ restricted to indices in $\mathcal{I}_L$
 3: Define $D_{\mathcal{I}_R}$ as a view of $D_\mathcal{I}$ restricted to indices in $\mathcal{I}_R$
 4: **return** $D_{\mathcal{I}_L}, D_{\mathcal{I}_R}$

---

**Algorithm 11** `LossFromCholesky`: ridge leaf loss from Cholesky sufficient statistics

---

**Require:** Cholesky factor $\mathbf{L}$ of $\mathbf{A} = \Phi^\top \Phi + \kappa_s I$; vector $\boldsymbol{b} = \Phi^\top \boldsymbol{y}$; response norm $\|\boldsymbol{y}\|^2$
 1: $\boldsymbol{z} \leftarrow \texttt{triSolve}(\mathbf{L}, \boldsymbol{b})$                $\triangleright$ Solve $\mathbf{L}\boldsymbol{z} = \boldsymbol{b}$ by forward substitution.
 2: $\ell \leftarrow \|\boldsymbol{y}\|^2 - \|\boldsymbol{z}\|^2$
 3: **return** $\ell$

---

**Algorithm 12** `SolveRidge`: solve $(\mathbf{L}\mathbf{L}^\top)\boldsymbol{\beta} = \boldsymbol{b}$

---

**Require:** Cholesky factor $\mathbf{L}$ and vector $\boldsymbol{b}$
 1: $\boldsymbol{z} \leftarrow \texttt{triSolve}(\mathbf{L}, \boldsymbol{b})$              $\triangleright$ Solve $\mathbf{L}\boldsymbol{z} = \boldsymbol{b}$.
 2: $\boldsymbol{\beta} \leftarrow \texttt{triSolve}(\mathbf{L}^\top, \boldsymbol{z})$           $\triangleright$ Solve $\mathbf{L}^\top \boldsymbol{\beta} = \boldsymbol{z}$.
 3: **return** $\boldsymbol{\beta}$

---

**Algorithm 13** `triSolve`: triangular solve

---

**Require:** Triangular matrix $\mathbf{L} \in \mathbb{R}^{k \times k}$; vector $\boldsymbol{b} \in \mathbb{R}^k$
 1: **for** $i = 1$ **to** $k$ **do**
 2: $\quad z_i \leftarrow \left( b_i - \sum_{j=1}^{i-1} L_{ij} z_j \right) / L_{ii}$
 3: **end for**
 4: **return** $\boldsymbol{z}$

---

**Algorithm 14** `cholupdate`: rank-one Cholesky update/downdate interface

---

**Require:** Cholesky factor $\mathbf{L}$ of $\mathbf{A}$; vector $\boldsymbol{x}$; sign $s \in \{+1, -1\}$
 1: Apply a stable rank-one Cholesky update/downdate routine to obtain $\mathbf{L}'$      $\triangleright$ $\mathbf{L}'\mathbf{L}'^\top = \mathbf{A} + s\boldsymbol{x}\boldsymbol{x}^\top$.
 2: **return** $\mathbf{L}'$

---

# D. Experiment Details

This section describes the details of our experimental setup. We first introduce the regression datasets used in our study, together with their sizes and prediction targets (Appendix D.1). We then summarize the dataset-specific preprocessing steps applied (Appendix D.2). We also describe the synthetic data generating process used in our controlled benchmarks (Appendix D.3). Next, we describe the hardware platform on which all experiments were conducted and the runtime constraints (Appendix D.4). Finally, we provide implementation details of the baseline algorithms and the software packages used in our experiments (Appendix D.5).

## D.1. Datasets

We conduct our experiments on a collection of regression datasets obtained from the UCI Machine Learning Repository (Dua & Graff, 2017), the Medical Cost Personal dataset from Choi (2018), and two publicly available regression benchmarks hosted on Kaggle, namely the California Housing dataset (Nugent, 2017) and the Walmart Store Sales dataset (H, 2021). Each dataset contains a set of numerical features (continuous and/or binary after encoding) and a designated prediction target variable as follows:

- **Airfoil**: contains frequency, attack-angle, chord-length, free-stream-velocity, and suction-side-displacement-thickness; the prediction target is *scaled-sound-pressure*. License: CC BY 4.0

- **Auction**: contains the features capacities of bidders 1–4, price currently verified, product currently verified, and current verified winner of the product; the prediction target is *runtime of the verification procedure*. License: CC BY 4.0

- **Auto MPG**: contains displacement, cylinders, horsepower, weight, acceleration, model_year, and origin; the prediction target is *mpg*. License: CC BY 4.0

- **Energy (Cooling / Heating)**: contains relative compactness ($X1$), surface area ($X2$), wall area ($X3$), roof area ($X4$), overall height ($X5$), orientation ($X6$), glazing area ($X7$), and glazing area distribution ($X8$); the prediction targets are *heating load ($Y1$)* and *cooling load ($Y2$)*. License: CC BY 4.0

- **Insurance**: contains *age*, *sex*, *bmi*, *children*, *smoker*, and categorical encodings for *region_northeast*, *region_northwest*, *region_southeast*, and *region_southwest*; the prediction target is *charges*. License: DbCL v1.0

- **Optical Net**: contains node number, thread number, T/R, processor utilization, channel waiting time, input waiting time, and network response time; the prediction target is *channel utilization*. License: CC BY 4.0

- **Real Estate**: contains transaction date ($X1$), house age ($X2$), distance to the nearest MRT station ($X3$), number of convenience stores ($X4$), latitude ($X5$), and longitude ($X6$); the prediction target is *house price of unit area ($Y$)*. License: CC BY 4.0

- **Servo**: contains *pgain* and *vgain*; the prediction target is *class (numeric)*. License: CC BY 4.0

- **Synch**: contains load current ($Iy$), power factor ($PF$), power factor error ($e$), and change of excitation current ($dIf$); the prediction target is *excitation current of the synchronous machine ($If$)*. License: CC BY 4.0

- **Yacht**: contains longitudinal position, prismatic coefficient, length-displacement, beam-draught ratio, length-beam ratio, and Froude number; the prediction target is *residuary resistance*. License: CC BY 4.0

- **California Housing**: contains longitude, latitude, housing median age, total rooms, total bedrooms, population, households, median income, and ocean proximity; the prediction target is *median house value*. License: CC0: Public Domain

- **Temperature**: contains numerical weather prediction (NWP) forecasts from the LDAPS model, present-day in-situ temperature observations, and geographical auxiliary variables over Seoul, South Korea; the prediction target is *next-day air temperature*. License: CC BY 4.0

- **Seoul Bike**: contains hourly weather conditions and temporal information including temperature, humidity, wind speed, visibility, dew point temperature, solar radiation, rainfall, snowfall, and hour of day; the prediction target is *hourly rented bike count*. License: CC BY 4.0

- **Walmart**: contains holiday indicator, temperature, fuel price, consumer price index (CPI), and unemployment rate; the prediction target is *weekly sales*. License: CC0: Public Domain

*Table 3.* Dataset statistics and feature dimensionality after binarization. "20-threshold" denotes binarization using 20 uniform thresholds per continuous feature. "Full" denotes full enumeration of all possible split thresholds when computationally feasible.

| Dataset | #Rows | Original #Features | #Binary | #Continuous | #Features (20-threshold) | #Features (full) |
|---|---|---|---|---|---|---|
| *Small / Medium-scale datasets* | | | | | | |
| Airfoil | 1503 | 5 | 0 | 5 | 64 | 158 |
| Auction | 2043 | 7 | 2 | 5 | 31 | 47 |
| Auto MPG | 392 | 7 | 0 | 7 | 98 | 629 |
| Energy (Cooling) | 768 | 8 | 1 | 7 | 43 | 43 |
| Energy (Heating) | 768 | 8 | 1 | 7 | 43 | 43 |
| Insurance | 1338 | 9 | 6 | 3 | 51 | 604 |
| Optical Net | 640 | 7 | 1 | 6 | 93 | 2553 |
| Real Estate | 414 | 6 | 0 | 6 | 101 | 978 |
| Servo | 167 | 2 | 0 | 2 | 7 | 7 |
| Synch | 557 | 4 | 0 | 4 | 80 | 405 |
| Yacht | 308 | 6 | 0 | 6 | 58 | 58 |
| *Large-scale datasets (full binarization infeasible)* | | | | | | |
| California Housing | 20433 | 13 | 5 | 8 | 165 | – |
| Temperature (Max) | 7590 | 21 | 0 | 21 | 359 | – |
| Temperature (Min) | 7590 | 21 | 0 | 21 | 359 | – |
| Seoul Bike | 8760 | 9 | 0 | 9 | 131 | – |
| Walmart | 6435 | 5 | 1 | 4 | 81 | – |

## D.2. Data Preprocessing

Across all datasets, we preserve the original feature names and place the prediction target in the last column of each CSV. We do not apply a unified strategy to standardize or normalize feature values unless necessary. Below we list only the required preprocessing steps; datasets not mentioned require no additional preprocessing.

- **Auction:** We drop the categorical verification label *verification.result* and keep seven numeric inputs (*process.b1–b4.capacity*, *property.price*, *property.product*, *property.winner*).

- **Auto MPG:** We remove rows with missing *horsepower* ($\sim$1.5% of rows). All seven numeric predictors are kept in the continuous split.

- **Energy (Heating):** We use predictors $X1$–$X8$ and drop $Y2$.

- **Energy (Cooling):** We use predictors $X1$–$X8$ and drop $Y1$.

- **Insurance:** For experiments that include categorical signals we use 0/1 encodings for *sex* and *smoker* and full one-hot encoding for *region*.

- **Optical Net:** The raw file is semicolon-separated with commas as decimal points; we standardize decimals, drop *Spatial Distribution*, *Temporal Distribution*, and any unnamed columns, and retain the remaining seven continuous predictors.

- **Servo:** We remove the categorical columns *motor* and *screw*.

- **Synch:** The raw file uses semicolons and comma decimals; we normalize decimals to dots and cast to float. Based on the dataset notes, we rescale *Iy* by a factor of 10 to correct its magnitude.

- **California Housing:** We remove missing entries in *total_bedrooms* and apply full one-hot encoding to the categorical variable *ocean_proximity*, resulting in five binary indicators. All remaining attributes are treated as continuous predictors, and the target variable is *median_house_value*.

- **Temperature (Max):** We remove station identifiers and date information and retain meteorological forecasts, present-day maximum and minimum temperatures, and geographical variables as continuous predictors. Samples with missing values are discarded. The regression target is *Next_Tmax*.

- **Temperature (Min):** We remove station identifiers and date information and retain meteorological forecasts, present-day maximum and minimum temperatures, and geographical variables as continuous predictors. Samples with missing values are discarded. The regression target is *Next_Tmin*.

- **Seoul Bike:** We remove calendar attributes and categorical indicators (*Date*, *Seasons*, *Holiday*, *Functional Day*) and retain hourly meteorological variables as continuous predictors. The hour-of-day variable is treated as numeric, and the regression target is *Rented_Bike_Count*.

- **Walmart:** We remove store identifiers and calendar variables, retaining only numeric economic and environmental predictors. The binary variable *Holiday_Flag* is kept as a 0/1 indicator, while temperature, fuel price, CPI, and unemployment rate are treated as continuous features. The regression target is *Weekly_Sales*.

### D.3. Synthetic Data Generating Process

We generate a synthetic regression dataset to test the speedup of CLARITree relative to the implementation without the rank-one Cholesky update, with a piecewise-linear structure. For $i = 1, \ldots, n$, covariates are sampled as

$$\boldsymbol{x}_i \sim \mathcal{N}(\mathbf{0}, \boldsymbol{\Sigma}), \qquad \boldsymbol{\Sigma} \in \mathbb{R}^{k \times k}, \quad \Sigma_{jr} = \rho^{|j-r|}, \quad 1 \le j, r \le k.$$

Samples are assigned to $G$ latent groups using the first coordinate. Let $q_1, \ldots, q_{G-1}$ denote the empirical $(\ell/G)$-quantiles of $\{x_{i1}\}_{i=1}^{n}$, and define

$$g_i = \sum_{\ell=1}^{G-1} \mathbf{1}\{x_{i1} > q_\ell\}, \qquad g_i \in \{0, \ldots, G-1\}.$$

Each group $g$ is associated with a distinct regression coefficient vector $\boldsymbol{\beta}^{(g)} \in \mathbb{R}^k$ given by

$$\boldsymbol{\beta}^{(g)} = s_g\, \boldsymbol{d} \odot \boldsymbol{\xi}^{(g)}, \quad \boldsymbol{\xi}^{(g)} \sim \mathcal{N}(\mathbf{0}, \mathbf{I}_k), \quad \boldsymbol{d} \in \mathbb{R}^k,$$

where the $r$-th entry of $\boldsymbol{d}$ is $d_r = \exp\left(-2\frac{r-1}{k-1}\right)$, $s_g = 1 + 0.5g$. The response is generated according to

$$y_i = \boldsymbol{x}_i^\top \boldsymbol{\beta}^{(g_i)} + \varepsilon_i, \qquad \varepsilon_i \sim \mathcal{N}(0, \sigma^2).$$

This construction induces correlated features, axis-aligned regime boundaries, and an exactly linear conditional mean within each regime. As a result, it provides a controlled setting in which different regression solvers operate on identical statistical problems, isolating computational effects without confounding modeling differences.

### D.4. Experiment Platform

All experiments were conducted on a shared high-performance computing cluster. Each compute node is equipped with an Intel Xeon Gold 6226 CPU (2.7 GHz, 48 cores, 768 GB RAM). Unless otherwise noted, training and evaluation were performed using the CPU cores. All algorithms were executed in single-threaded mode. The time limitation is 600 seconds for all experiments.

### D.5. Software Packages

**Generalized, Unbiased, Interaction Detection, and Estimation (GUIDE):**

We used the GUIDE executable binary `guide.gz` for Linux (compiled with `gfortran 11.4.0` on Ubuntu 22.04 LTS, version 45.0). The executable was obtained from the official GUIDE website (`https://pages.stat.wisc.edu/~loh/guide.html`).

**Optimal Sparse Regression Tree (OSRT):** We used the open-source implementation of OSRT (version 0.2.2), which is publicly available at `https://github.com/ruizhang1996/optimal-sparse-regression-tree-public`.

**Separable Trees with Dynamic Programming (STreeD):** We used the open-source implementation of STreeD (version 1.3.7), which is publicly available at `https://github.com/AlgTUDelft/pystreed`. In our experiments, we removed the default constraint on the minimum number of instances per leaf. The original implementation enforces that each leaf must contain at least five times the number of features, which can lead to suboptimal performance on some datasets. We remove this restriction to ensure a fair comparison and to allow STreeD to achieve its best possible performance.

**Fast Linear Model Trees by PILOT (PILOT):** We used the open-source implementation of PILOT, which is publicly available at `https://github.com/STAN-UAntwerp`.

**Learning With Continuous Classes (M5):** We used the open-source implementation of M5 (M5P), which is publicly available at `https://smarie.github.io/python-m5p`. The M5 algorithm performs an exhaustive threshold search at each node and selects splits based on impurity reduction computed using the target mean values rather than the linear regression error. Unlike STreeD, M5 does not provide explicit control over the number of candidate thresholds evaluated at each split. Consequently, M5 always considers the full set of possible thresholds during split selection.

# E. Experimental Results

In this section, we present the complete set of experimental results for CLARITree across all datasets described in Appendix D. In addition to the main test performance results in Table 2, we also report the corresponding training $R^2$ values in Table 4 for the configurations achieving the best test $R^2$. To further study the impact of threshold binarization, we include results under three threshold settings: threshold = 5, threshold = 20, and the full-threshold setting. Finally, we report the training completion rates under the full-threshold setting in Figure 19.

To avoid excessive space usage in the paper, for appendix figures on small- and medium-scale datasets we report results under both the full-threshold setting and the 20-threshold binarization setting. For large-scale datasets, only the 20-threshold results are shown, since even under the 20-threshold setting, optimal tree methods remain computationally intractable within the runtime budget. The 5-threshold setting is omitted from the figures because its behavior is qualitatively similar and does not substantially change the observed trends. In the appendix tables, however, we provide the complete results for all three threshold settings. For readability, all figures use "Greedy" to denote Greedy CholeskyTree.

Since M5 does not rely on threshold binarization, its results are identical across different threshold settings. Therefore, we only report the M5 results together with the 20-threshold setting, which serves as our default experimental configuration.

## E.1. Small / Medium-Scale Datasets Figure

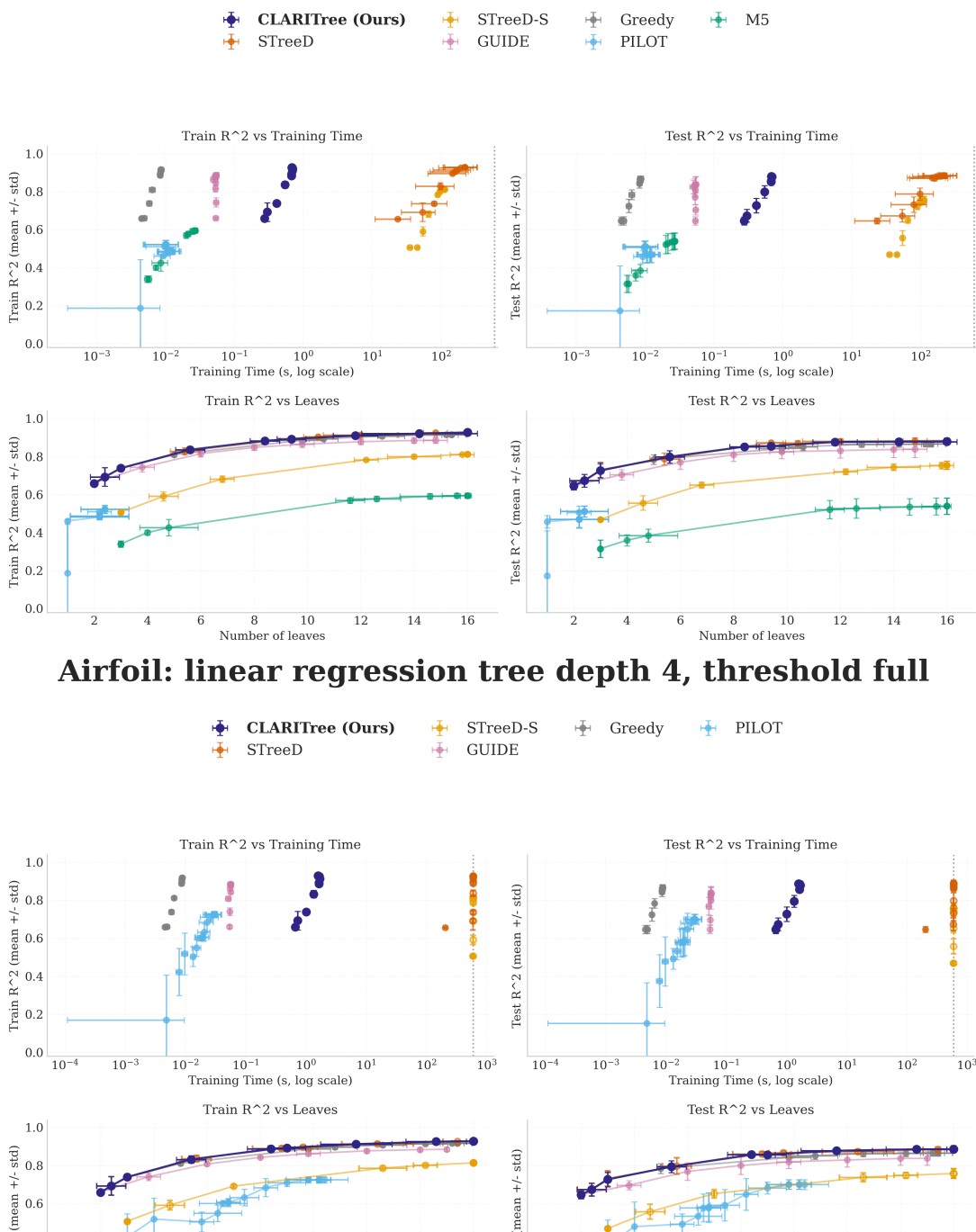

*Figure 5.* Performance of CLARITree and baselines on Airfoil. Each trained with a maximum tree depth of 4. Results averaged over five random 80/20 splits, with error bars showing ± 1 standard deviation. The dashed vertical line in the top row denotes the default time limit of 10 minutes.

# Auction: linear regression tree depth 4, threshold 20

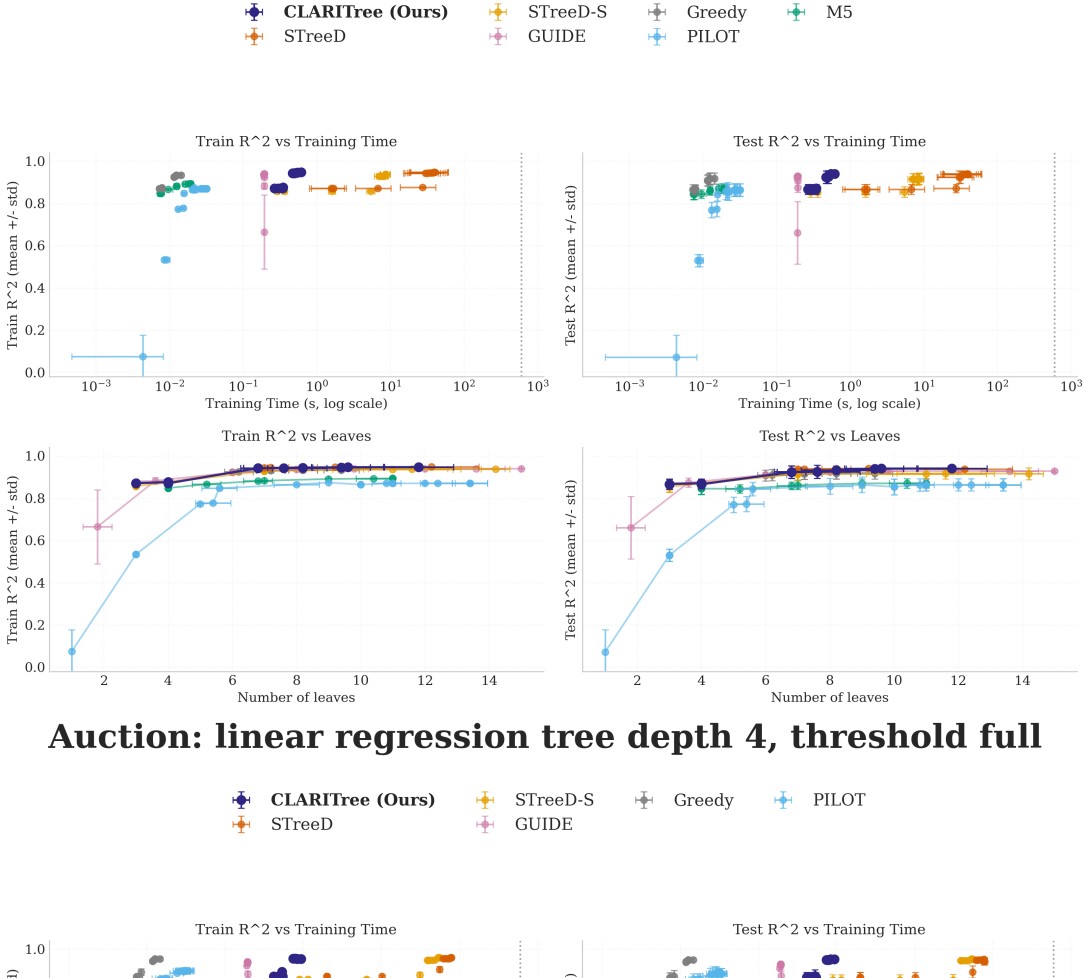

# Auction: linear regression tree depth 4, threshold full

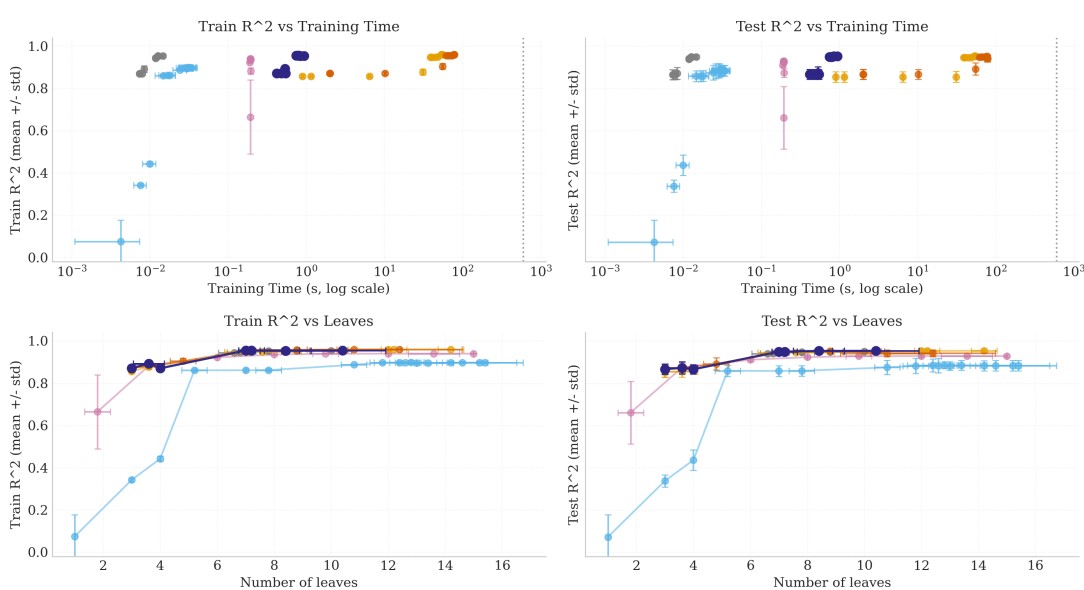

*Figure 6.* Performance of CLARITree and baselines on Auction. Each trained with a maximum tree depth of 4. Results averaged over five random 80/20 splits, with error bars showing ± 1 standard deviation. The dashed vertical line in the top row denotes the default time limit of 10 minutes.

## Auto Mpg: linear regression tree depth 4, threshold 20

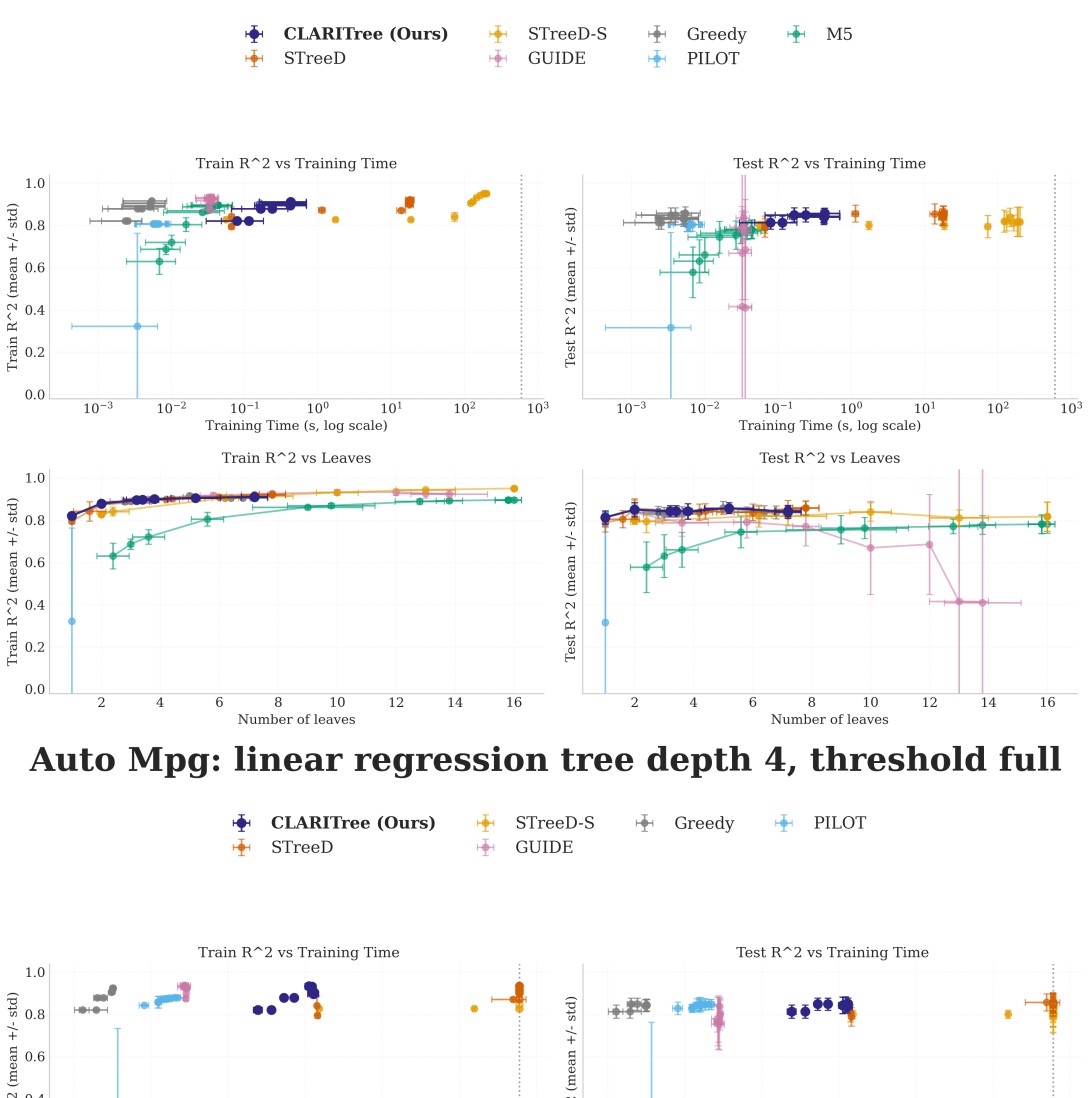

## Auto Mpg: linear regression tree depth 4, threshold full

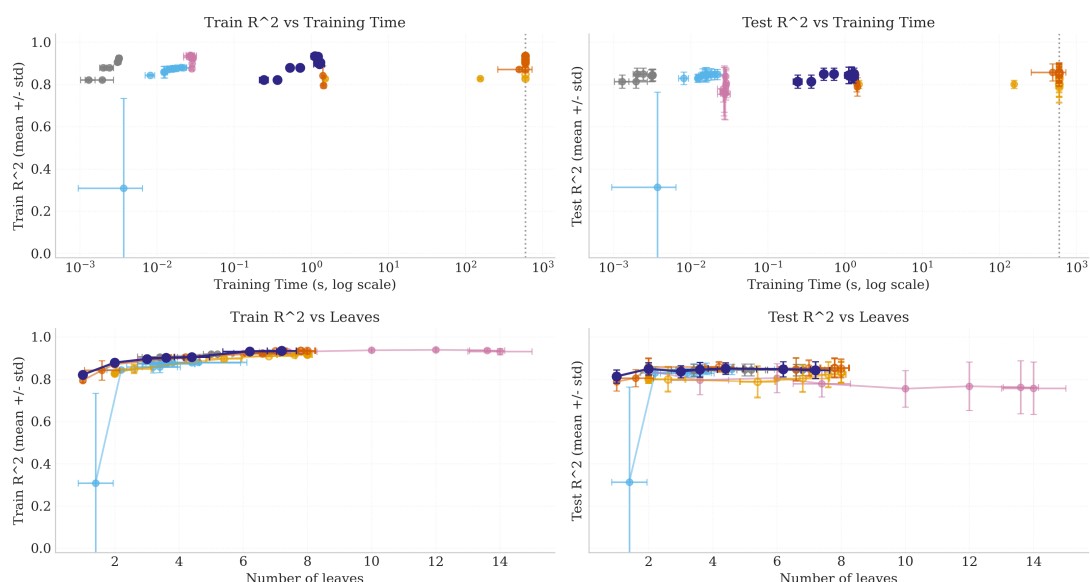

*Figure 7.* Performance of CLARITree and baselines on Auto Mpg. Each trained with a maximum tree depth of 4. Results averaged over five random 80/20 splits, with error bars showing $\pm 1$ standard deviation. The dashed vertical line in the top row denotes the default time limit of 10 minutes.

## Energy (Cooling): linear regression tree depth 4, threshold 20

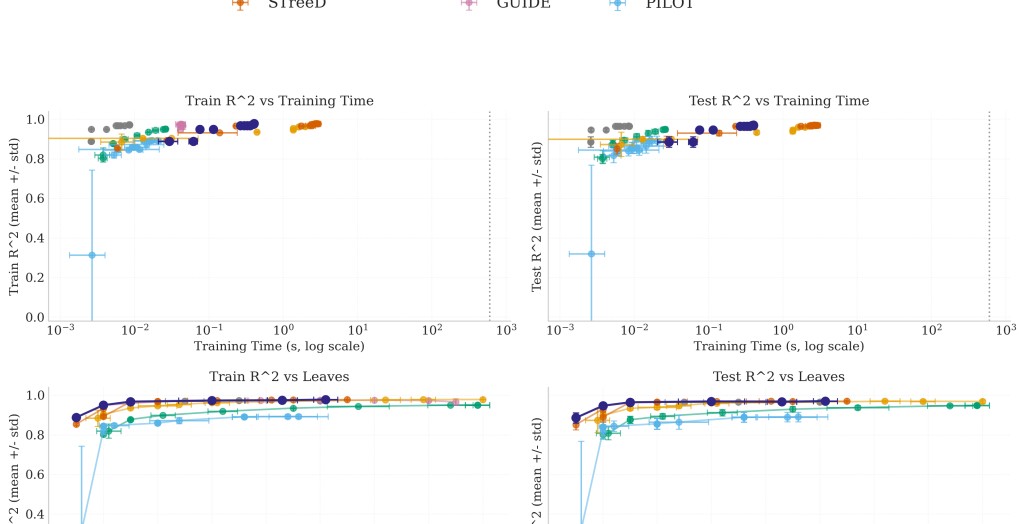

## Energy (Cooling): linear regression tree depth 4, threshold full

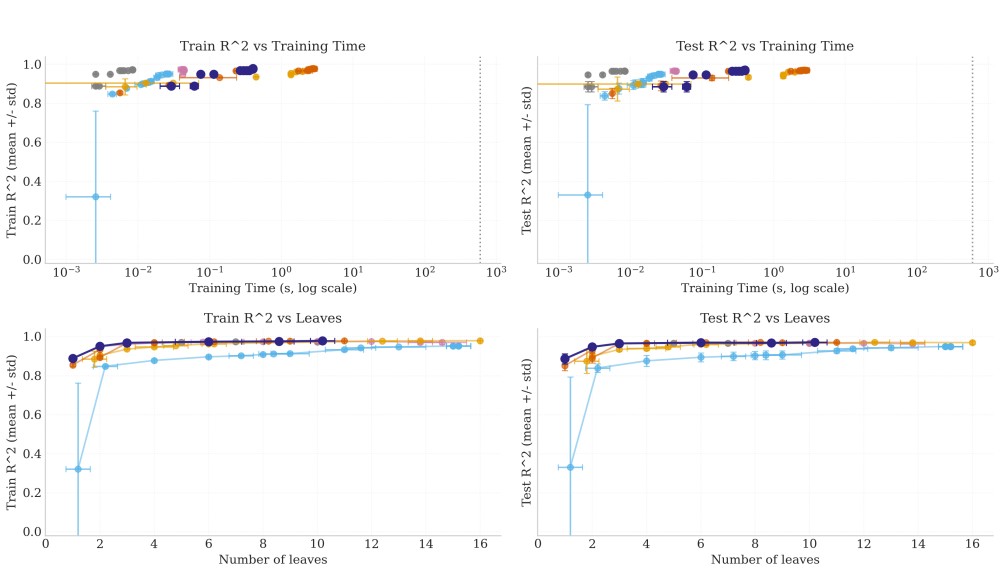

*Figure 8.* Performance of CLARITree and baselines on Energy (Cooling). Each trained with a maximum tree depth of 4. Results averaged over five random 80/20 splits, with error bars showing ± 1 standard deviation. The dashed vertical line in the top row denotes the default time limit of 10 minutes.

## Energy (Heating): linear regression tree depth 4, threshold 20

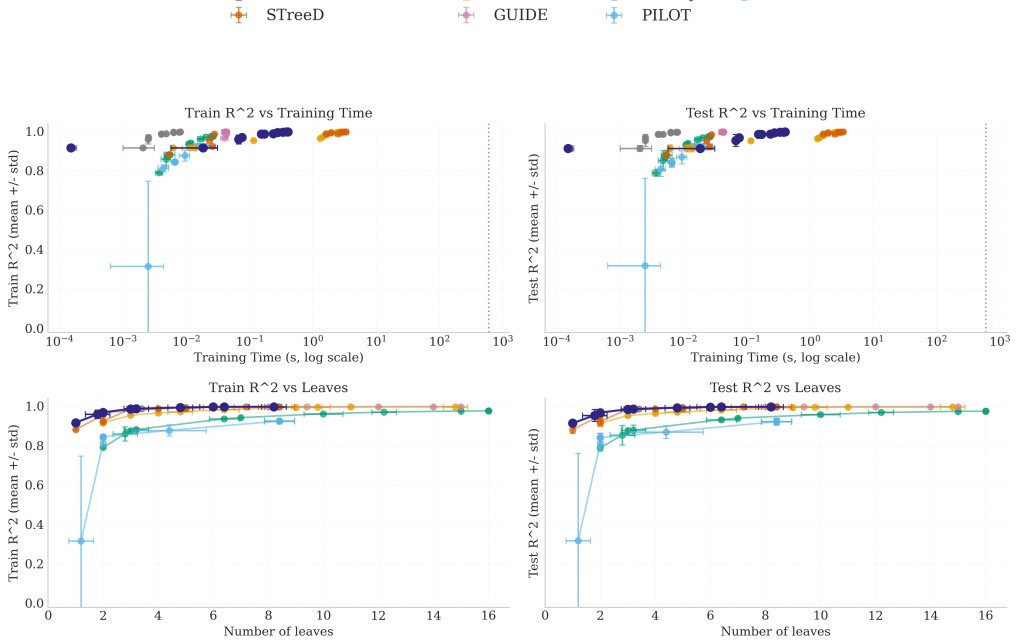

## Energy (Heating): linear regression tree depth 4, threshold full

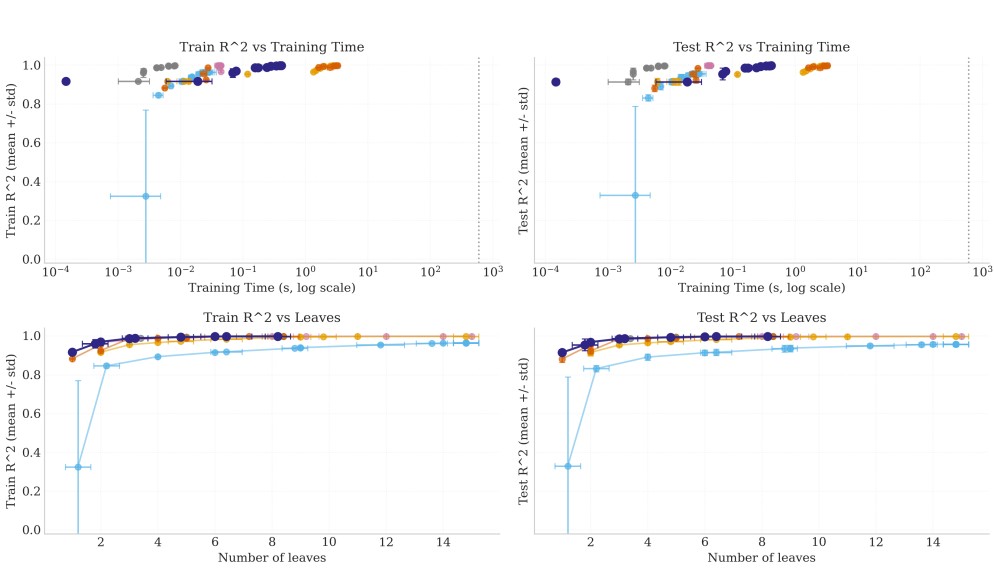

*Figure 9.* Performance of CLARITree and baselines on Energy (Heating). Each trained with a maximum tree depth of 4. Results averaged over five random 80/20 splits, with error bars showing ± 1 standard deviation. The dashed vertical line in the top row denotes the default time limit of 10 minutes.

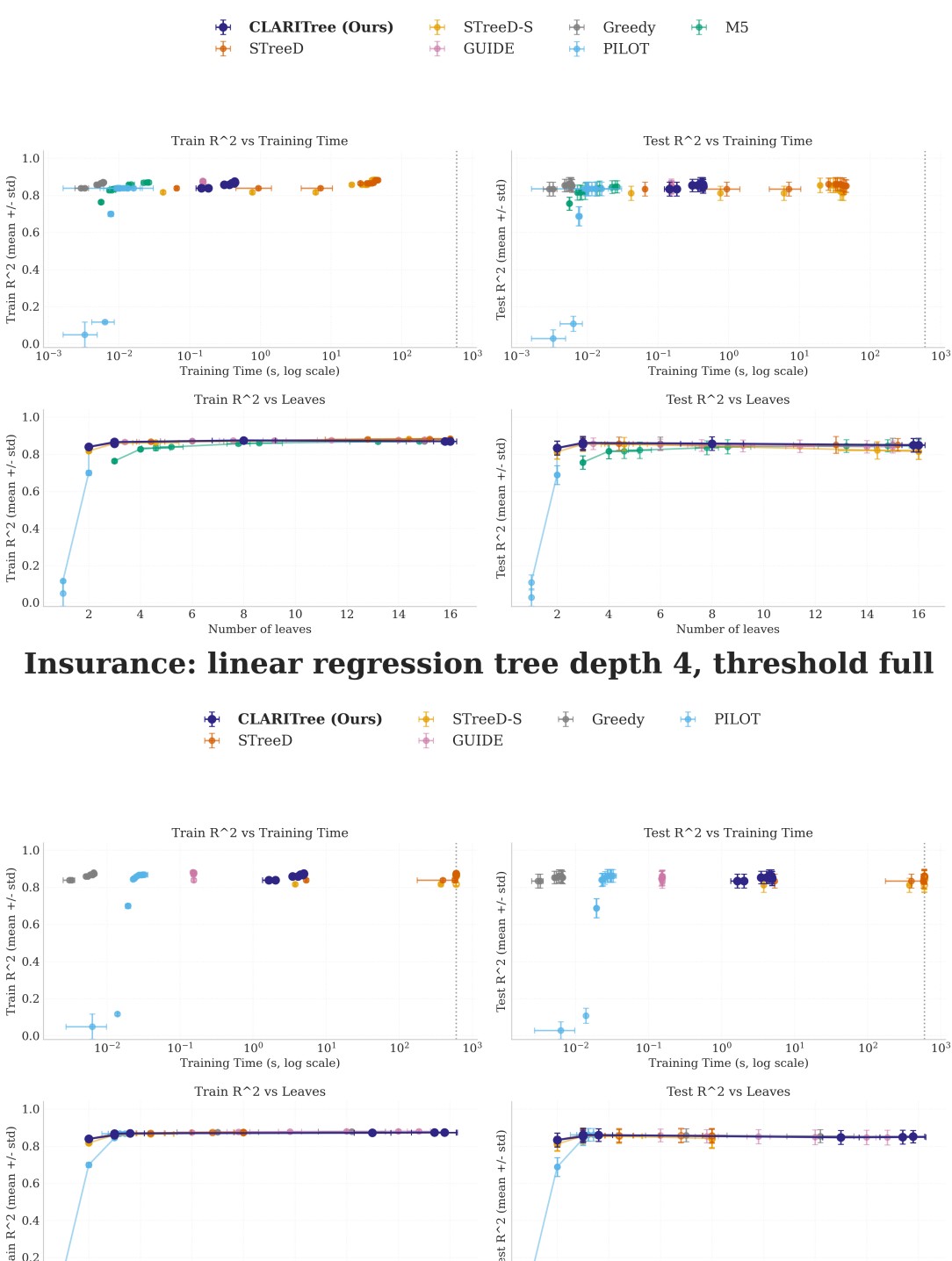

*Figure 10.* Performance of CLARITree and baselines on Insurance. Each trained with a maximum tree depth of 4. Results averaged over five random 80/20 splits, with error bars showing ± 1 standard deviation. The dashed vertical line in the top row denotes the default time limit of 10 minutes.

# Optical Net: linear regression tree depth 4, threshold 20

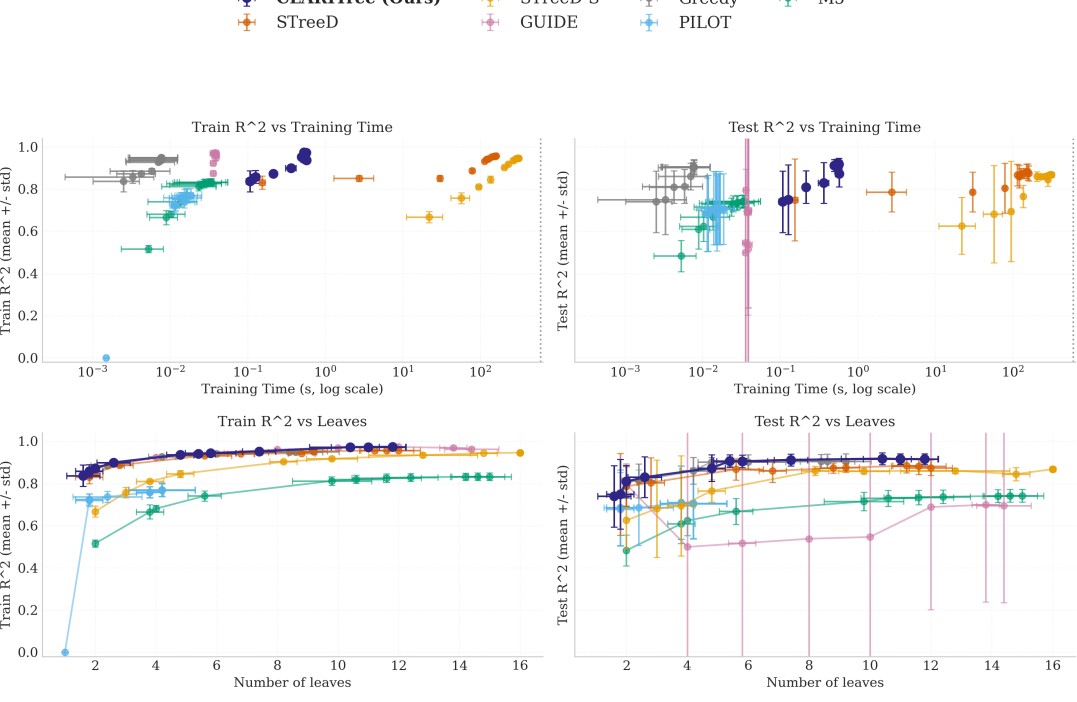

# Optical Net: linear regression tree depth 4, threshold full

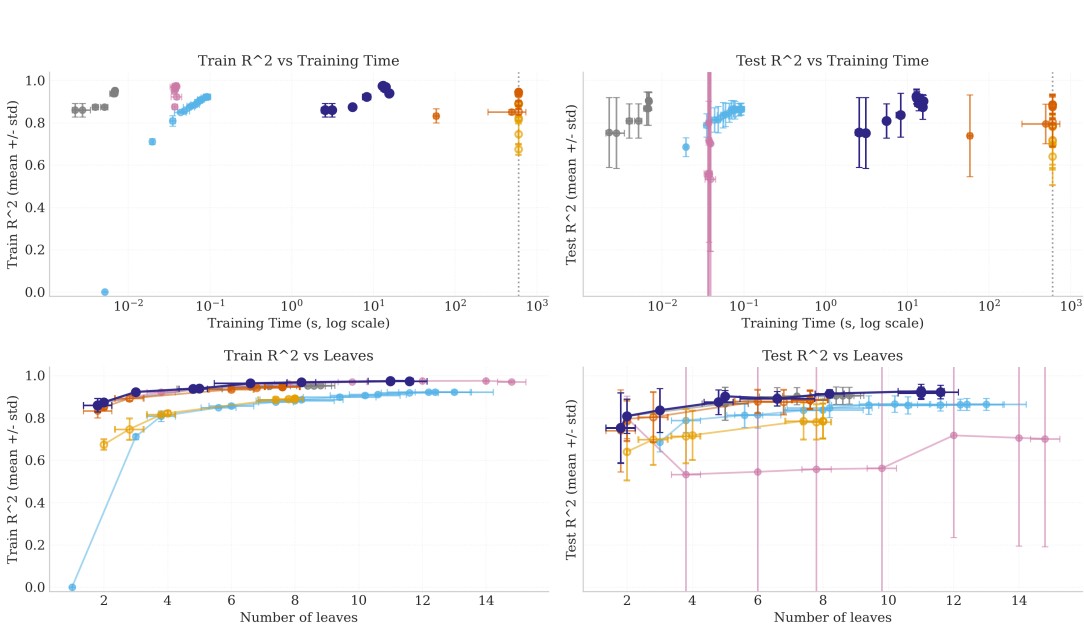

*Figure 11.* Performance of CLARITree and baselines on Optical Net. Each trained with a maximum tree depth of 4. Results averaged over five random 80/20 splits, with error bars showing $\pm 1$ standard deviation. The dashed vertical line in the top row denotes the default time limit of 10 minutes.

# Real Estate: linear regression tree depth 4, threshold 20

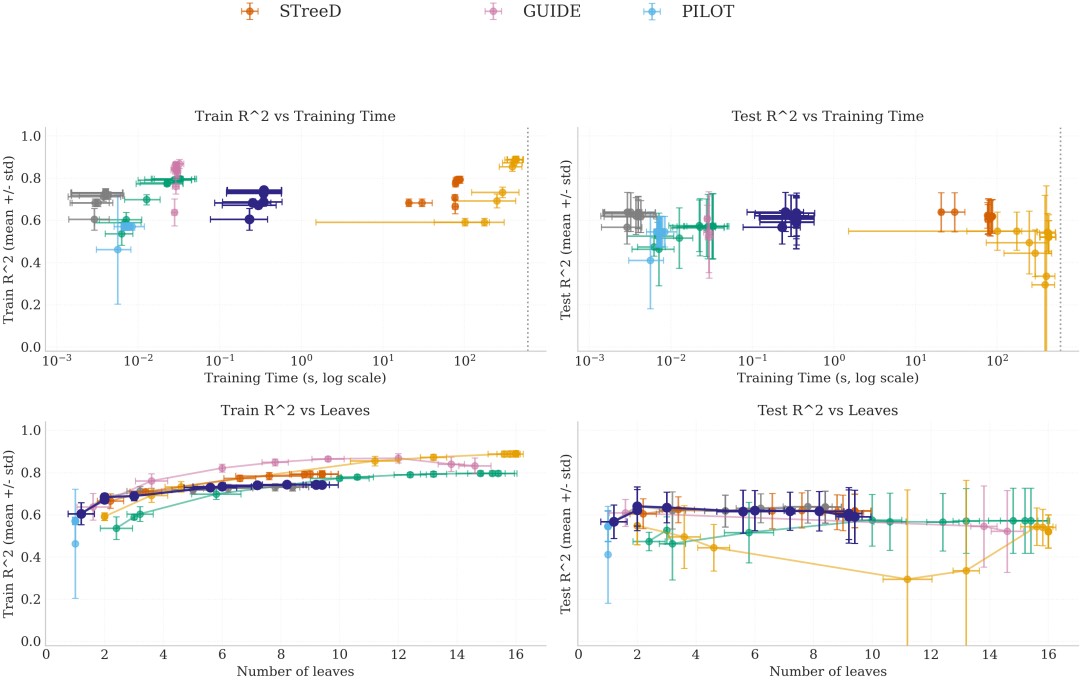

# Real Estate: linear regression tree depth 4, threshold full

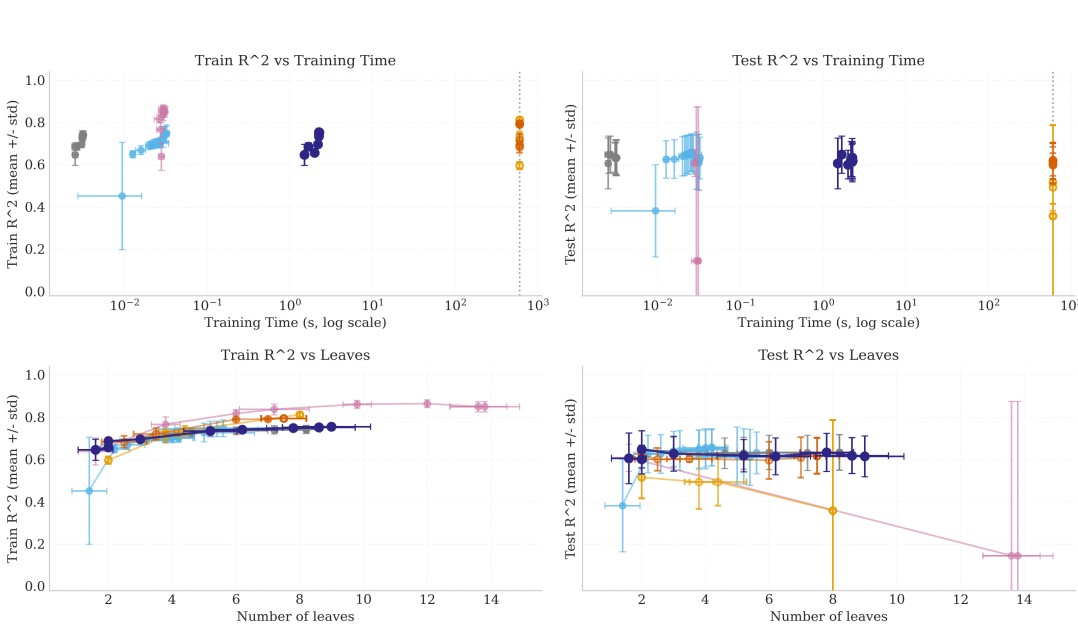

*Figure 12.* Performance of CLARITree and baselines on Real Estate. Each trained with a maximum tree depth of 4. Results averaged over five random 80/20 splits, with error bars showing $\pm 1$ standard deviation. The dashed vertical line in the top row denotes the default time limit of 10 minutes.

## Servo: linear regression tree depth 4, threshold 20

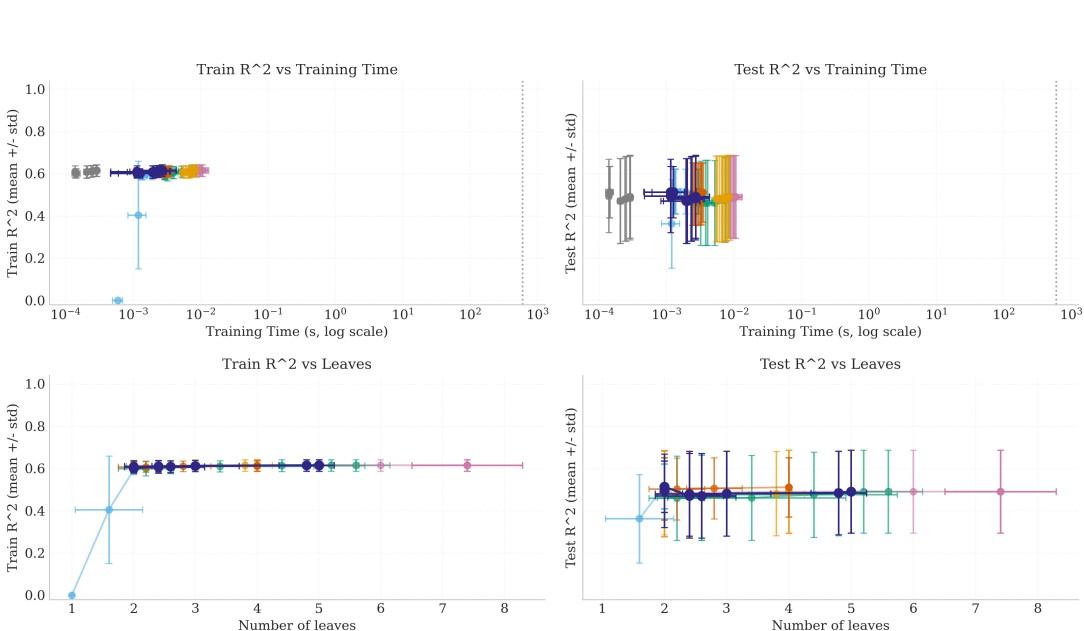

## Servo: linear regression tree depth 4, threshold full

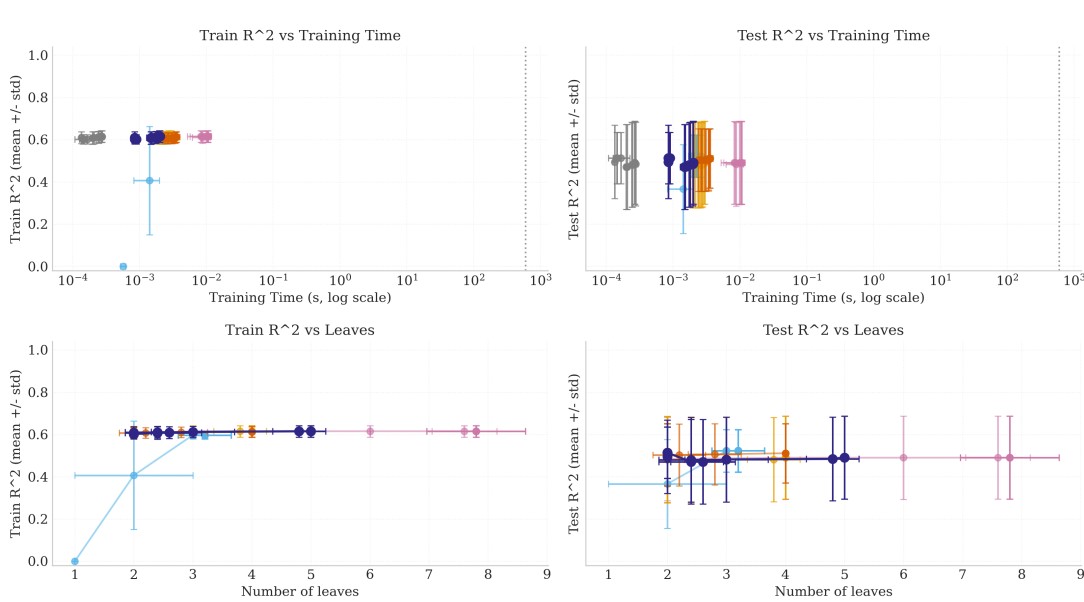

*Figure 13.* Performance of CLARITree and baselines on Servo. Each trained with a maximum tree depth of $4$. Results averaged over five random 80/20 splits, with error bars showing $\pm 1$ standard deviation. The dashed vertical line in the top row denotes the default time limit of 10 minutes. Note that Servo is an extremely small dataset with only a limited number of unique feature values (at most 7 per feature), leading to unusually high variance across splits; therefore, the results should not be over-interpreted.

# Synch: linear regression tree depth 4, threshold 20

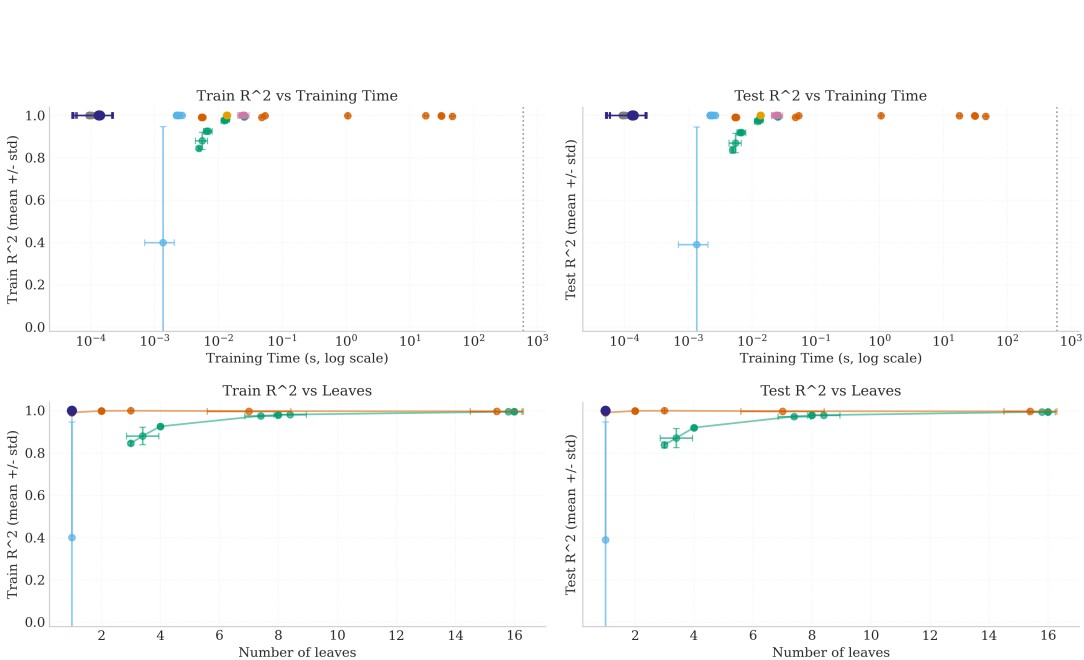

# Synch: linear regression tree depth 4, threshold full

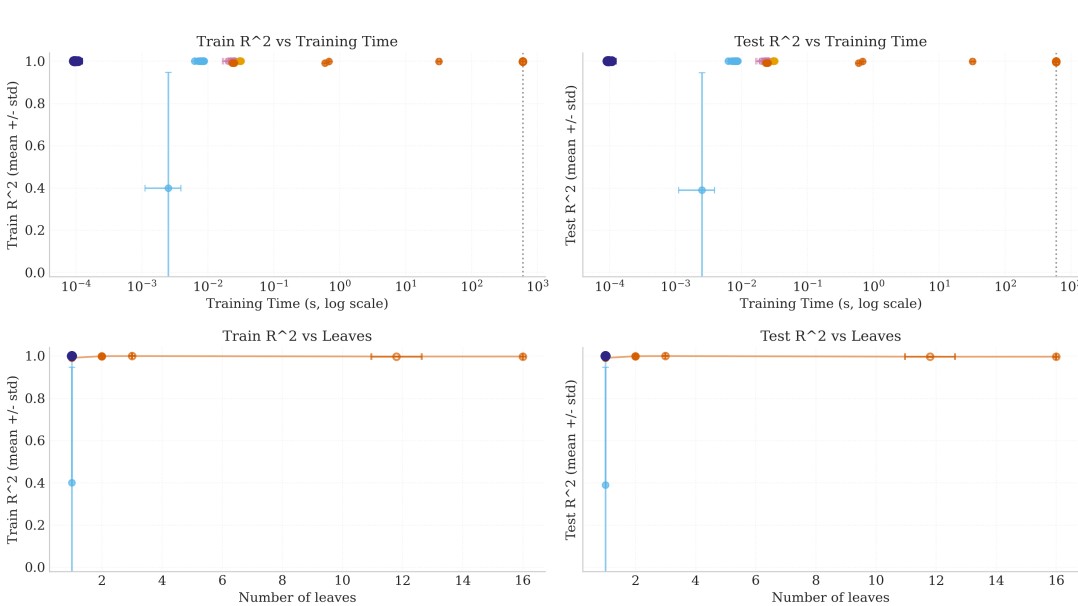

*Figure 14.* Performance of CLARITree and baselines on Synch. Each trained with a maximum tree depth of 4. Results averaged over five random 80/20 splits, with error bars showing ± 1 standard deviation. The dashed vertical line in the top row denotes the default time limit of 10 minutes. Note that Synch can be fitted extremely well using only a single-feature regression. STreeD relies on iterative optimization for node-wise regularized regression, which can be numerically unstable, whereas STreeD-S reduces each split to a one-dimensional ridge regression with a stable closed-form solution.

# Yacht: linear regression tree depth 4, threshold 20

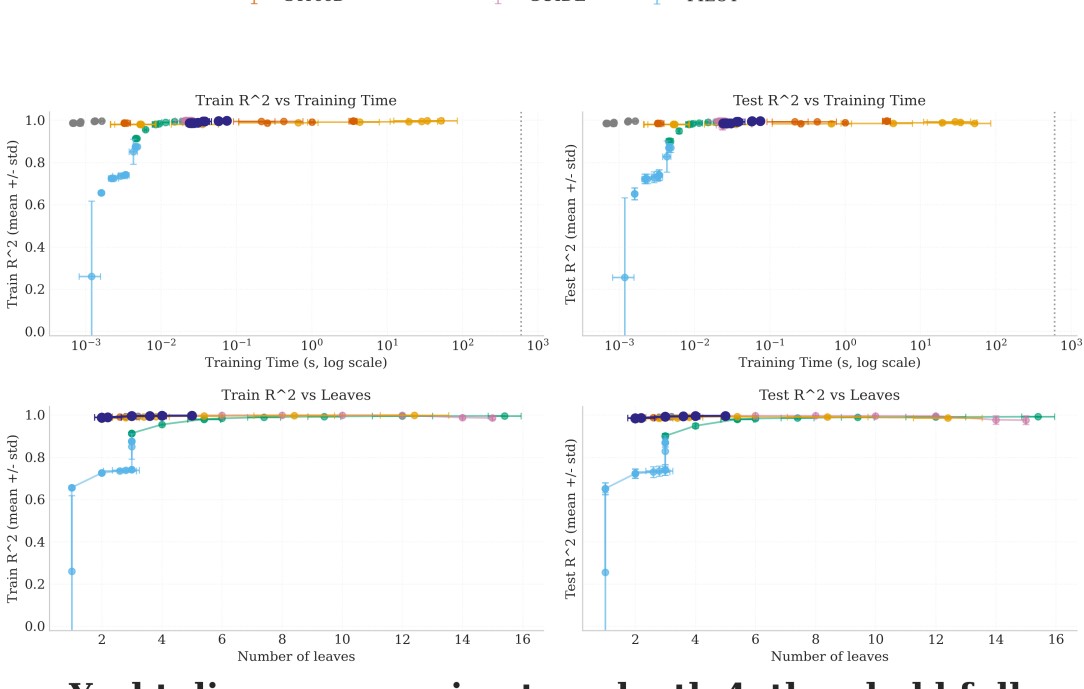

# Yacht: linear regression tree depth 4, threshold full

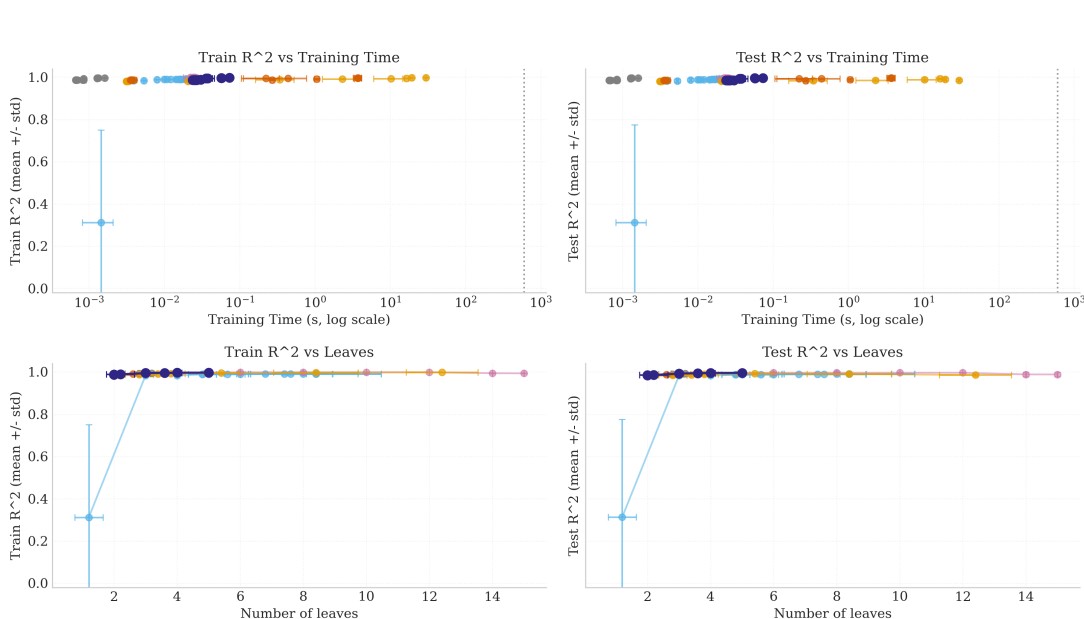

*Figure 15.* Performance of CLARITree and baselines on Yacht. Each trained with a maximum tree depth of 4. Results averaged over five random 80/20 splits, with error bars showing ± 1 standard deviation. The dashed vertical line in the top row denotes the default time limit of 10 minutes.

**E.2. Large-Scale Datasets Figure**

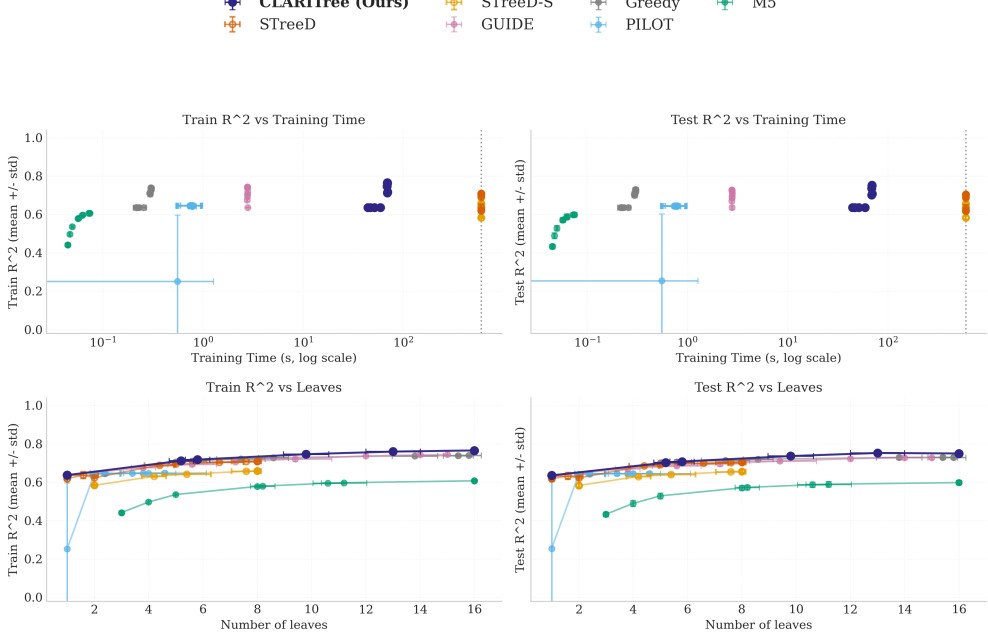

*Figure 16.* Performance of CLARITree and baselines on California Housing. Each trained with a maximum tree depth of 4. Results averaged over five random 80/20 splits, with error bars showing ± 1 standard deviation. The dashed vertical line in the top row denotes the default time limit of 10 minutes.

## Temperature (Max): linear regression tree depth 4, threshold 20

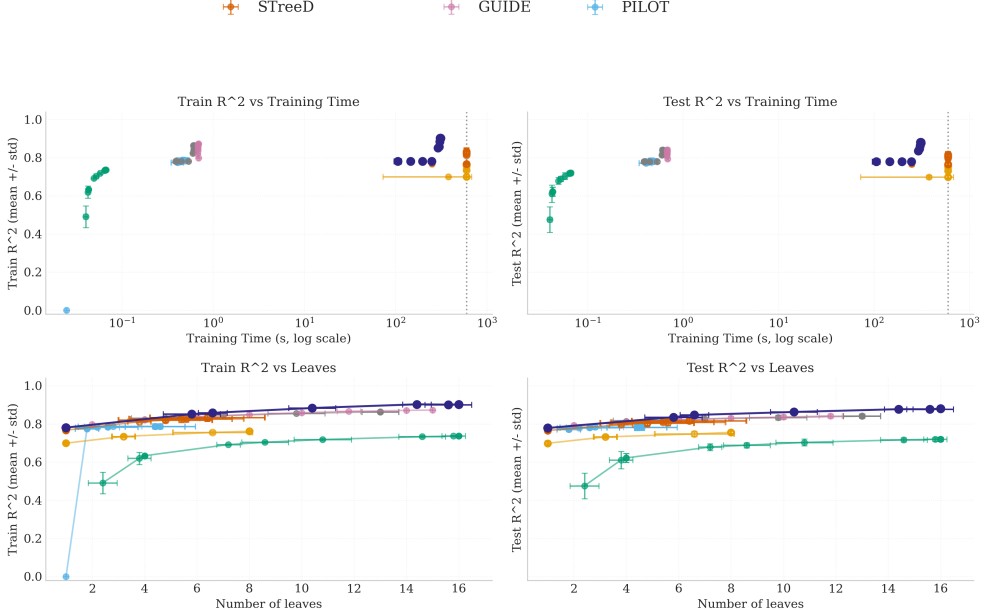

## Temperature (Min): linear regression tree depth 4, threshold 20

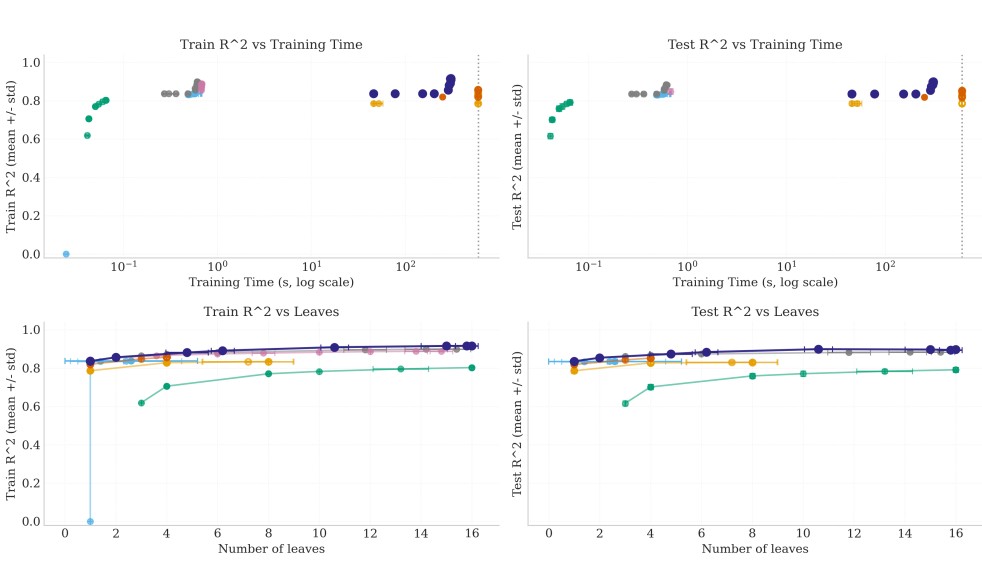

*Figure 17.* Performance of CLARITree and baselines on Temperature (Max) and Temperature (Min). Each trained with a maximum tree depth of 4. Results averaged over five random 80/20 splits, with error bars showing $\pm 1$ standard deviation. The dashed vertical line in the top row denotes the default time limit of 10 minutes.

# Seoul Bike: linear regression tree depth 4, threshold 20

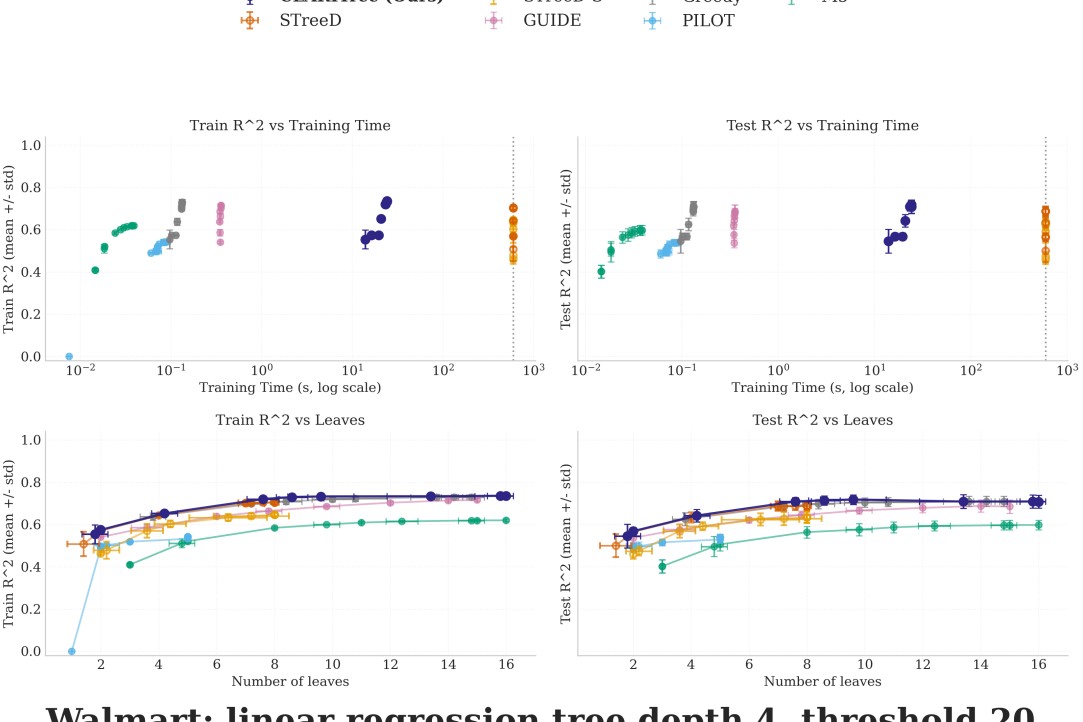

# Walmart: linear regression tree depth 4, threshold 20

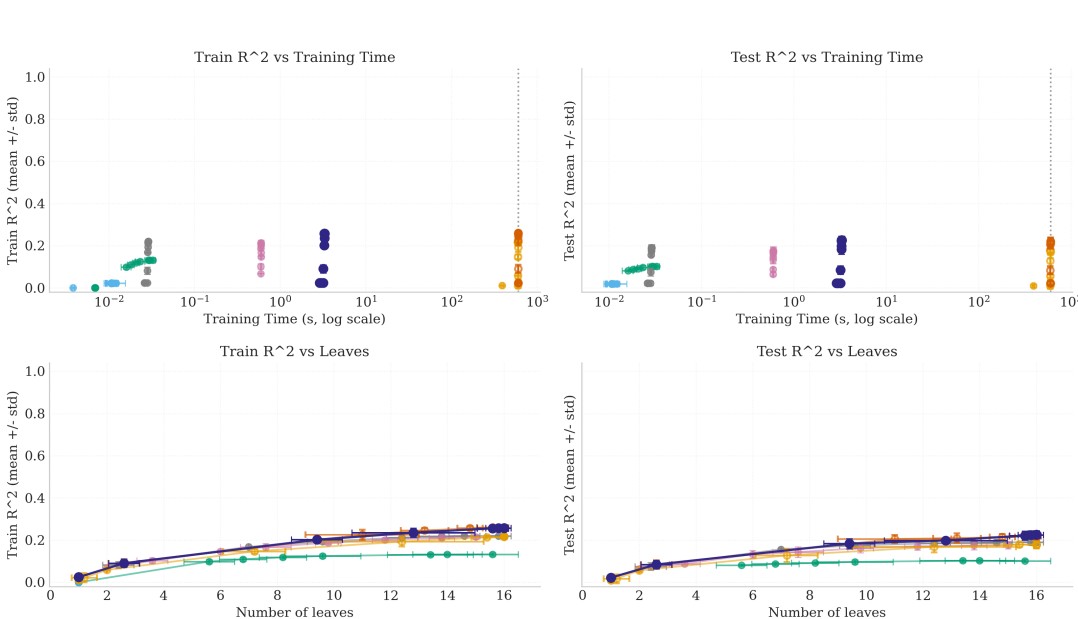

*Figure 18.* Performance of CLARITree and baselines on Seoul Bike and Walmart. Each trained with a maximum tree depth of 4. Results averaged over five random 80/20 splits, with error bars showing $\pm 1$ standard deviation. The dashed vertical line in the top row denotes the default time limit of 10 minutes.

*Table 4.* In-sample performance comparison of piecewise linear regression tree methods (Depth = 4, Thresholds = 20). We report Train $R^2$ (mean $\pm$ std) for each dataset; the corresponding Train MSE ratio (relative to CLARITree) is shown in parentheses. Best results per dataset are highlighted in bold, and the second-best results are underlined. An asterisk ($*$) indicates that the selected configuration exceeded the training time limit (600s).

| Dataset | CLARITree | STreeD | STreeD-S | GUIDE | CholeskyTree | PILOT | M5 |
|---|---|---|---|---|---|---|---|
| *Small / Medium-scale datasets* | | | | | | | |
| Airfoil | 0.93 ± 0.01 (1.00) | **0.93 ± 0.00 (1.00)** | 0.81 ± 0.00 (2.71) | 0.89 ± 0.02 (1.57) | 0.92 ± 0.01 (1.14) | 0.52 ± 0.02 (6.86) | 0.60 ± 0.01 (5.71) |
| Auction | **0.95 ± 0.00 (1.00)** | **0.95 ± 0.00 (1.00)** | 0.94 ± 0.00 (1.20) | 0.94 ± 0.00 (1.20) | 0.93 ± 0.01 (1.40) | 0.87 ± 0.01 (2.60) | 0.89 ± 0.00 (2.20) |
| Auto MPG | **0.90 ± 0.01 (1.00)** | 0.87 ± 0.01 (1.30) | 0.84 ± 0.02 (1.60) | 0.87 ± 0.01 (1.30) | 0.89 ± 0.01 (1.10) | 0.81 ± 0.01 (1.90) | **0.90 ± 0.01 (1.00)** |
| Energy (Cooling) | 0.97 ± 0.00 (1.00) | 0.97 ± 0.00 (1.00) | **0.98 ± 0.00 (0.67)** | 0.97 ± 0.00 (1.00) | 0.97 ± 0.00 (1.00) | 0.89 ± 0.01 (3.67) | 0.95 ± 0.00 (1.67) |
| Energy (Heating) | **1.00 ± 0.00** | **1.00 ± 0.00** | **1.00 ± 0.00** | **1.00 ± 0.00** | **1.00 ± 0.00** | 0.92 ± 0.00 | 0.98 ± 0.00 |
| Insurance | **0.87 ± 0.01 (1.00)** | **0.87 ± 0.01 (1.00)** | 0.86 ± 0.01 (1.08) | **0.87 ± 0.01 (1.00)** | **0.87 ± 0.01 (1.00)** | 0.84 ± 0.01 (1.23) | 0.86 ± 0.01 (1.08) |
| Optical Net | **0.97 ± 0.00 (1.00)** | 0.96 ± 0.00 (1.33) | 0.92 ± 0.01 (2.67) | 0.87 ± 0.01 (4.33) | 0.95 ± 0.00 (1.67) | 0.77 ± 0.03 (7.67) | 0.83 ± 0.02 (5.67) |
| Real Estate | **0.68 ± 0.02 (1.00)** | **0.68 ± 0.02 (1.00)** | 0.59 ± 0.02 (1.28) | 0.64 ± 0.06 (1.13) | **0.68 ± 0.02 (1.00)** | 0.57 ± 0.02 (1.34) | 0.59 ± 0.01 (1.28) |
| Servo | 0.60 ± 0.02 (1.00) | 0.61 ± 0.02 (0.97) | **0.62 ± 0.03 (0.95)** | **0.62 ± 0.03 (0.95)** | 0.60 ± 0.02 (1.00) | 0.59 ± 0.02 (1.02) | 0.59 ± 0.02 (1.02) |
| Synch | **1.00 ± 0.00** | **1.00 ± 0.00** | **1.00 ± 0.00** | **1.00 ± 0.00** | **1.00 ± 0.00** | **1.00 ± 0.00** | **1.00 ± 0.00** |
| Yacht | **1.00 ± 0.00** | **1.00 ± 0.00** | **1.00 ± 0.00** | **1.00 ± 0.00** | **1.00 ± 0.00** | 0.87 ± 0.01 | 0.99 ± 0.00 |
| *Large-scale datasets* | | | | | | | |
| California Housing | **0.77 ± 0.00 (1.00)** | 0.71 ± 0.00 (1.26)* | 0.66 ± 0.00 (1.48)* | 0.74 ± 0.00 (1.13) | 0.74 ± 0.00 (1.13) | 0.65 ± 0.00 (1.52) | 0.61 ± 0.00 (1.70) |
| Seoul Bike | **0.73 ± 0.00 (1.00)** | 0.70 ± 0.01 (1.11)* | 0.65 ± 0.01 (1.30)* | 0.72 ± 0.01 (1.04) | 0.73 ± 0.01 (1.00) | 0.54 ± 0.00 (1.70) | 0.62 ± 0.01 (1.41) |
| Temperature (Max) | **0.90 ± 0.00 (1.00)** | 0.83 ± 0.02 (1.70)* | 0.76 ± 0.00 (2.40)* | 0.86 ± 0.00 (1.40) | 0.86 ± 0.00 (1.40) | 0.79 ± 0.01 (2.10) | 0.74 ± 0.00 (2.60) |
| Temperature (Min) | **0.92 ± 0.00 (1.00)** | 0.86 ± 0.00 (1.75)* | 0.83 ± 0.00 (2.13)* | 0.85 ± 0.00 (1.88) | 0.90 ± 0.00 (1.25) | 0.84 ± 0.01 (2.00) | 0.80 ± 0.00 (2.50) |
| Walmart | **0.26 ± 0.00 (1.00)** | 0.26 ± 0.01 (1.00)* | 0.21 ± 0.01 (1.07)* | 0.21 ± 0.01 (1.07) | 0.22 ± 0.00 (1.05) | 0.02 ± 0.00 (1.32) | 0.13 ± 0.01 (1.18) |

## E.3. Tables

In this section, we report the training $R^2$ results corresponding to the test $R^2$ in Table 2 from the main paper and additionally provide both training and test $R^2$ results for experiments using full threshold enumeration.

*Table 5.* Out-of-sample performance comparison of piecewise linear regression tree methods (Depth = 4, Thresholds = 5). We report Test $R^2$ (mean $\pm$ std) for each dataset; the corresponding Test MSE ratio (relative to CLARITree) is shown in parentheses. Best results per dataset are highlighted in bold, and the second-best results are underlined. An asterisk ($*$) indicates that the selected configuration exceeded the training time limit (600s).

| Dataset | CLARITree | STreeD | STreeD-S | GUIDE | CholeskyTree | PILOT |
|---|---|---|---|---|---|---|
| *Small / Medium-scale datasets* | | | | | | |
| Airfoil | **0.89 ± 0.01 (1.00)** | **0.89 ± 0.01 (1.00)** | 0.74 ± 0.01 (2.36) | 0.84 ± 0.04 (1.45) | **0.89 ± 0.01 (1.00)** | 0.47 ± 0.04 (4.82) |
| Auction | 0.92 ± 0.03 (1.00) | 0.92 ± 0.03 (1.00) | 0.92 ± 0.03 (1.00) | **0.93 ± 0.01 (0.87)** | 0.92 ± 0.03 (1.00) | 0.86 ± 0.03 (1.75) |
| Auto MPG | **0.85 ± 0.03 (1.00)** | 0.85 ± 0.04 (1.00) | 0.82 ± 0.06 (1.20) | 0.84 ± 0.04 (1.07) | 0.85 ± 0.04 (1.00) | 0.80 ± 0.03 (1.33) |
| Energy (Cooling) | 0.97 ± 0.01 (1.00) | 0.97 ± 0.00 (1.00) | 0.97 ± 0.01 (1.00) | -2.1e2 ± 4.4e2 (7.2e3) | 0.97 ± 0.00 (1.00) | 0.89 ± 0.02 (3.67) |
| Energy (Heating) | **1.00 ± 0.00** | **1.00 ± 0.00** | **1.00 ± 0.00** | **1.00 ± 0.00** | **1.00 ± 0.00** | 0.92 ± 0.02 |
| Insurance | **0.86 ± 0.03 (1.00)** | **0.86 ± 0.03 (1.00)** | 0.85 ± 0.04 (1.07) | **0.86 ± 0.03 (1.00)** | **0.86 ± 0.03 (1.00)** | 0.83 ± 0.04 (1.21) |
| Optical Net | 0.90 ± 0.02 (1.00) | 0.90 ± 0.03 (1.00) | 0.83 ± 0.03 (1.70) | 0.80 ± 0.10 (2.00) | **0.91 ± 0.02 (0.90)** | 0.71 ± 0.17 (2.90) |
| Real Estate | **0.65 ± 0.09 (1.00)** | **0.65 ± 0.09 (1.00)** | 0.56 ± 0.10 (1.26) | 0.62 ± 0.05 (1.09) | **0.65 ± 0.09 (1.00)** | 0.55 ± 0.07 (1.29) |
| Servo | 0.51 ± 0.12 (1.00) | 0.51 ± 0.14 (1.00) | 0.48 ± 0.20 (1.06) | 0.49 ± 0.20 (1.04) | 0.51 ± 0.12 (1.00) | **0.52 ± 0.11 (0.98)** |
| Synch | **1.00 ± 0.00** | **1.00 ± 0.00** | **1.00 ± 0.00** | **1.00 ± 0.00** | **1.00 ± 0.00** | **1.00 ± 0.00** |
| Yacht | **1.00 ± 0.00** | **1.00 ± 0.00** | 0.99 ± 0.00 | 0.99 ± 0.00 | **1.00 ± 0.00** | 0.70 ± 0.05 |
| *Large-scale datasets* | | | | | | |
| California Housing | **0.75 ± 0.01 (1.00)** | 0.73 ± 0.01 (1.08)* | 0.67 ± 0.01 (1.32)* | 0.73 ± 0.02 (1.08) | 0.72 ± 0.01 (1.12) | 0.65 ± 0.01 (1.40) |
| Seoul Bike | **0.71 ± 0.02 (1.00)** | 0.70 ± 0.03 (1.03) | 0.67 ± 0.03 (1.14) | 0.67 ± 0.02 (1.14) | 0.70 ± 0.03 (1.03) | 0.54 ± 0.01 (1.59) |
| Temperature (Max) | **0.87 ± 0.02 (1.00)** | 0.83 ± 0.02 (1.31)* | 0.76 ± 0.01 (1.85)* | 0.83 ± 0.01 (1.31) | 0.84 ± 0.01 (1.23) | 0.78 ± 0.01 (1.69) |
| Temperature (Min) | **0.90 ± 0.01 (1.00)** | 0.87 ± 0.01 (1.30)* | 0.85 ± 0.01 (1.50)* | 0.85 ± 0.01 (1.50) | 0.88 ± 0.00 (1.20) | 0.83 ± 0.01 (1.70) |
| Walmart | 0.17 ± 0.02 (1.00) | **0.18 ± 0.02 (0.99)** | 0.13 ± 0.02 (1.05) | 0.16 ± 0.02 (1.01) | 0.16 ± 0.02 (1.01) | 0.02 ± 0.01 (1.18) |

*Table 6.* In-sample performance comparison of piecewise linear regression tree methods (Depth = 4, Thresholds = 5). We report Train $R^2$ (mean $\pm$ std) for each dataset; the corresponding Train MSE ratio (relative to CLARITree) is shown in parentheses. Best results per dataset are highlighted in bold, and the second-best results are underlined. An asterisk ($*$) indicates that the selected configuration exceeded the training time limit (600s).

| Dataset | CLARITree | STreeD | STreeD-S | GUIDE | CholeskyTree | PILOT |
|---|---|---|---|---|---|---|
| *Small / Medium-scale datasets* | | | | | | |
| Airfoil | **0.92 ± 0.00 (1.00)** | 0.92 ± 0.01 (1.00) | 0.79 ± 0.00 (2.63) | 0.89 ± 0.02 (1.38) | 0.91 ± 0.01 (1.13) | 0.49 ± 0.02 (6.38) |
| Auction | 0.93 ± 0.00 (1.00) | 0.93 ± 0.01 (1.00) | 0.93 ± 0.01 (1.00) | **0.94 ± 0.00 (0.86)** | 0.93 ± 0.01 (1.00) | 0.87 ± 0.01 (1.86) |
| Auto MPG | 0.89 ± 0.01 (1.00) | **0.91 ± 0.01 (0.82)** | 0.90 ± 0.01 (0.91) | 0.87 ± 0.01 (1.18) | 0.88 ± 0.02 (1.09) | 0.81 ± 0.01 (1.73) |
| Energy (Cooling) | 0.97 ± 0.00 (1.00) | **0.98 ± 0.00 (0.67)** | **0.98 ± 0.00 (0.67)** | 0.97 ± 0.00 (1.00) | 0.97 ± 0.00 (1.00) | 0.89 ± 0.01 (3.67) |
| Energy (Heating) | **1.00 ± 0.00** | **1.00 ± 0.00** | **1.00 ± 0.00** | **1.00 ± 0.00** | **1.00 ± 0.00** | 0.92 ± 0.00 |
| Insurance | 0.86 ± 0.01 (1.00) | 0.86 ± 0.01 (1.00) | 0.85 ± 0.01 (1.07) | **0.87 ± 0.01 (0.93)** | **0.87 ± 0.01 (0.93)** | 0.84 ± 0.01 (1.14) |
| Optical Net | **0.95 ± 0.00 (1.00)** | 0.95 ± 0.01 (1.00) | 0.91 ± 0.01 (1.80) | 0.87 ± 0.01 (2.60) | **0.95 ± 0.00 (1.00)** | 0.77 ± 0.03 (4.60) |
| Real Estate | **0.68 ± 0.02 (1.00)** | **0.68 ± 0.02 (1.00)** | 0.59 ± 0.02 (1.28) | 0.65 ± 0.06 (1.09) | **0.68 ± 0.02 (1.00)** | 0.57 ± 0.02 (1.34) |
| Servo | 0.60 ± 0.02 (1.00) | 0.61 ± 0.02 (0.97) | **0.62 ± 0.03 (0.95)** | **0.62 ± 0.03 (0.95)** | 0.60 ± 0.02 (1.00) | 0.59 ± 0.02 (1.02) |
| Synch | **1.00 ± 0.00** | **1.00 ± 0.00** | **1.00 ± 0.00** | **1.00 ± 0.00** | **1.00 ± 0.00** | **1.00 ± 0.00** |
| Yacht | **1.00 ± 0.00** | **1.00 ± 0.00** | **1.00 ± 0.00** | **1.00 ± 0.00** | **1.00 ± 0.00** | 0.70 ± 0.08 |
| *Large-scale datasets* | | | | | | |
| California Housing | **0.76 ± 0.00 (1.00)** | 0.74 ± 0.00 (1.08)* | 0.67 ± 0.00 (1.37)* | 0.74 ± 0.01 (1.08) | 0.73 ± 0.00 (1.13) | 0.65 ± 0.00 (1.46) |
| Seoul Bike | **0.73 ± 0.01 (1.00)** | **0.73 ± 0.01 (1.00)** | 0.69 ± 0.02 (1.15) | 0.70 ± 0.01 (1.11) | **0.73 ± 0.01 (1.00)** | 0.54 ± 0.00 (1.70) |
| Temperature (Max) | **0.90 ± 0.00 (1.00)** | 0.85 ± 0.00 (1.50)* | 0.77 ± 0.00 (2.30)* | 0.84 ± 0.00 (1.60) | 0.87 ± 0.01 (1.30) | 0.79 ± 0.00 (2.10) |
| Temperature (Min) | **0.91 ± 0.00 (1.00)** | 0.88 ± 0.00 (1.33)* | 0.85 ± 0.00 (1.67)* | 0.85 ± 0.00 (1.67) | 0.89 ± 0.00 (1.22) | 0.83 ± 0.00 (1.89) |
| Walmart | 0.21 ± 0.00 (1.00) | **0.22 ± 0.00 (0.99)** | 0.16 ± 0.00 (1.06) | 0.19 ± 0.00 (1.03) | 0.18 ± 0.02 (1.04) | 0.02 ± 0.00 (1.24) |

*Table 7.* Out-of-sample performance comparison of piecewise linear regression tree methods (Depth = 4, Thresholds = Full). We report Test $R^2$ (mean ± std) for each dataset; the corresponding Test MSE ratio (relative to CLARITree) is shown in parentheses. Best results per dataset are highlighted in bold, and the second-best results are underlined. An asterisk (∗) indicates that the selected configuration exceeded the training time limit (600s).

| Dataset | CLARITree | STreeD | STreeD-S | GUIDE | CholeskyTree | PILOT |
|---|---|---|---|---|---|---|
| *Small / Medium-scale datasets* | | | | | | |
| Airfoil | **0.89 ± 0.01 (1.00)** | 0.89 ± 0.02 (1.00)* | 0.76 ± 0.03 (2.18)* | 0.84 ± 0.04 (1.45) | 0.87 ± 0.02 (1.18) | 0.70 ± 0.03 (2.73) |
| Auction | **0.95 ± 0.01 (1.00)** | 0.94 ± 0.02 (1.20) | **0.95 ± 0.01 (1.00)** | 0.93 ± 0.01 (1.40) | **0.95 ± 0.01 (1.00)** | 0.88 ± 0.02 (2.40) |
| Auto MPG | 0.84 ± 0.03 (1.00) | **0.86 ± 0.04 (0.87)*** | 0.82 ± 0.06 (1.12)* | 0.84 ± 0.03 (1.00) | 0.84 ± 0.03 (1.00) | 0.85 ± 0.02 (0.94) |
| Energy (Cooling) | 0.97 ± 0.01 (1.00) | **0.97 ± 0.00 (1.00)** | 0.97 ± 0.01 (1.00) | 0.97 ± 0.01 (1.00) | **0.97 ± 0.00 (1.00)** | 0.95 ± 0.01 (1.67) |
| Energy (Heating) | **1.00 ± 0.00** | **1.00 ± 0.00** | **1.00 ± 0.00** | **1.00 ± 0.00** | **1.00 ± 0.00** | 0.96 ± 0.01 |
| Insurance | **0.86 ± 0.03 (1.00)** | **0.86 ± 0.03 (1.00)*** | 0.85 ± 0.04 (1.07)* | **0.86 ± 0.03 (1.00)** | **0.86 ± 0.03 (1.00)** | **0.86 ± 0.03 (1.00)** |
| Optical Net | **0.93 ± 0.03 (1.00)** | 0.89 ± 0.04 (1.57)* | 0.79 ± 0.08 (3.00)* | 0.80 ± 0.10 (2.86) | 0.91 ± 0.04 (1.29) | 0.86 ± 0.03 (2.00) |
| Real Estate | 0.62 ± 0.08 (1.00) | 0.62 ± 0.00 (1.00)* | 0.52 ± 0.10 (1.26)* | 0.61 ± 0.06 (1.03) | **0.65 ± 0.09 (0.92)** | **0.65 ± 0.09 (0.92)** |
| Servo | 0.51 ± 0.12 (1.00) | 0.51 ± 0.14 (1.00) | 0.48 ± 0.20 (1.06) | 0.49 ± 0.20 (1.04) | 0.51 ± 0.12 (1.00) | **0.52 ± 0.10 (0.98)** |
| Synch | **1.00 ± 0.00** | **1.00 ± 0.00*** | **1.00 ± 0.00** | **1.00 ± 0.00** | **1.00 ± 0.00** | **1.00 ± 0.00** |
| Yacht | **1.00 ± 0.00** | **1.00 ± 0.00** | 0.99 ± 0.01 | **1.00 ± 0.00** | **1.00 ± 0.00** | 0.99 ± 0.01 |

*Table 8.* In-sample performance comparison of piecewise linear regression tree methods (Depth = 4, Thresholds = Full). We report Train $R^2$ (mean ± std) for each dataset; the corresponding Train MSE ratio (relative to CLARITree) is shown in parentheses. Best results per dataset are highlighted in bold, and the second-best results are underlined. An asterisk (∗) indicates that the selected configuration exceeded the training time limit (600s).

| Dataset | CLARITree | STreeD | STreeD-S | GUIDE | CholeskyTree | PILOT |
|---|---|---|---|---|---|---|
| *Small / Medium-scale datasets* | | | | | | |
| Airfoil | **0.93 ± 0.00 (1.00)** | **0.93 ± 0.00 (1.00)*** | 0.81 ± 0.00 (2.71)* | 0.88 ± 0.01 (1.71) | 0.92 ± 0.01 (1.14) | 0.73 ± 0.02 (3.86) |
| Auction | 0.95 ± 0.01 (1.00) | **0.96 ± 0.00 (0.80)** | **0.96 ± 0.00 (0.80)** | 0.94 ± 0.00 (1.20) | 0.95 ± 0.01 (1.00) | 0.90 ± 0.01 (2.00) |
| Auto MPG | 0.90 ± 0.01 (1.00) | 0.87 ± 0.01 (1.30)* | **0.92 ± 0.00 (0.80)*** | 0.88 ± 0.01 (1.20) | 0.91 ± 0.01 (0.90) | 0.87 ± 0.01 (1.30) |
| Energy (Cooling) | 0.97 ± 0.00 (1.00) | 0.97 ± 0.00 (1.00) | **0.98 ± 0.00 (0.67)** | 0.97 ± 0.00 (1.00) | 0.97 ± 0.00 (1.00) | 0.95 ± 0.00 (1.67) |
| Energy (Heating) | **1.00 ± 0.00** | **1.00 ± 0.00** | **1.00 ± 0.00** | **1.00 ± 0.00** | **1.00 ± 0.00** | 0.96 ± 0.00 |
| Insurance | **0.87 ± 0.01 (1.00)** | **0.87 ± 0.01 (1.00)*** | 0.86 ± 0.01 (1.08)* | **0.87 ± 0.01 (1.00)** | **0.87 ± 0.01 (1.00)** | **0.87 ± 0.01 (1.00)** |
| Optical Net | **0.97 ± 0.00 (1.00)** | 0.95 ± 0.01 (1.67)* | 0.89 ± 0.01 (3.67)* | 0.88 ± 0.01 (4.00) | 0.95 ± 0.00 (1.67) | 0.92 ± 0.00 (2.67) |
| Real Estate | **0.73 ± 0.01 (1.00)** | 0.69 ± 0.00 (1.15)* | 0.60 ± 0.02 (1.48)* | 0.64 ± 0.06 (1.33) | 0.69 ± 0.02 (1.15) | 0.70 ± 0.02 (1.11) |
| Servo | 0.60 ± 0.02 (1.00) | 0.61 ± 0.02 (0.97) | **0.62 ± 0.03 (0.95)** | **0.62 ± 0.03 (0.95)** | 0.60 ± 0.02 (1.00) | 0.60 ± 0.02 (1.00) |
| Synch | **1.00 ± 0.00** | **1.00 ± 0.00*** | **1.00 ± 0.00** | **1.00 ± 0.00** | **1.00 ± 0.00** | **1.00 ± 0.00** |
| Yacht | **1.00 ± 0.00** | **1.00 ± 0.00** | **1.00 ± 0.00** | **1.00 ± 0.00** | **1.00 ± 0.00** | 0.99 ± 0.00 |

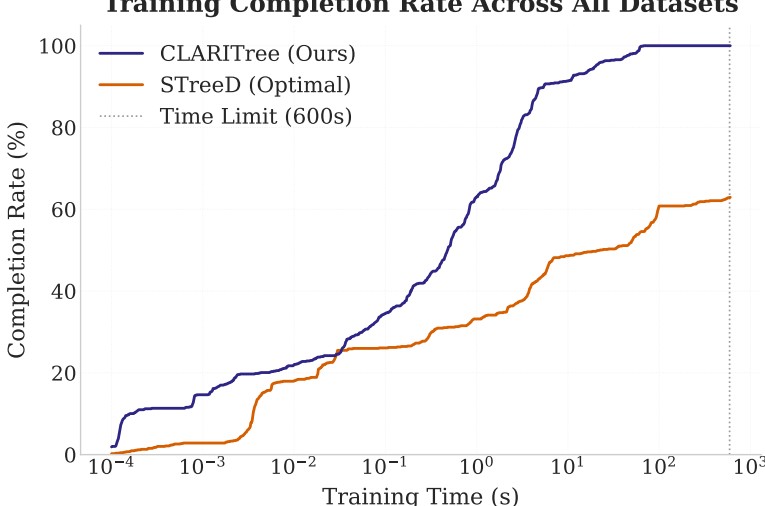

*Figure 19.* **Training completion rate under a 10-minute budget.** Empirical completion curves aggregated across all small/medium-scale datasets. The dashed vertical line marks the 600 s time limit used in our default protocol.

### E.4. Completion Ratio

In this section, we report the completion ratio for methods using full thresholds, focusing on small- and medium-scale datasets. Even in this regime, we observe that STreeD completes only about 60% of the runs, indicating substantial computational limitations when full threshold enumeration is employed.

# F. Special Case: Constant-Leaf Variant of CLARITree

In this section, we study a special case of CLARITree in which each leaf is an intercept-only regressor (a constant-leaf tree). Throughout this section, we refer to this variant as CLARITreeConst. We present the full algorithm and provide a theoretical runtime analysis, along with extensive experimental results demonstrating that CLARITreeConst remains competitive in the constant-leaf regime. Finally, we provide insights into extending CLARITreeConst to multi-step lookahead, following the general paradigm of (Babbar et al., 2025).

## F.1. Framework

In a constant regression tree, each leaf predicts a constant. For **constant leaves**, the optimal value minimizing the squared loss is given by the sample mean:

$$\widehat{y}^{(t)} = \arg\min_c \sum_{(\boldsymbol{x}_i, y_i) \in t} (y_i - c)^2 = \frac{1}{|t|} \sum_{(\boldsymbol{x}_i, y_i) \in t} y_i. \tag{6}$$

This corresponds to the intercept-only special case of our leaf model, hence no coefficient regularization is needed.

## F.2. Full Algorithm

We present the full algorithm for CLARITREECONST. Its overall structure is identical to that of Algorithm 5; the only difference is that we no longer maintain a Gram matrix, so rank-one Cholesky updates are no longer necessary. The complete algorithm is summarized in Algorithm 15. Most subroutines are shared with CLARITree and are provided in Appendix C.

## F.3. Theoretical Analysis

We also provide the runtime complexity of CLARITreeConst. After a one-time global presort of all $k$ features in $\mathcal{O}(kn \log n)$, evaluating every threshold of a feature costs $\mathcal{O}(n)$ by prefix updates. Hence:

**Theorem F.1** (Runtime of CLARITreeConst). *Including a one-time presort of all features, the runtime is*

$$\mathcal{O}\left(kn \log n + d^2 n k^2 T\right).$$

We also show that there exist data distributions for which the ratio of the MSE between Greedy and CLARITreeConst can be made arbitrarily large.

**Theorem F.2** (Arbitrary MSE Gap between CLARITreeConst and Greedy). *For every depth budget $d \geq 2$, there exist data distributions such that*

$$\frac{\text{MSE}_{Greedy}}{\text{MSE}_{CLARITreeConst}} \geq \frac{1}{2\varepsilon}.$$

---

**Algorithm 15** `CLARITreeConst`: Constant-Leaf Variant of CLARITree

---

**Require:** node data $D_{\mathcal{I}} = \{(\boldsymbol{x}_i, y_i)\}_{i \in \mathcal{I}}$; node-wise sorted index lists $\Pi_{\mathcal{I}} = \{\pi_f(\mathcal{I})\}_{f \in \mathcal{F}}$; feature set $\mathcal{F}$; leaf penalty $\lambda$; maximum depth $d$; precomputed global sorted index lists $\pi_f$ for each $f \in \mathcal{F}$

**Ensure:** constant-regressor tree $T^*$ and its objective OBJ

1: $\widehat{y} \leftarrow \texttt{Mean}(\boldsymbol{y}_{\mathcal{I}})$
2: $\text{OBJ}_{init} \leftarrow \texttt{MSELoss}(\boldsymbol{y}_{\mathcal{I}}, \widehat{y}) + \lambda$
3: **if** $d = 0$ **then**
4:     **return** $\texttt{Leaf}(\widehat{y}, \text{OBJ}_{init}),\ \text{OBJ}_{init}$
5: **end if**
6: $\text{OBJ}^* \leftarrow \infty$
7: $f^*, \tau^* \leftarrow \texttt{None}$
8: **for** each feature $f \in \mathcal{F}$ **do**
9:     **for** each threshold $\tau$ in unique values of $f$ (consistent with $\pi_f(\mathcal{I})$) **do**
10:         $\mathcal{I}_L, \mathcal{I}_R \leftarrow \texttt{PartitionIndices}(\Pi_{\mathcal{I}}, f, \tau)$                   ▷ $\mathcal{I}_L$ obtained from sorted indices; $\mathcal{I}_R = \mathcal{I} \setminus \mathcal{I}_L$
11:         $\Pi_{\mathcal{I}_L}, \Pi_{\mathcal{I}_R} \leftarrow \texttt{SplitSortedIndices}(\Pi_{\mathcal{I}}, \mathcal{I}_L)$
12:         $D_{\mathcal{I}_L}, D_{\mathcal{I}_R} \leftarrow \texttt{RestrictData}(D_{\mathcal{I}}, \mathcal{I}_L)$                         ▷ views via indices
13:         $\widehat{y}_L \leftarrow \texttt{Mean}(\boldsymbol{y}_{\mathcal{I}_L})$
14:         $\widehat{y}_R \leftarrow \texttt{Mean}(\boldsymbol{y}_{\mathcal{I}_R})$
15:         $\text{OBJLEAF}_L \leftarrow \texttt{MSELoss}(\boldsymbol{y}_{\mathcal{I}_L}, \widehat{y}_L) + \lambda$
16:         $\text{OBJLEAF}_R \leftarrow \texttt{MSELoss}(\boldsymbol{y}_{\mathcal{I}_R}, \widehat{y}_R) + \lambda$
17:         $T_L, \text{OBJ}_L \leftarrow \texttt{GreedyConst}(\widehat{y}_L, D_{\mathcal{I}_L}, \Pi_{\mathcal{I}_L}, \mathcal{F}, \lambda, d, d' + 1, \pi_f, \text{OBJLEAF}_L)$
18:         $T_R, \text{OBJ}_R \leftarrow \texttt{GreedyConst}(\widehat{y}_R, D_{\mathcal{I}_R}, \Pi_{\mathcal{I}_R}, \mathcal{F}, \lambda, d - 1, \pi_f, \text{OBJLEAF}_R)$
19:         $\text{OBJ}_{split} \leftarrow \text{OBJ}_L + \text{OBJ}_R$
20:         **if** $\text{OBJ}_{split} < \text{OBJ}^*$ **then**
21:             $\text{OBJ}^* \leftarrow \text{OBJ}_{split}$
22:             $f^* \leftarrow f,\ \tau^* \leftarrow \tau$
23:             $(\Pi_L^{best}, \mathcal{I}_L^{best}) \leftarrow (\Pi_{\mathcal{I}_L}, \mathcal{I}_L)$
24:             $(\Pi_R^{best}, \mathcal{I}_R^{best}) \leftarrow (\Pi_{\mathcal{I}_R}, \mathcal{I}_R)$
25:         **end if**
26:     **end for**
27: **end for**
28: **if** $f^* \neq \texttt{None}$ **then**                                         ▷ A feasible split was selected
29:     $D_{\mathcal{I}_L^{best}}, D_{\mathcal{I}_R^{best}} \leftarrow \texttt{RestrictData}(D_{\mathcal{I}}, \mathcal{I}_L^{best})$
30:     $T_L, \text{OBJ}_L \leftarrow \texttt{CLARITreeConst}(D_{\mathcal{I}_L^{best}}, \Pi_L^{best}, \mathcal{F}, \lambda, d - 1, \pi_f)$
31:     $T_R, \text{OBJ}_R \leftarrow \texttt{CLARITreeConst}(D_{\mathcal{I}_R^{best}}, \Pi_R^{best}, \mathcal{F}, \lambda, d - 1, \pi_f)$
32:     $\text{OBJ}_{final} \leftarrow \text{OBJ}_L + \text{OBJ}_R$
33:     **if** $\text{OBJ}_{final} < \text{OBJ}_{init}$ **then**
34:         **return** $\texttt{Node}(f^*, \tau^*, T_L, T_R),\ \text{OBJ}_{final}$
35:     **else**
36:         **return** $\texttt{Leaf}(\widehat{y}, \text{OBJ}_{init}),\ \text{OBJ}_{init}$
37:     **end if**
38: **else**
39:     **return** $\texttt{Leaf}(\widehat{y}, \text{OBJ}_{init}),\ \text{OBJ}_{init}$
40: **end if**

---

### F.4. Further Comparisons With Other Methods for Constant-Leaf Regression

Here, we evaluate CLARITreeConst and the baselines on all the datasets. All datasets were evaluated across five 80-20 train-test splits, with the average and standard deviation reported. All datasets were evaluated across 20-thresholds for continuous features. For other methods, we also use their constant-regression versions for a fair comparison. Generally, we reduced the structural complexity penalty across all datasets to achieve a wider range of sparsity (from 1 to 32 leaves). All specific parameter settings are provided in the supplementary code package. We also show the Out-of-sample performance comparison for constant-regressor tree baselines at Depth = 5. We report Test $R^2$ and Train $R^2$ (mean $\pm$ std) for each dataset in Table 9, Table 10.

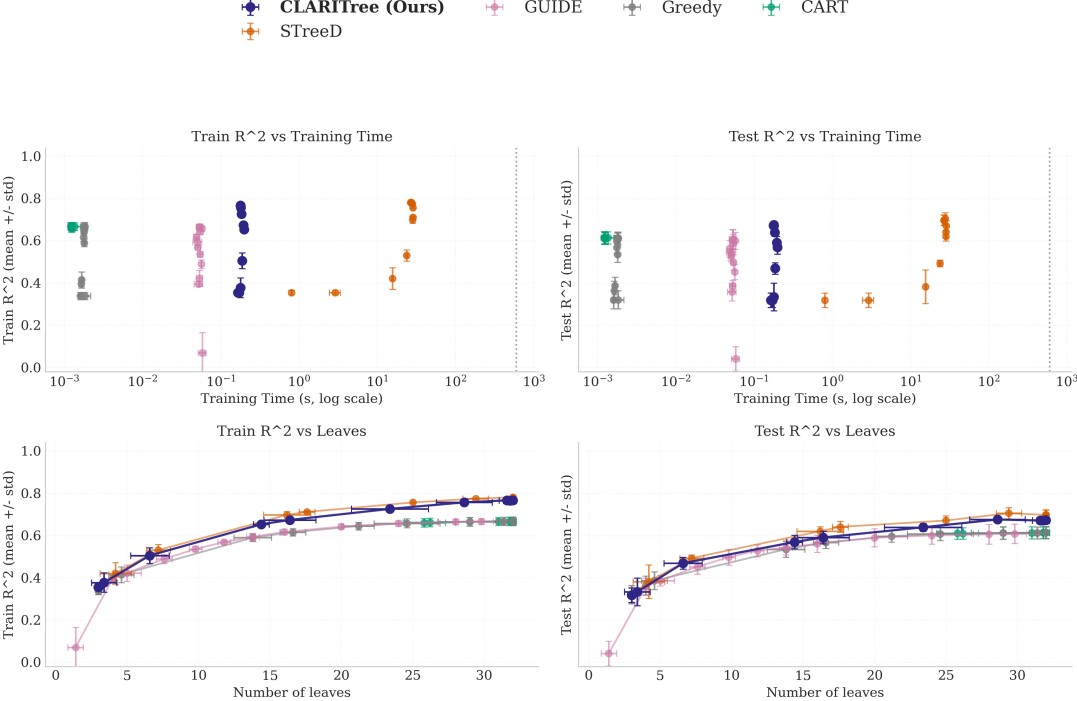

*Figure 20.* Performance of constant regression variant of CLARITree and baselines on additional datasets (part 1). Each trained with a maximum tree depth of 5. Results averaged over five random 80/20 splits, with error bars showing $\pm$ 1 standard deviation. The dashed vertical line in the top row denotes the default time limit of 10 minutes.

**Small / Medium-Scale Datasets**

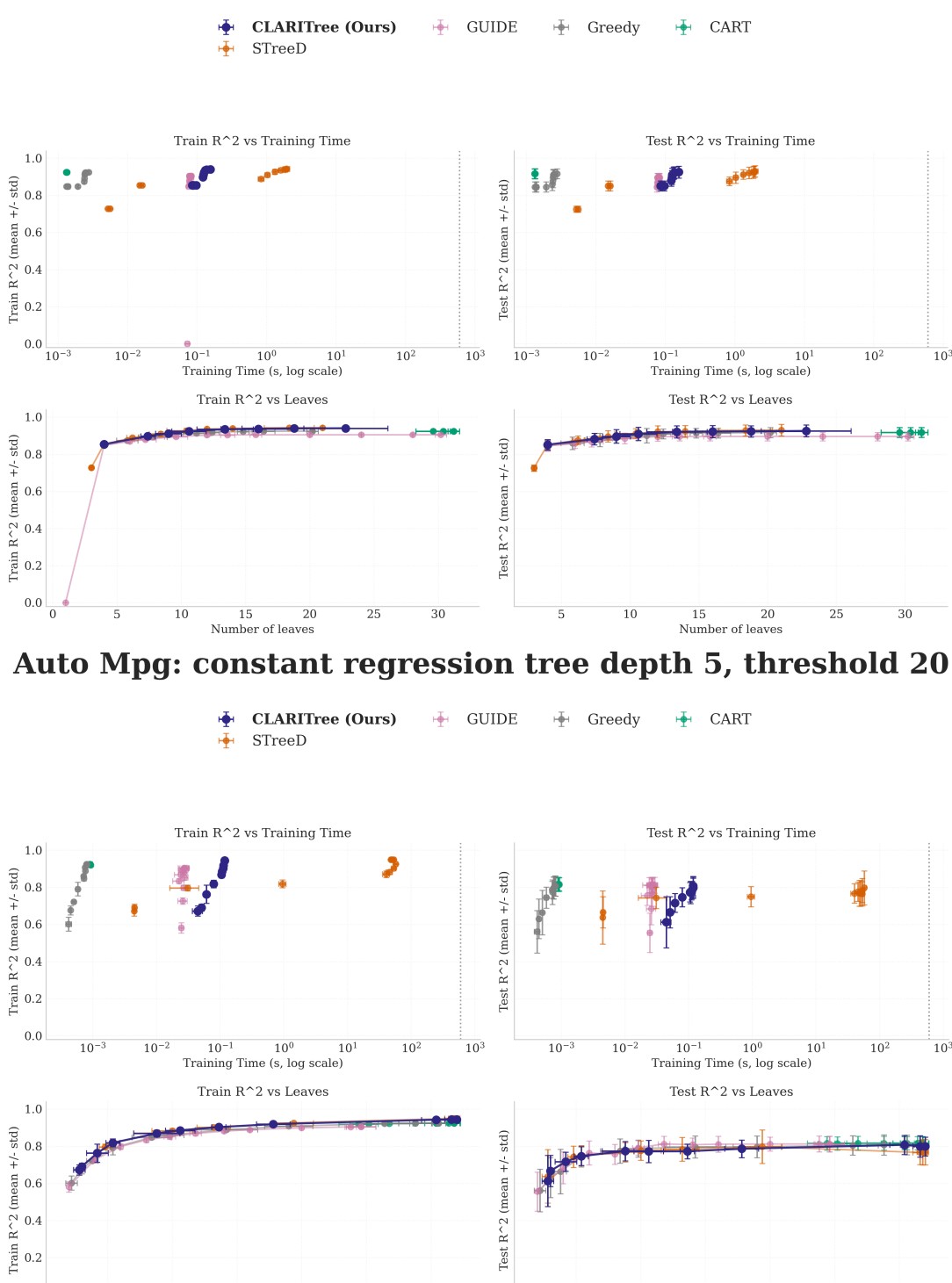

*Figure 21.* Performance of constant regression variant of CLARITree and baselines on additional datasets (part 2). Each trained with a maximum tree depth of 5. Results averaged over five random 80/20 splits, with error bars showing ± 1 standard deviation. The dashed vertical line in the top row denotes the default time limit of 10 minutes.

## Energy (Cooling): constant regression tree depth 5, threshold 20

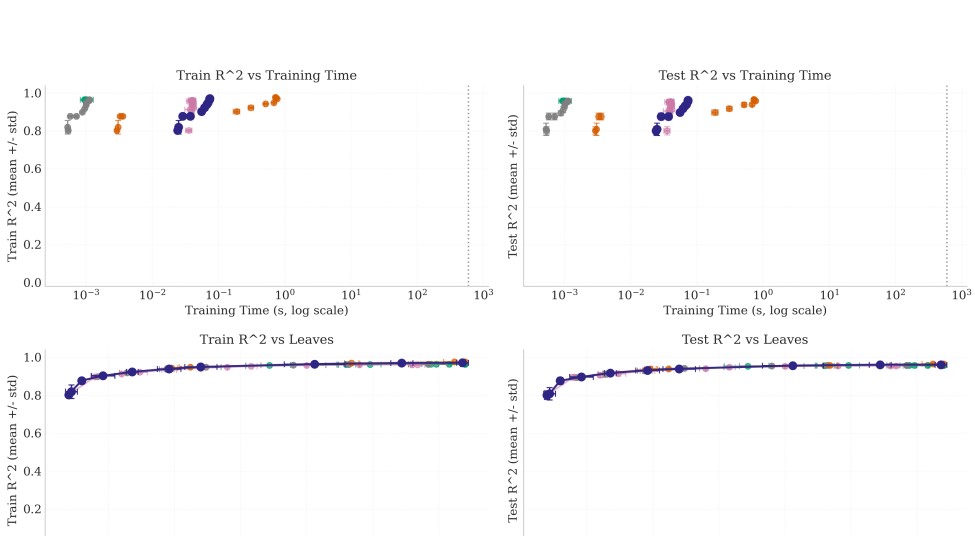

## Energy (Heating): constant regression tree depth 5, threshold 20

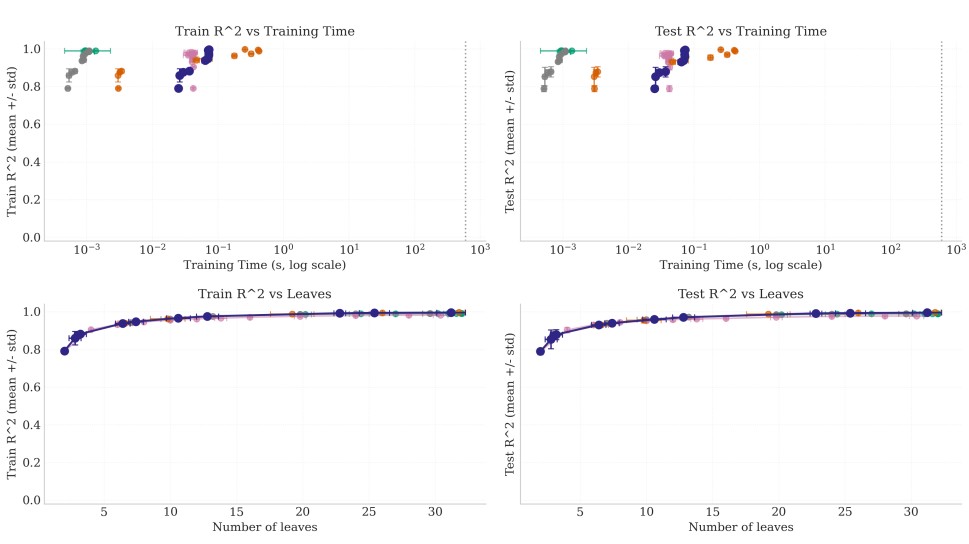

*Figure 22.* Performance of constant regression variant of CLARITree and baselines on additional datasets (part 3). Each trained with a maximum tree depth of 5. Results averaged over five random 80/20 splits, with error bars showing $\pm 1$ standard deviation. The dashed vertical line in the top row denotes the default time limit of 10 minutes.

## Insurance: constant regression tree depth 5, threshold 20

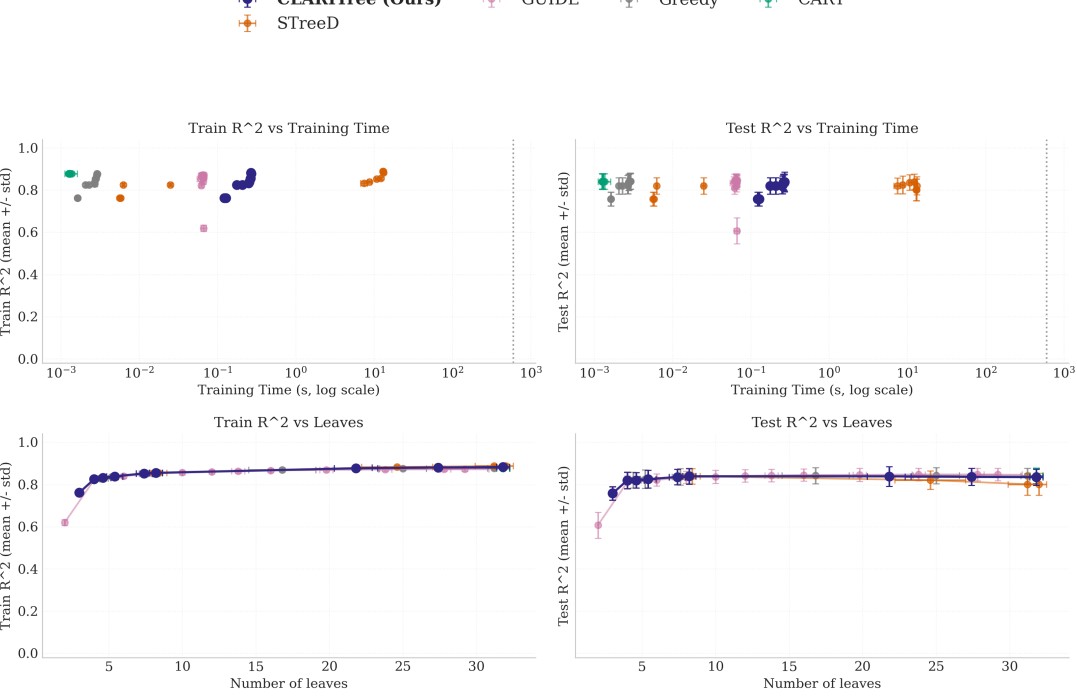

## Optical Net: constant regression tree depth 5, threshold 20

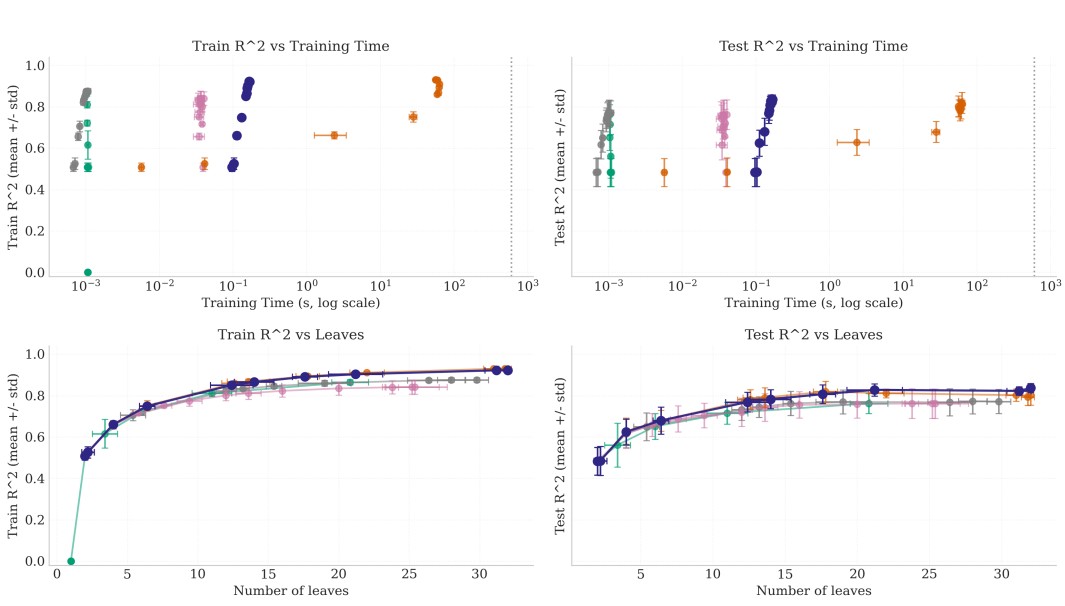

*Figure 23.* Performance of constant regression variant of CLARITree and baselines on additional datasets (part 4). Each trained with a maximum tree depth of 5. Results averaged over five random 80/20 splits, with error bars showing $\pm 1$ standard deviation. The dashed vertical line in the top row denotes the default time limit of 10 minutes.

## Real Estate: constant regression tree depth 5, threshold 20

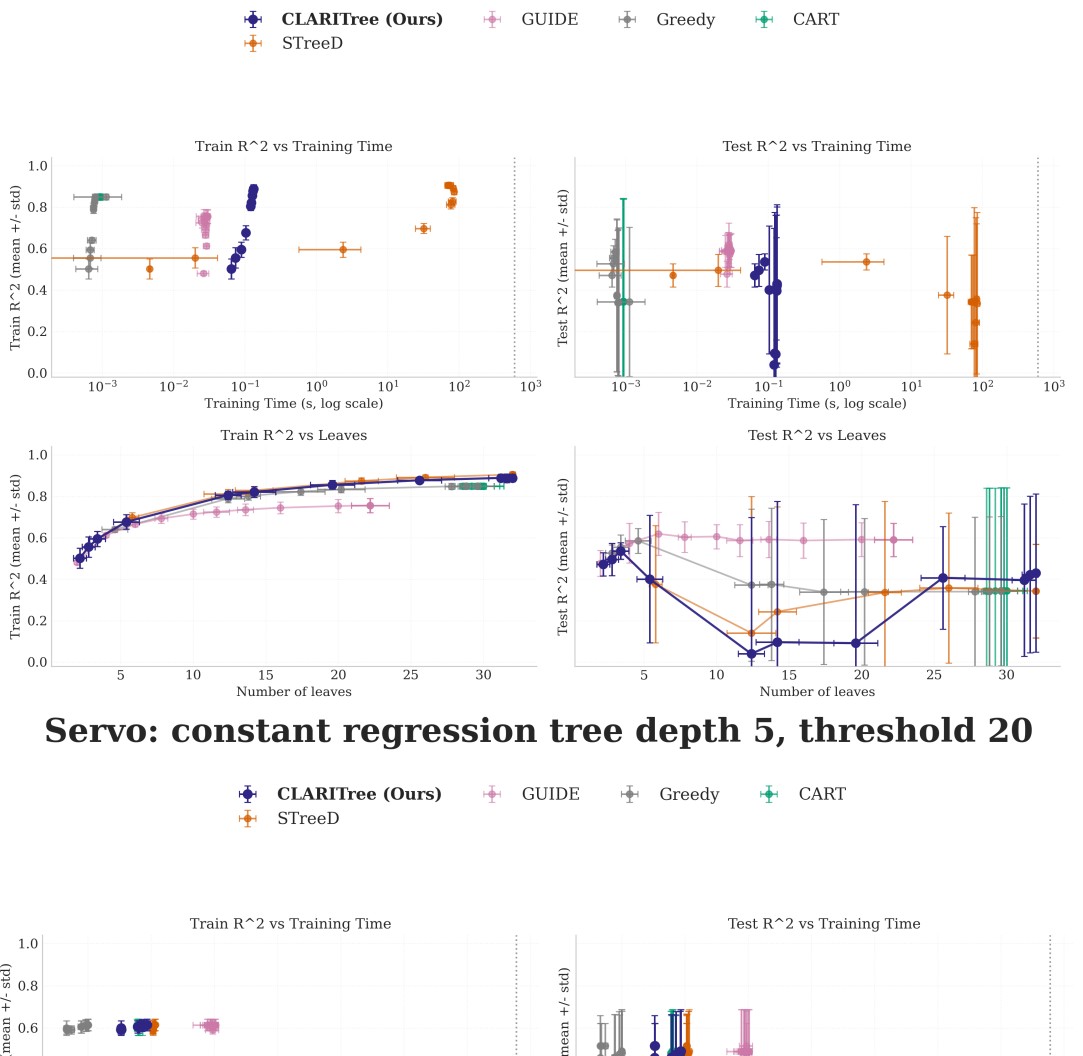

## Servo: constant regression tree depth 5, threshold 20

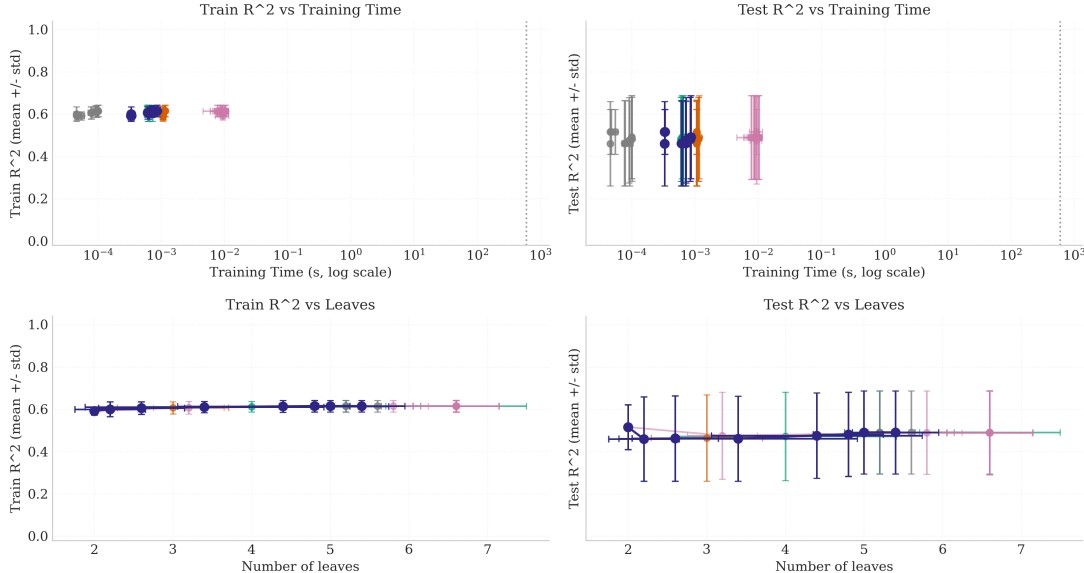

*Figure 24.* Performance of constant regression variant of CLARITree and baselines on additional datasets (part 5). Each trained with a maximum tree depth of 5. Results averaged over five random 80/20 splits, with error bars showing $\pm 1$ standard deviation. The dashed vertical line in the top row denotes the default time limit of 10 minutes.

# Synch: constant regression tree depth 5, threshold 20

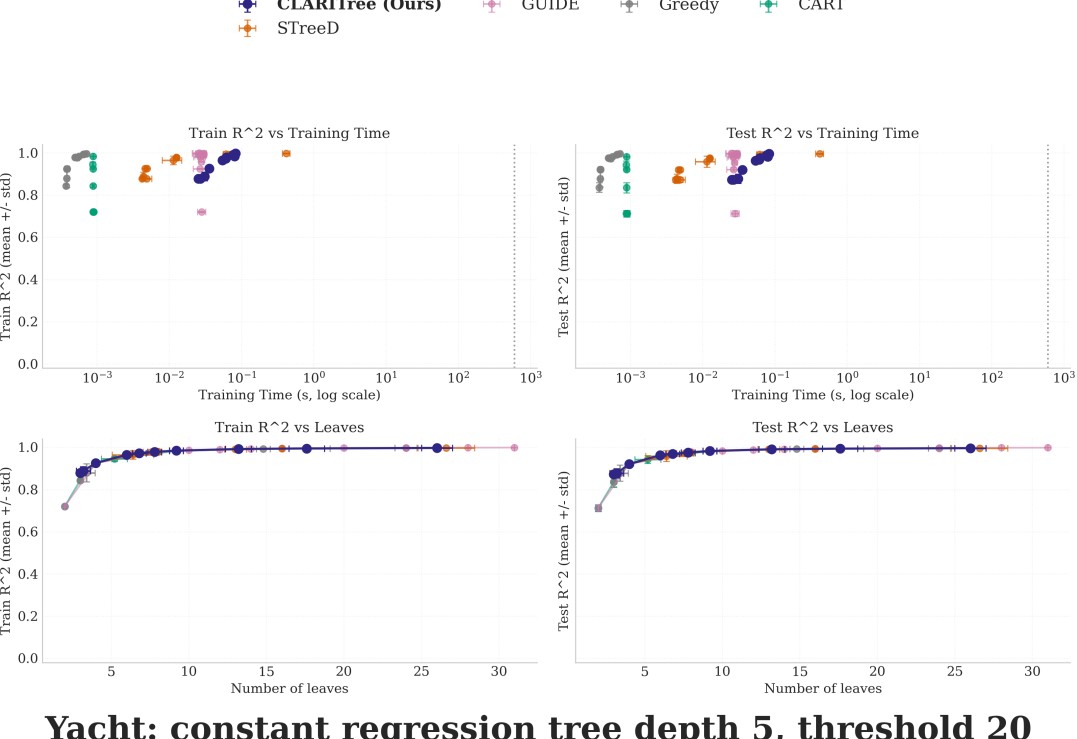

# Yacht: constant regression tree depth 5, threshold 20

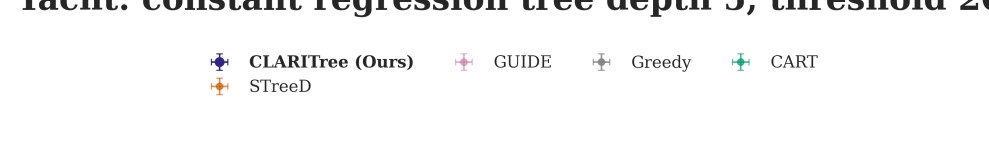

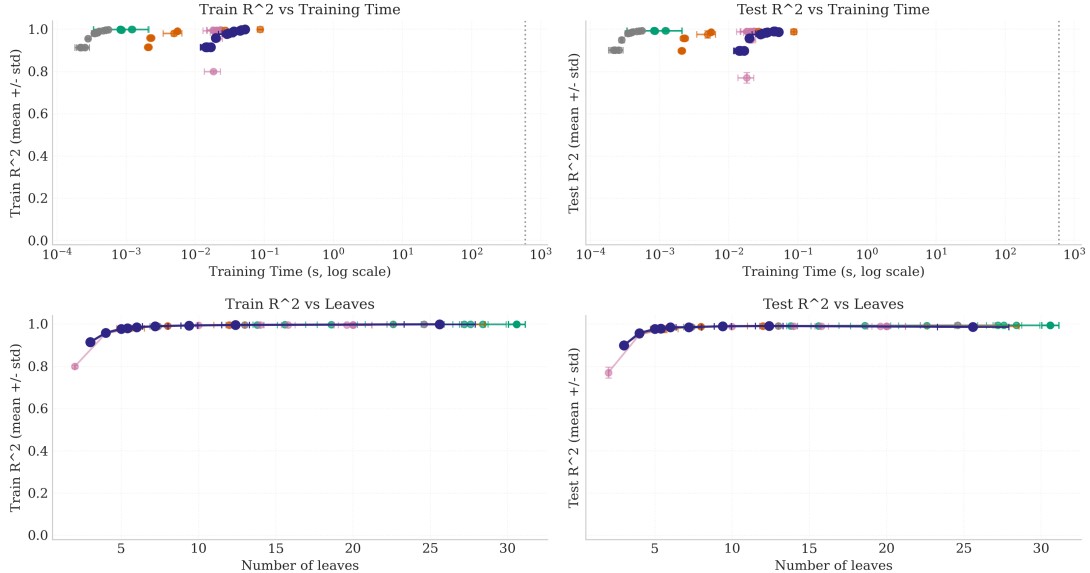

*Figure 25.* Performance of constant regression variant of CLARITree and baselines on additional datasets (part 6). Each trained with a maximum tree depth of 5. Results averaged over five random 80/20 splits, with error bars showing $\pm$ 1 standard deviation. The dashed vertical line in the top row denotes the default time limit of 10 minutes.

# Walmart: constant regression tree depth 5, threshold 20

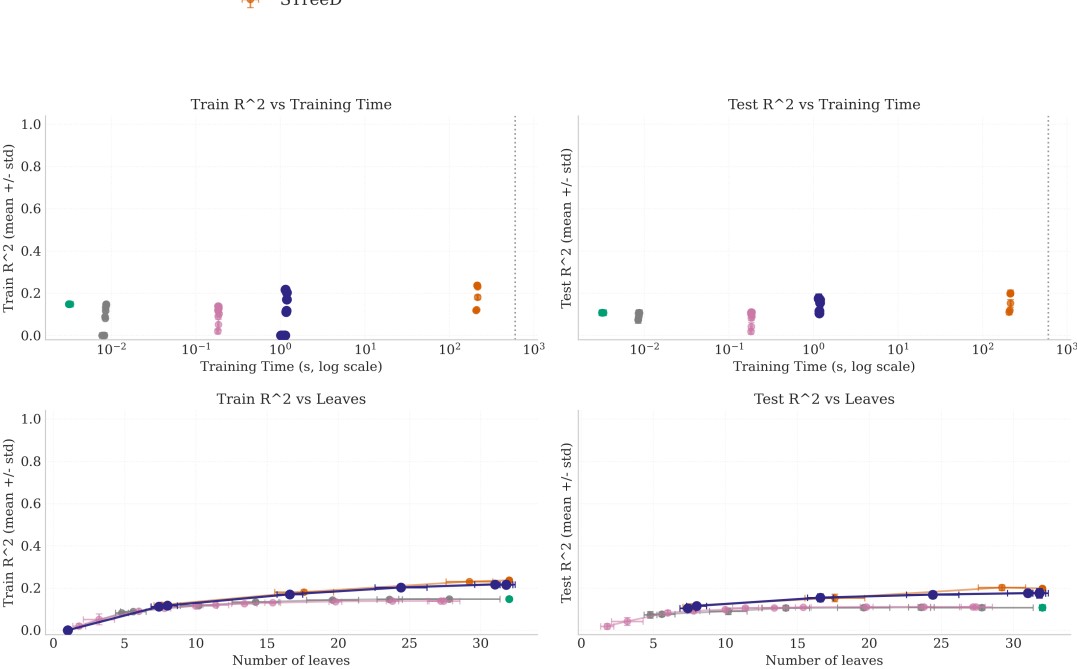

*Figure 26.* Performance of constant regression variant of CLARITree and baselines on additional datasets (part 1). Each trained with a maximum tree depth of 5. Results averaged over five random 80/20 splits, with error bars showing $\pm$ 1 standard deviation. Dashed lines and hollow markers indicate runs that hit the prescribed time limit. The dashed vertical line in the top row denotes the default time limit of 10 minutes.

**Large-Scale Datasets**

## California Housing: constant regression tree depth 5, threshold 20

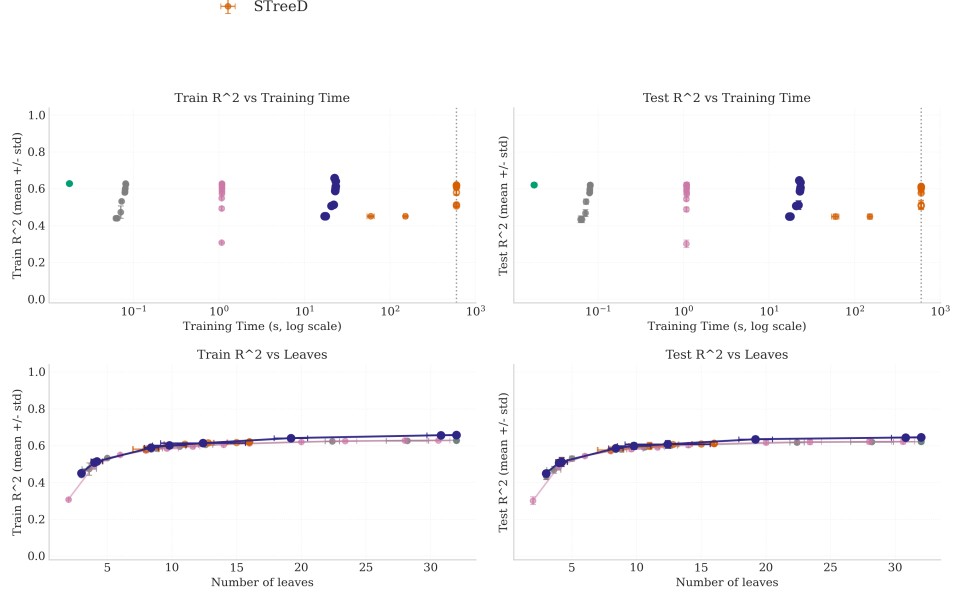

## Seoul Bike: constant regression tree depth 5, threshold 20

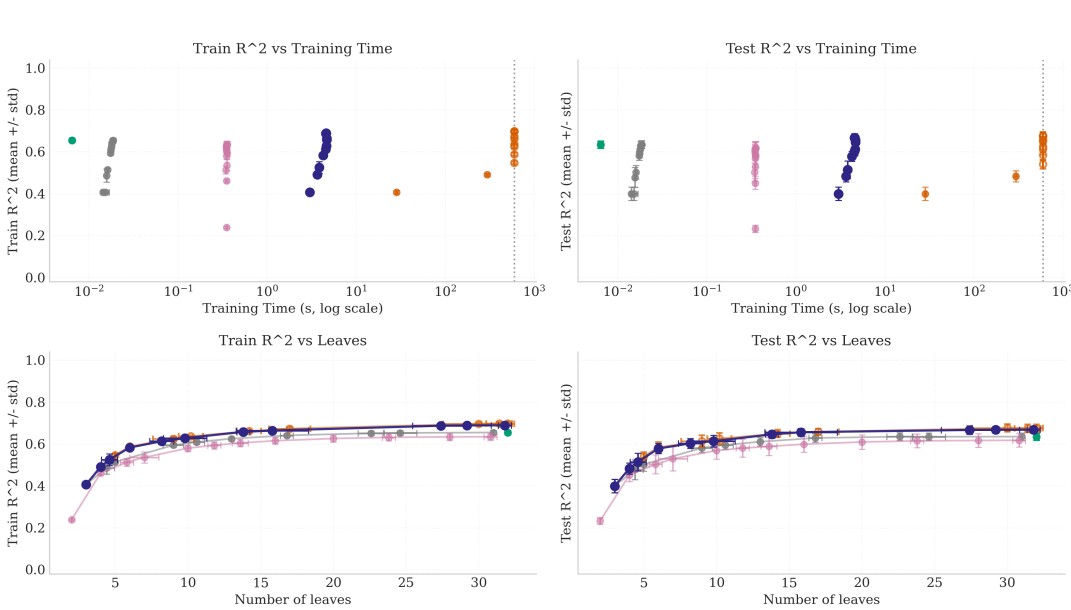

*Figure 27.* Performance of constant regression variant of CLARITree and baselines on additional datasets (part 2). Each trained with a maximum tree depth of 5. Results averaged over five random 80/20 splits, with error bars showing $\pm 1$ standard deviation. Dashed lines and hollow markers indicate runs that hit the prescribed time limit. The dashed vertical line in the top row denotes the default time limit of 10 minutes.

## Temperature (Max): constant regression tree depth 5, threshold 20

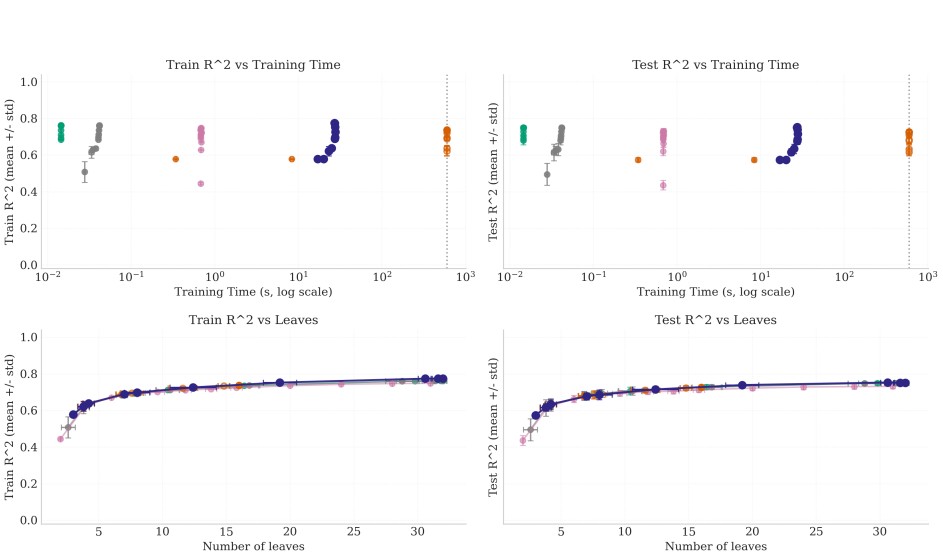

## Temperature (Min): constant regression tree depth 5, threshold 20

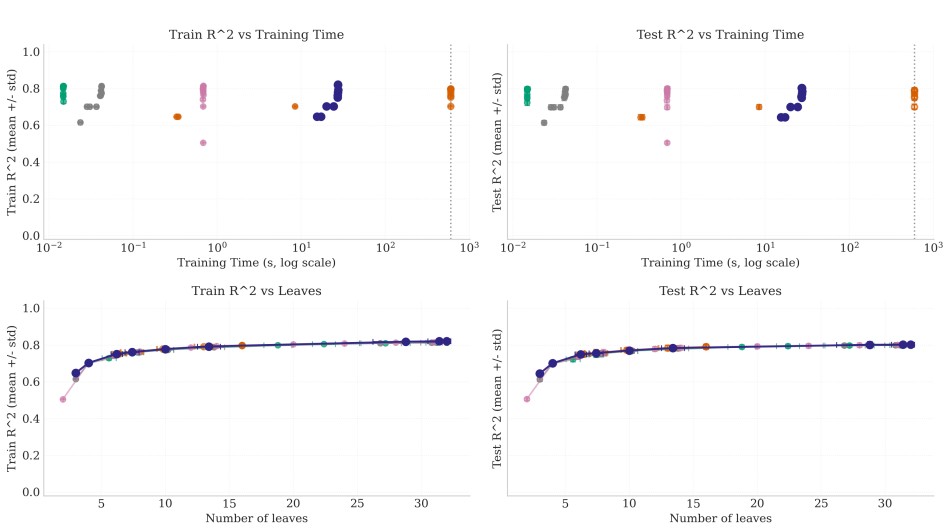

*Figure 28.* Performance of constant regression variant of CLARITree and baselines on additional datasets (part 3). Each trained with a maximum tree depth of 5. Results averaged over five random 80/20 splits, with error bars showing $\pm 1$ standard deviation. The dashed vertical line in the top row denotes the default time limit of 10 minutes.

*Table 9.* Out-of-sample performance comparison of constant regression tree methods (Depth = 5, Thresholds = 20). We report Test $R^2$ (mean $\pm$ std) for each dataset; the corresponding Test MSE ratio (relative to CLARITree) is shown in parentheses. Best results per dataset are highlighted in bold, and the second-best results are underlined. An asterisk ($*$) indicates that the selected configuration exceeded the training time limit (600s).

| Dataset | CLARITree | STreeD | GUIDE | CholeskyTree | CART |
|---|---|---|---|---|---|
| *Small / Medium-scale datasets* | | | | | |
| Airfoil | 0.67 $\pm$ 0.01 (1.00) | **0.70 $\pm$ 0.02 (0.91)** | 0.61 $\pm$ 0.05 (1.18) | 0.61 $\pm$ 0.03 (1.18) | 0.61 $\pm$ 0.03 (1.18) |
| Auction | 0.92 $\pm$ 0.03 (1.00) | **0.93 $\pm$ 0.03 (0.87)** | 0.90 $\pm$ 0.02 (1.25) | 0.92 $\pm$ 0.03 (1.00) | 0.92 $\pm$ 0.03 (1.00) |
| Auto MPG | 0.77 $\pm$ 0.04 (1.00) | 0.80 $\pm$ 0.09 (0.87) | 0.81 $\pm$ 0.04 (0.83) | 0.81 $\pm$ 0.05 (0.83) | **0.82 $\pm$ 0.04 (0.78)** |
| Energy (Cooling) | 0.96 $\pm$ 0.01 (1.00) | **0.97 $\pm$ 0.00 (0.75)** | 0.96 $\pm$ 0.01 (1.00) | 0.96 $\pm$ 0.01 (1.00) | 0.96 $\pm$ 0.01 (1.00) |
| Energy (Heating) | **1.00 $\pm$ 0.00** | 1.00 $\pm$ 0.00 | 0.98 $\pm$ 0.00 | 0.99 $\pm$ 0.00 | 0.99 $\pm$ 0.00 |
| Insurance | 0.84 $\pm$ 0.04 (1.00) | 0.84 $\pm$ 0.04 (1.00) | **0.85 $\pm$ 0.03 (0.94)** | 0.84 $\pm$ 0.04 (1.00) | 0.84 $\pm$ 0.04 (1.00) |
| Optical Net | **0.84 $\pm$ 0.02 (1.00)** | 0.81 $\pm$ 0.02 (1.19) | 0.76 $\pm$ 0.07 (1.50) | 0.77 $\pm$ 0.07 (1.44) | 0.76 $\pm$ 0.05 (1.50) |
| Real Estate | 0.50 $\pm$ 0.08 (1.00) | 0.50 $\pm$ 0.08 (1.00) | **0.59 $\pm$ 0.08 (0.82)** | 0.53 $\pm$ 0.04 (0.94) | 0.34 $\pm$ 0.50 (1.32) |
| Servo | **0.52 $\pm$ 0.11 (1.00)** | **0.52 $\pm$ 0.11 (1.00)** | **0.52 $\pm$ 0.11 (1.00)** | **0.52 $\pm$ 0.11 (1.00)** | 0.46 $\pm$ 0.20 (1.13) |
| Synch | **1.00 $\pm$ 0.00** | **1.00 $\pm$ 0.00** | **1.00 $\pm$ 0.00** | **1.00 $\pm$ 0.00** | 0.98 $\pm$ 0.00 |
| Yacht | 0.99 $\pm$ 0.01 (1.00) | **0.99 $\pm$ 0.00 (1.00)** | 0.99 $\pm$ 0.01 (1.00) | **0.99 $\pm$ 0.00 (1.00)** | **0.99 $\pm$ 0.00 (1.00)** |
| *Large-scale datasets* | | | | | |
| California Housing | **0.64 $\pm$ 0.01 (1.00)** | 0.61 $\pm$ 0.01 (1.08)* | 0.62 $\pm$ 0.01 (1.06) | 0.62 $\pm$ 0.01 (1.06) | 0.62 $\pm$ 0.01 (1.06) |
| Seoul Bike | 0.67 $\pm$ 0.02 (1.00) | **0.68 $\pm$ 0.02 (0.97)*** | 0.62 $\pm$ 0.03 (1.15) | 0.64 $\pm$ 0.02 (1.09) | 0.63 $\pm$ 0.02 (1.12) |
| Temperature (Max) | **0.75 $\pm$ 0.00 (1.00)** | 0.73 $\pm$ 0.01 (1.08)* | 0.73 $\pm$ 0.01 (1.08) | 0.75 $\pm$ 0.01 (1.00) | 0.75 $\pm$ 0.01 (1.00) |
| Temperature (Min) | **0.80 $\pm$ 0.01 (1.00)** | 0.79 $\pm$ 0.01 (1.05)* | **0.80 $\pm$ 0.01 (1.00)** | **0.80 $\pm$ 0.01 (1.00)** | **0.80 $\pm$ 0.01 (1.00)** |
| Walmart | 0.18 $\pm$ 0.02 (1.00) | **0.20 $\pm$ 0.01 (0.98)** | 0.11 $\pm$ 0.01 (1.09) | 0.11 $\pm$ 0.01 (1.09) | 0.11 $\pm$ 0.01 (1.09) |

## F.5. Multi-Step CLARITreeConst

**Formulating the Multi-Step CLARITreeConst optimization problem**  We formulate the Multi-Step CLARITreeConst construction as a recursive optimization problem solved via dynamic programming (Lin et al., 2020; McTavish et al., 2022), with a lookahead depth parameter $d_\ell < d$. The algorithm explores all combinations of splits up to depth $d_\ell$, with Greedy behavior beyond this depth. For a dataset $D$ and the current depth $d'$, the objective is defined as:

$$\mathcal{L}(D, d', d_\ell, d, \lambda) = \begin{cases} \textit{Phase 1: Recursive split (prefix)} \\ \min_{f \in \mathcal{F}, \tau \in \mathbb{R}} \Big\{ \lambda + \text{Leaf}(D), \ \mathcal{L}(D_{f \leq \tau}, d'+1, d_\ell, d, \lambda) + \mathcal{L}(D_{f > \tau}, d'+1, d_\ell, d, \lambda) \Big\}, & \text{if } d' < d_\ell, \\ \textit{Phase 2: Greedy completion (suffix)} \\ \min_{f \in \mathcal{F}, \tau \in \mathbb{R}} \Big\{ \lambda + \text{Leaf}(D), \ \mathcal{L}_g(D_{f \leq \tau}, d - d', \lambda) + \mathcal{L}_g(D_{f > \tau}, d - d', \lambda) \Big\}, & \text{if } d' = d_\ell, \end{cases}$$
$$(7)$$

where $D_{f \leq \tau}$ and $D_{f > \tau}$ denote the partitions induced by thresholding feature $f$ at value $\tau$, $\mathcal{L}_g(D, d - d', \lambda)$ is the loss of a Greedy of depth $d - d'$ built on dataset $D$, and $\text{Leaf}(D)$ denotes the loss from fitting a constant predictor directly on $D$. The overall objective $\mathcal{L}^*(D, d, \lambda)$ in (2) is approximated by $\mathcal{L}(D, 0, d_\ell, d, \lambda)$.

### F.5.1. Multi-Step CLARITreeConst Full Algorithm

We now present the Multi-Step CLARITreeConst Full Algorithm summarized in Algorithm 16. For simplicity of implementation, our implementation builds upon OSRT (Zhang et al., 2023), which provides an optimal solver for constant regression trees.

*Table 10.* In-sample performance comparison of constant regression tree methods (Depth = 5, Thresholds = 20). We report Train $R^2$ (mean $\pm$ std) for each dataset; the corresponding Train MSE ratio (relative to CLARITree) is shown in parentheses. Best results per dataset are highlighted in bold, and the second-best results are underlined. An asterisk ($*$) indicates that the selected configuration exceeded the training time limit (600s).

| Dataset | CLARITree | STreeD | GUIDE | CholeskyTree | CART |
|---|---|---|---|---|---|
| *Small / Medium-scale datasets* | | | | | |
| Airfoil | 0.77 ± 0.01 (1.00) | **0.78 ± 0.00 (0.96)** | 0.67 ± 0.01 (1.43) | 0.67 ± 0.02 (1.43) | 0.67 ± 0.02 (1.43) |
| Auction | **0.94 ± 0.00 (1.00)** | **0.94 ± 0.00 (1.00)** | 0.90 ± 0.01 (1.67) | 0.92 ± 0.01 (1.33) | 0.92 ± 0.01 (1.33) |
| Auto MPG | 0.90 ± 0.00 (1.00) | **0.93 ± 0.00 (0.70)** | 0.90 ± 0.01 (1.00) | 0.92 ± 0.01 (0.80) | 0.92 ± 0.01 (0.80) |
| Energy (Cooling) | 0.97 ± 0.00 (1.00) | **0.98 ± 0.00 (0.67)** | 0.96 ± 0.00 (1.33) | 0.96 ± 0.00 (1.33) | 0.96 ± 0.00 (1.33) |
| Energy (Heating) | **1.00 ± 0.00** | **1.00 ± 0.00** | 0.98 ± 0.00 | 0.99 ± 0.00 | 0.99 ± 0.00 |
| Insurance | 0.85 ± 0.01 (1.00) | 0.85 ± 0.01 (1.00) | 0.87 ± 0.01 (0.87) | 0.85 ± 0.01 (1.00) | **0.88 ± 0.01 (0.80)** |
| Optical Net | **0.92 ± 0.00 (1.00)** | 0.91 ± 0.01 (1.13) | 0.83 ± 0.03 (2.13) | 0.87 ± 0.01 (1.63) | 0.86 ± 0.01 (1.75) |
| Real Estate | 0.56 ± 0.05 (1.00) | 0.56 ± 0.05 (1.00) | 0.74 ± 0.03 (0.59) | 0.55 ± 0.05 (1.02) | **0.85 ± 0.01 (0.34)** |
| Servo | 0.59 ± 0.02 (1.00) | 0.59 ± 0.02 (1.00) | 0.59 ± 0.02 (1.00) | 0.59 ± 0.02 (1.00) | **0.60 ± 0.03 (0.98)** |
| Synch | **1.00 ± 0.00** | **1.00 ± 0.00** | **1.00 ± 0.00** | **1.00 ± 0.00** | 0.98 ± 0.00 |
| Yacht | 0.99 ± 0.00 (1.00) | **1.00 ± 0.00 (0.00)** | **1.00 ± 0.00 (0.00)** | **1.00 ± 0.00 (0.00)** | **1.00 ± 0.00 (0.00)** |
| *Large-scale datasets* | | | | | |
| California Housing | **0.66 ± 0.01 (1.00)** | 0.62 ± 0.00 (1.12)* | 0.63 ± 0.00 (1.09) | 0.63 ± 0.00 (1.09) | 0.63 ± 0.00 (1.09) |
| Seoul Bike | 0.69 ± 0.01 (1.00) | **0.70 ± 0.01 (0.97)*** | 0.63 ± 0.02 (1.19) | 0.66 ± 0.01 (1.10) | 0.66 ± 0.01 (1.10) |
| Temperature (Max) | **0.78 ± 0.00 (1.00)** | 0.74 ± 0.00 (1.18)* | 0.75 ± 0.00 (1.14) | 0.76 ± 0.00 (1.09) | 0.76 ± 0.00 (1.09) |
| Temperature (Min) | **0.82 ± 0.00 (1.00)** | 0.80 ± 0.00 (1.11)* | 0.81 ± 0.00 (1.06) | 0.81 ± 0.00 (1.06) | 0.81 ± 0.00 (1.06) |
| Walmart | 0.22 ± 0.01 (1.00) | **0.24 ± 0.00 (0.97)** | 0.14 ± 0.00 (1.10) | 0.14 ± 0.01 (1.10) | 0.15 ± 0.01 (1.09) |

---

**Algorithm 16** MultiStepCLARITreeConst($\ell, D, \lambda, d_l, d, p$)

---

**Require:** $\ell, D, \lambda, d_l, d, p$ {Loss function, samples, regularizer, lookahead depth, depth budget, postprocess flag}
1: ModifiedOSRT ← OSRT from (Zhang et al., 2023) reconfigured to use GETBOUNDS (Algorithm 17)
2: $t_{\text{lookahead}}$ ← ModifiedOSRT($\ell, D, \lambda, d_l$) {Call ModifiedOSRT with depth budget $d_l$}
3: **if** $p$ **then** {Fill in the leaves of this prefix}
4:     **for** leaf $u \in t_{\text{lookahead}}$ **do**
5:         $d_u$ ← depth of leaf
6:         $D(u)$ ← subproblem associated with $u$
7:         $\lambda_u \leftarrow \lambda \frac{|D|}{|D(u)|}$ {Renormalize $\lambda$ for the subproblem in question}
8:         $t_u$ ← OSRT($D(u), d - d_u, \lambda_u$) {Find the optimal regression subtree for $D(u)$}
9:         **if** $t_u$ is not a leaf **then**
10:           Replace leaf $u$ with subtree $t_u$
11:         **end if**
12:     **end for**
13: **end if**
14: **return** $t_{\text{lookahead}}$

---

---

**Algorithm 17** GETBOUNDS$(D, d_l, d, d', N) \rightarrow lb, ub$

---

**Require:** $D, d_l, d, d', N$ {Support, lookahead depth, current search depth, maximum search depth, size of full dataset in OSRT call}
1: **if** $d' = d_l$ **then**
2:     $T_g \leftarrow$ GREEDYCONST$(D, d - d_l, \lambda)$ {Find greedy constant regression tree rooted at $D$}
3:     $S(T_g) \leftarrow$ number of leaves in $T_g$
4:     $\alpha \leftarrow \frac{1}{N} \sum_{(x,y) \in D} (y - T_g(x))^2 + \lambda S(T_g)$
5:     $lb \leftarrow \alpha$
6:     $ub \leftarrow \alpha$ {Subproblem solved because $ub = lb$}
7: **else**
8:     $lb \leftarrow$ Algorithm 1 in (Zhang et al., 2023) {K-Means lower bound in OSRT}
9:     $ub \leftarrow \lambda$ {Upper bound via constant predictor loss}
10: **end if**
11: **return** $lb, ub$ {Return lower and upper bounds}

---

### F.5.2. MULTI-STEP CLARITREECONST THEORETICAL ANALYSIS

In this section, we present the theoretical analysis of the runtime complexity, identify The optimal lookahead depth of Multi-Step CLARITreeConst.

We provide a runtime analysis of the Multi-Step CLARITreeConst:

**Theorem F.3** (Multi-Step CLARITreeConst). *For depth $d$ and lookahead $d_\ell$, the runtime is*

$$\mathcal{O}\left(n(d - d_\ell)k^{d_\ell + 1}T^{d_\ell} + nk^{d - d_\ell}T^{d - d_\ell - 1}\right).$$

*If we cache repeated subproblems, the runtime reduces to* $\mathcal{O}\left(\frac{n(d-d_\ell)k^{d_\ell+1}T^{d_\ell-1}}{d_\ell!} + \frac{nk^{d-d_\ell}T^{d-d_\ell-1}}{(d-d_\ell)!}\right)$.

Thus, Multi-Step CLARITreeConst exhibits a complexity that reflects a trade-off between the prefix exploration cost and the suffix regression cost; the minimum occurs when the two terms are balanced, as formalized in the following Corollary F.4.

**Corollary F.4** (Optimal Lookahead Depth for Multi-Step CLARITreeConst). *For constant regression trees, the asymptotically optimal lookahead depth is* $d_\ell = \frac{d-1}{2}$, *independent of caching.*

Proofs are deferred to Appendix F.6.

### F.5.3. SIMPLE EXPERIMENTAL RESULTS

In this section, we use two medium-scale datasets to illustrate the experimental results of the multi-step CLARITreeConst method implemented on top of OSRT (Zhang et al., 2023). We also present representative slices to help illustrate the intuition behind Corollary F.4.

While multi-step lookahead in regression trees is an interesting direction for future work, we do not pursue it further in this paper. This decision is motivated by the fact that even one-step lookahead has already been shown to be highly effective in regression settings.

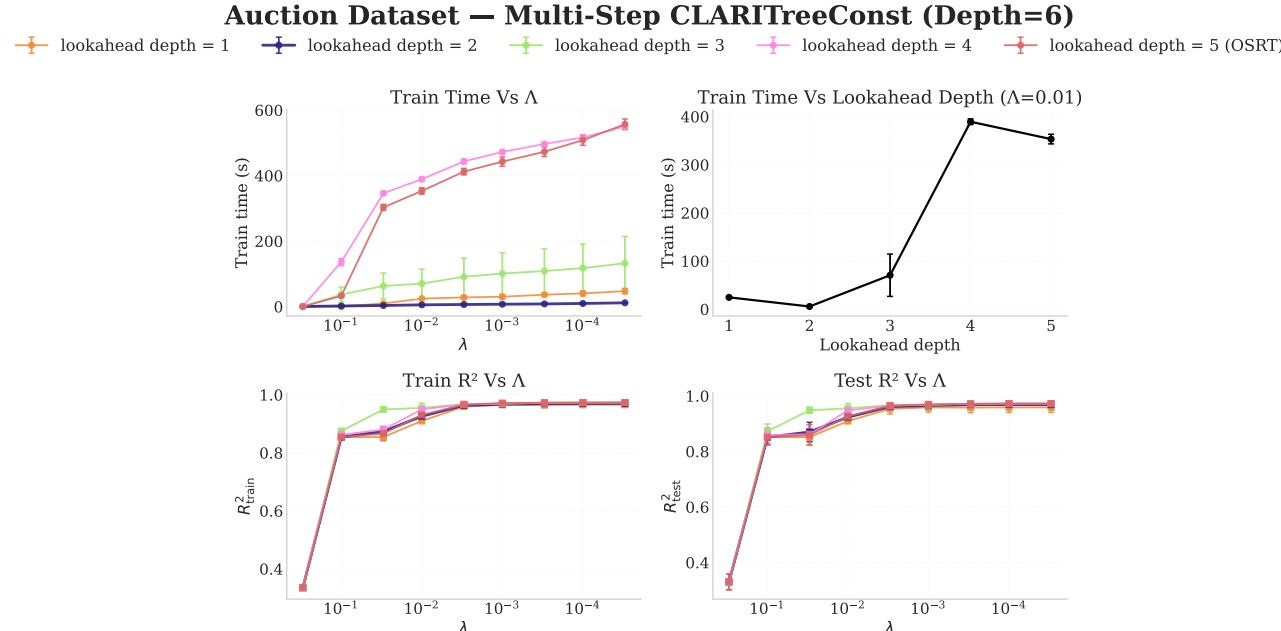

*Figure 29.* Results of the Multi-Step CLARITreeConst framework across different lookahead depths (depth = 5). We report training time, training $R^2$, and test $R^2$ as functions of the regularization strength $\lambda$, along with a fixed $\lambda = 0.01$ slice illustrating the dependence on lookahead depth.

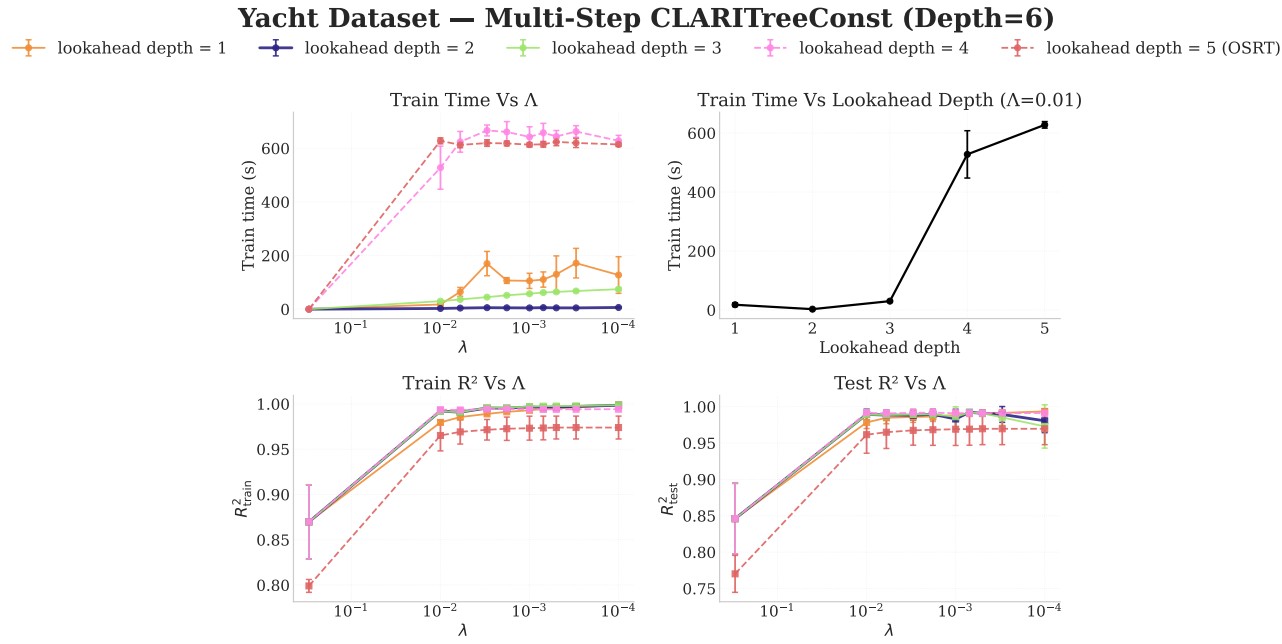

*Figure 30.* Results of the Multi-Step CLARITreeConst framework across different lookahead depths (depth = 5). We report training time, training $R^2$, and test $R^2$ as functions of the regularization strength $\lambda$, along with a fixed $\lambda = 0.01$ slice illustrating the dependence on lookahead depth. Dashed lines indicate runs that hit the 600-second time limitation.

## F.6. Proof Details for Special Cases

**Runtime Analysis**  In a constant regression tree, each leaf prediction is given by the sample mean. Computing the mean requires the same order of operations as computing the majority class in classification, as both procedures involve enumerating all samples within the leaf. Hence, the computational complexity is $\mathcal{O}(n)$.

As a result, for both CLARITREECONST and MULTI-STEP CLARITREECONST, the overall runtime complexity analysis is identical to that of standard decision trees. We can therefore directly rely on existing complexity results for decision trees established in the literature (Theorem 6.1, Corollary 6.2, and Theorem 6.4 of Babbar et al., 2025).

**Accuracy Analysis  Data Generating Process (DGP).**  Fix a depth budget $d \geq 2$ and choose an integer $U > 2d$. Let the features be $X = (g, H, M) \in \{0,1\}^{1+(d-1)+U}$ with independent coordinates $X_i \sim \mathrm{Ber}(\frac{1}{2})$:

$$g := X_1, \qquad H := (X_2, \ldots, X_d), \qquad M := (X_{d+1}, \ldots, X_{d+U}).$$

Fix $\varepsilon > 0$. Define $Y$ by the mixture

$$Y = \begin{cases} g \oplus \mathrm{Maj}(H), & \text{with probability } 1 - \varepsilon, \\ \mathrm{Maj}(M), & \text{with probability } \varepsilon, \end{cases}$$

where $\oplus$ is XOR and $\mathrm{Maj}(\cdot)$ is the majority function. Tie-breaking rule: whenever a majority tie occurs (e.g., $|H|$ or $U$ is even), break it with an independent fair coin, independent of $(g, H, M)$. This ensures $\mathbb{E}[\mathrm{Maj}(H)] = \mathbb{E}[\mathrm{Maj}(M)] = \frac{1}{2}$.

### Auxiliary Lemma

**Lemma F.5** (Conditional one-step reduction equals $4\mathrm{Cov}^2$). *Let $Y, X = (g, H, M)$ follows the definition in Appendix F.6 and $X_i \in \{0,1\}$ is the new feature plan to split. Fix a set of prefix coordinates $M_\mathcal{S} := \{X_j = x_j : j \in \mathcal{S}\}$ and condition on it. By the independence, $\Pr(X_i = 1 \mid M_\mathcal{S}) = \Pr(X_i = 1) = \frac{1}{2}$. Write $p_b := \Pr(Y = 1 \mid X = b, M_\mathcal{S})$ and $p := \mathbb{E}[Y \mid M_\mathcal{S}]$. Then the MSE reduction obtained by splitting on $X_i$ at the node $M_\mathcal{S}$ is*

$$\Delta_{\mathrm{MSE}}(X_i \mid M_\mathcal{S}) := \mathrm{Var}(Y \mid M_\mathcal{S}) - \mathbb{E}\big[\mathrm{Var}(Y \mid X_i, M_\mathcal{S})\big|M_\mathcal{S}\big] = 4\mathrm{Cov}(Y, X_i \mid M_\mathcal{S})^2.$$

*Proof.*  By definition $p = \mathbb{E}[Y \mid M_\mathcal{S}] = \Pr(Y = 1 \mid M_\mathcal{S}) = \frac{1}{2}(p_0 + p_1)$. Since $Y \in \{0,1\}$, we have

$$\mathrm{Var}(Y \mid M_\mathcal{S}) = p(1-p) = \tfrac{1}{2}(p_0 + p_1) - \tfrac{1}{4}(p_0 + p_1)^2.$$

Moreover, using $\Pr(X_i = 1 \mid M_\mathcal{S}) = \frac{1}{2}$,

$$\mathbb{E}\big[\mathrm{Var}(Y \mid X_i, M_\mathcal{S})\big|M_\mathcal{S}\big] = \tfrac{1}{2}\big(p_0(1-p_0) + p_1(1-p_1)\big) = \tfrac{1}{2}(p_0 + p_1) - \tfrac{1}{2}(p_0^2 + p_1^2).$$

Subtracting yields

$$\Delta_{\mathrm{MSE}}(X_i \mid M_\mathcal{S}) = \mathrm{Var}(Y \mid M_\mathcal{S}) - \mathbb{E}\big[\mathrm{Var}(Y \mid X_i, M_\mathcal{S})\big|M_\mathcal{S}\big] = \tfrac{1}{4}(p_1 - p_0)^2.$$

On the other hand, we have

$$\mathrm{Cov}(Y, X_i \mid M_\mathcal{S}) = \mathbb{E}[YX_i \mid M_\mathcal{S}] - \mathbb{E}[Y \mid M_\mathcal{S}]\,\mathbb{E}[X_i \mid M_\mathcal{S}] = \tfrac{1}{2}p_1 - \tfrac{1}{2}(p_0 + p_1) \cdot \tfrac{1}{2} = \tfrac{1}{4}(p_1 - p_0),$$

so $\Delta_{\mathrm{MSE}}(X_i \mid M_\mathcal{S}) = 4\mathrm{Cov}(Y, X_i \mid M_\mathcal{S})^2$.  $\square$

*Remark* F.6.  By Lemma F.5, one-step greedy MSE splitting in our DGP is equivalent to choosing the feature with maximal $|\mathrm{Cov}(Y, X \mid M_\mathcal{S})|$.

**Lemma F.7** (Conditional covariance–influence identity). *Let $f : \{0,1\}^n \to \{0,1\}$ be monotone and $X \sim \mathrm{Ber}(\frac{1}{2})^n$. Fix $\mathcal{S} \subseteq [n]$ and condition on $M_\mathcal{S} := \{X_j = x_j : j \in \mathcal{S}\}$. Define the conditional influence for $i \notin \mathcal{S}$ by*

$$\mathrm{Inf}_i(f \mid M_\mathcal{S}) := \Pr\big(f(X) \neq f(X^{\oplus i})\big|M_\mathcal{S}\big),$$

*where $X^{\oplus i}$ denotes the random vector obtained by flipping the $i$-th coordinate of $X$, i.e. $X^{\oplus i} = (X_1, \ldots, 1 - X_i, \ldots, X_n)$.*

*Then, for all $i \notin \mathcal{S}$,*

$$\mathrm{Cov}(f(X), X_i \mid M_\mathcal{S}) = \tfrac{1}{4}\mathrm{Inf}_i(f \mid M_\mathcal{S}).$$

*Proof.* Because conditioning on $M_{\mathcal{S}}$ fixes the coordinates in $\mathcal{S}$, the remaining bits are still independent $\mathrm{Ber}(\frac{1}{2})$. In particular, for $i \notin \mathcal{S}$ we have $\Pr(X_i = 1 \mid M_{\mathcal{S}}) = \frac{1}{2}$.

By definition,
$$\mathrm{Cov}(f(X), X_i \mid M_{\mathcal{S}}) = \mathbb{E}[f(X)X_i \mid M_{\mathcal{S}}] - \mathbb{E}[f(X) \mid M_{\mathcal{S}}]\,\mathbb{E}[X_i \mid M_{\mathcal{S}}].$$

Using $\mathbb{E}[X_i \mid M_{\mathcal{S}}] = \frac{1}{2}$, we obtain
$$\mathrm{Cov}(f(X), X_i \mid M_{\mathcal{S}}) = \frac{1}{2}\Big( \mathbb{E}[f(X) \mid X_i = 1, M_{\mathcal{S}}] - \mathbb{E}[f(X) \mid M_{\mathcal{S}}]\Big).$$

Expanding $\mathbb{E}[f(X) \mid M_{\mathcal{S}}]$ via the law of total expectation,
$$\mathbb{E}[f(X) \mid M_{\mathcal{S}}] = \tfrac{1}{2}\,\mathbb{E}[f(X) \mid X_i = 1, M_{\mathcal{S}}] + \tfrac{1}{2}\,\mathbb{E}[f(X) \mid X_i = 0, M_{\mathcal{S}}],$$

so that
$$\mathrm{Cov}(f(X), X_i \mid M_{\mathcal{S}}) = \frac{1}{4}\Big( \mathbb{E}[f(X) \mid X_i = 1, M_{\mathcal{S}}] - \mathbb{E}[f(X) \mid X_i = 0, M_{\mathcal{S}}]\Big).$$

For monotone $f$, the difference
$$\mathbb{E}[f(X) \mid X_i = 1, M_{\mathcal{S}}] - \mathbb{E}[f(X) \mid X_i = 0, M_{\mathcal{S}}] = \Pr(f(X) \neq f(X^{\oplus i}) \mid M_{\mathcal{S}}) = \mathrm{Inf}_i(f \mid M_{\mathcal{S}}).$$

Thus,
$$\mathrm{Cov}(f(X), X_i \mid M_{\mathcal{S}}) = \tfrac{1}{4}\mathrm{Inf}_i(f \mid M_{\mathcal{S}}).$$

$\square$

**Lemma F.8** (Conditional influence of majority). *Let* $\mathrm{Maj}(M)$ *follows the definition in Appendix F.6. Fix* $\mathcal{S} \subseteq [U]$ *and condition on* $M_{\mathcal{S}} := \{X_j = x_j : j \in \mathcal{S}\}$, *where we assume all the* $M_{\mathcal{S}}$ *is sampled from* $M$. *For any* $\ell \notin \mathcal{S}$, *if* $k = \sum_{j \in \mathcal{S}} x_j$ *is the number of fixed ones, then when* $U$ *is odd, we have*
$$\mathrm{Inf}_\ell(\mathrm{Maj}(M) \mid M_{\mathcal{S}}) = \binom{U - 1 - |\mathcal{S}|}{\lfloor U/2 \rfloor - k} 2^{-(U-1-|\mathcal{S}|)},$$

*and when* $U$ *is even, we have*
$$\mathrm{Inf}_\ell(\mathrm{Maj}(M) \mid M_{\mathcal{S}}) = \frac{1}{2}\Big[\binom{U-1-|\mathcal{S}|}{\lfloor U/2 \rfloor - 1 - k} 2^{-(U-1-|\mathcal{S}|)} + \binom{U-1-|\mathcal{S}|}{\lfloor U/2 \rfloor - k} 2^{-(U-1-|\mathcal{S}|)}\Big].$$

*(with the convention that the binomial coefficient is 0 if its lower index is negative or exceeds the upper index). Moreover, there exist absolute constants* $c_-, c_+ > 0$ *such that, for all* $U \geq 2$,
$$\frac{c_-}{\sqrt{U - |\mathcal{S}|}} \leq \mathrm{Inf}_\ell(\mathrm{Maj}(M) \mid M_{\mathcal{S}}) \leq \frac{c_+}{\sqrt{U - |\mathcal{S}|}}.$$

*Proof.* By definition,
$$\mathrm{Inf}_\ell(\mathrm{Maj}(M) \mid M_{\mathcal{S}}) = \Pr\big(\mathrm{Maj}(M) \neq \mathrm{Maj}(M^{\oplus \ell}) \mid M_{\mathcal{S}}\big)$$
$$= \frac{1}{2}\big[\Pr\big(\mathrm{Maj}(M) \neq \mathrm{Maj}(M^{\oplus \ell}) \mid X_\ell = 1, M_{\mathcal{S}}\big) + \Pr\big(\mathrm{Maj}(M) \neq \mathrm{Maj}(M^{\oplus \ell}) \mid X_\ell = 0, M_{\mathcal{S}}\big)\big]$$
$$= \frac{1}{2}\big[\Pr(\mathrm{Maj}(M) = 1, \mathrm{Maj}(M^{\oplus \ell}) = 0 \mid X_\ell = 1, M_{\mathcal{S}}) + \Pr(\mathrm{Maj}(M) = 0, \mathrm{Maj}(M^{\oplus \ell}) = 1 \mid X_\ell = 0, M_{\mathcal{S}})\big]$$
$$= \Pr(\mathrm{Maj}(M) = 1, \mathrm{Maj}(M^{\oplus \ell}) = 0 \mid X_\ell = 1, M_{\mathcal{S}})$$

Thus, we only need to consider the case before flipping $X_\ell$ to one.

When $U$ is odd, the number of ones should equal to $\lfloor U/2 \rfloor$. Under the conditioning $M_{\mathcal{S}}$, suppose $k$ of these ones are already fixed; then the remaining $U - 1 - |\mathcal{S}|$ free bits must contribute exactly $\lfloor U/2 \rfloor - k$ ones. Since the free bits are independent $\mathrm{Ber}(\frac{1}{2})$ variables, the corresponding probability is
$$\binom{U - 1 - |\mathcal{S}|}{\lfloor U/2 \rfloor - k} 2^{-(U-1-|\mathcal{S}|)}.$$

When $U$ is even, the situation is slightly more involved. We need to consider two possibilities: (1) before flipping $X_\ell$ to one, there are $U/2 - 1$ ones, and after the flip, the tie-breaking rule makes $\mathrm{Maj} = 1$; or (2) before the flip, there are $U/2$ ones, and the tie-breaking rule makes $\mathrm{Maj} = 0$, so flipping $X_\ell$ changes the majority. As defined in Appendix F.6, the tie-breaking rule is an independent Bernoulli random variable, so each case occurs with probability $1/2$. Hence, the total probability is

$$\frac{1}{2}\left[\binom{U-1-|\mathcal{S}|}{\lfloor U/2 \rfloor - 1 - k}2^{-(U-1-|\mathcal{S}|)} + \binom{U-1-|\mathcal{S}|}{\lfloor U/2 \rfloor - k}2^{-(U-1-|\mathcal{S}|)}\right].$$

**Upper bound.** Let $m = U - 1 - |\mathcal{S}|$. By unimodality and Stirling's formula,

$$\mathrm{Inf}_\ell(\mathrm{Maj}(M) \mid M_\mathcal{S}) \leq \binom{m}{\lfloor m/2 \rfloor}2^{-m} \leq \sqrt{\frac{2}{\pi m}} \leq \frac{c_+}{\sqrt{U - |\mathcal{S}|}} \quad \text{with } c_+ = 2\sqrt{\frac{2}{\pi}}.$$

**Lower bound.** Write the target count among the $m$ free bits as $m/2 + \delta$, where $\delta = |\mathcal{S}|/2 - k$ if $U$ is odd and $\delta = |\mathcal{S}|/2 \pm \frac{1}{2} - k$ if $U$ is even. If $|\delta| \leq C_0$ for some absolute constant $C_0$, then the local CLT yields

$$\binom{m}{\frac{m}{2} + \delta}2^{-m} \geq \frac{c(C_0)}{\sqrt{m}} \quad \Rightarrow \quad \mathrm{Inf}_\ell(\mathrm{Maj}(M) \mid M_\mathcal{S}) \geq \frac{c_-}{\sqrt{U - |\mathcal{S}|}}.$$

$\square$

*Remark* F.9. In the regression-tree setting, the conditioning set $\mathcal{S}$ corresponds to the variables that have already been fixed along the current path, so its size satisfies $|\mathcal{S}| < d$, where $d$ is the depth budget of the tree. Consequently, $\delta = |\mathcal{S}|/2 - k$ (or $\delta = |\mathcal{S}|/2 \pm \frac{1}{2} - k$) is automatically bounded by $|\mathcal{S}| < d$. In particular, as long as the path depth $d$ is a fixed constant, the balance condition $|\delta| \leq C_0$ in the lower bound above holds with an absolute $C_0$, and hence the same constant $c_-$ applies.

**Main Theorem: Optimality of CLARITree under the DGP**

**Theorem F.10** (CLARITree can be arbitrarily better than greedy under MSE). *For any depth budget $d \geq 2$ and any $\varepsilon > 0$, under the distribution above with $U > d$,*

1. *Greedy lower bound. Every depth-$d$ greedy constant regression tree never queries $g$ or any $H$-coordinate along any root-to-leaf path of length at most $d$. Consequently,*

$$\mathrm{MSE}_{Greedy} \geq \frac{1 - \varepsilon^2}{4}.$$

2. *CLARITree upper bound. The depth-$d$ CLARITree tree with lookahead 1 splits on $g$ at the root and then, on each branch, greedily splits only on $H$ for the remaining $d-1$ levels, thereby computing $\mathrm{Maj}(H)$ exactly on the $(1-\varepsilon)$-mass. Its MSE satisfies*

$$\mathrm{MSE}_{CLARITreeConst} = \frac{\varepsilon}{2} - \frac{\varepsilon^2}{4}.$$

*Thus we have*

$$\frac{\mathrm{MSE}_{Greedy}}{\mathrm{MSE}_{CLARITreeConst}} \geq \frac{1}{2\varepsilon}.$$

*Consequently, $\mathrm{MSE}_{CLARITreeConst} \to 0$ as $\varepsilon \to 0$, whereas $\mathrm{MSE}_{Greedy} \geq (1 - \varepsilon^2)/4$ stays bounded away from 0; the gap can be made arbitrarily large.*

*Proof.* **Greedy selects only $M$-bits for the first $d$ levels.** Fix any node defined by conditioning on a subset $\mathcal{S} \subseteq \{d+1, \ldots, d+U\}$ of $M$-indices, with $M_\mathcal{S} = (X_j)_{j \in \mathcal{S}}$. We compute the conditional covariances.

*Covariance with $g$.* Since $g \perp (H, M)$ and $\mathrm{Maj}(H) \perp M$,

$$\mathrm{Cov}(Y, g \mid M_\mathcal{S}) = \mathbb{E}[Yg \mid M_\mathcal{S}] - \mathbb{E}[Y \mid M_\mathcal{S}]\mathbb{E}[g]$$
$$= (1 - \varepsilon)\,\mathbb{E}[g(g \oplus \mathrm{Maj}(H)) \mid M_\mathcal{S}] + \varepsilon\,\mathbb{E}[g\mathrm{Maj}(M) \mid M_\mathcal{S}]$$

$$- \Big( (1-\varepsilon)\, \mathbb{E}[g \oplus \mathrm{Maj}(H) \mid M_{\mathcal{S}}] + \varepsilon\, \mathbb{E}[\mathrm{Maj}(M) \mid M_{\mathcal{S}}] \Big) \tfrac{1}{2}$$

$$= \frac{1-\varepsilon}{4} + \frac{\varepsilon}{2}\, \Pr(\mathrm{Maj}(M){=}1 \mid M_{\mathcal{S}})$$

$$\quad - \frac{1}{2}\Big( \frac{1-\varepsilon}{2} + \varepsilon\, \Pr(\mathrm{Maj}(M){=}1 \mid M_{\mathcal{S}}) \Big)$$

$$= 0,$$

where we used $\mathbb{E}[g(g \oplus \mathrm{Maj}(H))] = \tfrac{1}{4}$ and $\mathbb{E}[g \oplus \mathrm{Maj}(H)] = \tfrac{1}{2}$, both consequences of $g \sim \mathrm{Ber}(\tfrac{1}{2})$ and $g \perp \mathrm{Maj}(H)$.

*Covariance with $H_j$.* Since $H_j \perp M$ and $\mathbb{E}[H_j] = \tfrac{1}{2}$,

$$\mathrm{Cov}(Y, H_j \mid M_{\mathcal{S}}) = (1-\varepsilon)\, \mathbb{E}[H_j(g \oplus \mathrm{Maj}(H)) \mid M_{\mathcal{S}}] + \varepsilon\, \mathbb{E}[H_j\mathrm{Maj}(M) \mid M_{\mathcal{S}}]$$

$$\quad - \Big( (1-\varepsilon)\, \mathbb{E}[g \oplus \mathrm{Maj}(H) \mid M_{\mathcal{S}}] + \varepsilon\, \mathbb{E}[\mathrm{Maj}(M) \mid M_{\mathcal{S}}] \Big) \tfrac{1}{2}$$

$$= (1-\varepsilon)\, \mathbb{E}\big[ H_j(g + \mathrm{Maj}(H) - 2g\mathrm{Maj}(H)) \big] + \tfrac{\varepsilon}{2}\, \mathbb{E}[\mathrm{Maj}(M)]$$

$$\quad - \tfrac{1}{2}\Big( (1-\varepsilon)\, \mathbb{E}[g + \mathrm{Maj}(H) - 2g\mathrm{Maj}(H)] + \varepsilon\, \mathbb{E}[\mathrm{Maj}(M)] \Big)$$

$$= (1-\varepsilon)\big( \tfrac{1}{4} + \mathbb{E}[H_j\mathrm{Maj}(H)] - 2\tfrac{1}{2}\, \mathbb{E}[H_j\mathrm{Maj}(H)] \big)$$

$$= 0,$$

using $g \perp (H, M)$ so that $\mathbb{E}[H_j g] = \tfrac{1}{4}$ and $\mathbb{E}[H_j g\mathrm{Maj}(H)] = \tfrac{1}{2}\, \mathbb{E}[H_j\mathrm{Maj}(H)]$.

*Covariance with $M_\ell$.* By the same expansion and using independence of $(g, H)$ and $M$,

$$\mathrm{Cov}(Y, M_\ell \mid M_{\mathcal{S}}) = \varepsilon\, \mathrm{Cov}(\mathrm{Maj}(M), M_\ell \mid M_{\mathcal{S}}).$$

By Lemmas F.7 and F.8, $\mathrm{Cov}(\mathrm{Maj}(M), M_\ell \mid M_{\mathcal{S}}) = \tfrac{1}{4}\mathrm{Inf}_\ell(\mathrm{Maj}_U \mid M_{\mathcal{S}}) = \Theta\big(1/\sqrt{U - |\mathcal{S}|}\big) > 0$ for any unseen $M_\ell \notin \mathcal{S}$. Therefore, by Lemma F.5, the one-step MSE gain is zero for $g$ and all $H_j$, and strictly positive for some unseen $M_\ell$. Greedy must split on $M$. Inducting on depth and using $U > d$ shows greedy picks only unseen $M$-bits for the first $d$ levels.

*Greedy MSE lower bound.* Fix a leaf $L$ of a depth-$d$ greedy tree. Since only $M$ was queried, on the $(1-\varepsilon)$-mass the label $g \oplus \mathrm{Maj}(H)$ remains unbiased with $\Pr(Y = 1 \mid L, (1-\varepsilon)\text{-mass}) = \tfrac{1}{2}$; on the $\varepsilon$-mass, $Y = \mathrm{Maj}(M)$ may be biased by the $M$-splits. Let $q_L := \Pr(\mathrm{Maj}(M) = 1 \mid L)$. Then

$$p_L := \Pr(Y = 1 \mid L) = (1-\varepsilon) \cdot \tfrac{1}{2} + \varepsilon q_L \in \Big[ \tfrac{1-\varepsilon}{2}, \tfrac{1+\varepsilon}{2} \Big].$$

The optimal constant prediction in $L$ has MSE $p_L(1 - p_L)$, minimized at the endpoints, giving $p_L(1 - p_L) \geq (1 - \varepsilon^2)/4$. Averaging over leaves yields $\mathrm{MSE}_{\mathrm{Greedy}} \geq (1 - \varepsilon^2)/4$.

**CLARITree splits on $g$, then on $H$.** Consider the CLARITree objective with lookahead 1. A root split on $g$ produces children where, on the $(1-\varepsilon)$-mass, $Y$ equals either $\mathrm{Maj}(H)$ or $1 - \mathrm{Maj}(H)$, while the $\varepsilon$-mass remains $\mathrm{Maj}(M)$. Thus, in either child, for any step with $r \in \{1, \ldots, d-1\}$ remaining unseen $H$-bits and at least one unseen $M$-bit,

$$\max_{X_j \in H} |\mathrm{Cov}(Y, X_j \mid g)| \asymp (1-\varepsilon)\frac{1}{\sqrt{r}}, \qquad \max_{X_\ell \in M} |\mathrm{Cov}(Y, X_\ell \mid g)| \asymp \varepsilon\frac{1}{\sqrt{U}},$$

by Lemmas F.7 and F.8 applied to $H$ and to $M$, respectively. As long as $\varepsilon$ is small enough, then the second step MSE gain is then larger for an unseen $H$-bit. By Lemma F.5, the one-step MSE gain is then always larger for an unseen $H$-bit, so the greedy completion after splitting on $g$ selects only $H$ for the remaining $d - 1$ levels. Since $|H| = d - 1$, the $(1 - \varepsilon)$-mass becomes perfectly pure in every leaf.

*Leaf-wise MSE under the mixture.* In any leaf after this policy, write the $(1-\varepsilon)$-mass label as a constant $y_\star \in \{0, 1\}$. Let $Z \sim \mathrm{Ber}(\varepsilon)$ indicate the $\varepsilon$-mass, independent of everything else. Then

$$Y = \begin{cases} y_\star, & Z = 0 \ (\text{prob. } 1 - \varepsilon), \\ \mathrm{Ber}(\tfrac{1}{2}), & Z = 1 \ (\text{prob. } \varepsilon). \end{cases}$$

Hence $\mathbb{E}[Y \mid \text{leaf}] = (1 - \varepsilon)y_\star + \frac{\varepsilon}{2}$ and $\text{Var}(Y \mid \text{leaf}) = \mathbb{E}[Y \mid \text{leaf}](1 - \mathbb{E}[Y \mid \text{leaf}]) = \frac{\varepsilon}{2} - \frac{\varepsilon^2}{4}$, independent of $y_\star$. This equals the optimal constant-regression MSE within each leaf; averaging over all leaves yields

$$\text{MSE}_{\text{CLARITreeConst}} = \frac{\varepsilon}{2} - \frac{\varepsilon^2}{4}.$$

Hence, a tree that first splits on $g$ achieves this error.

Although $g$ need not be the globally optimal first split, any tree chosen by the one-step lookahead rule of CLARITree, which maximizes the same objective, can only perform at least as well as the tree that splits on $g$ at the root.

Finally note that

$$\left(\frac{1 - \varepsilon^2}{4}\right) / \left(\frac{\varepsilon}{2} - \frac{\varepsilon^2}{4}\right) = \frac{1 + \varepsilon}{2\varepsilon} \geq \frac{1}{2\varepsilon} \tag{8}$$

$\square$

*Remark* F.11 (Finite samples). With $n$ i.i.d. samples, each empirical covariance concentrates around its mean at rate $\mathcal{O}\left(\sqrt{\log(d + U)/n}\right)$ by Hoeffding plus a union bound. For large enough $n$, the empirical order of covariances matches the population order with high probability, so empirical greedy and CLARITree trees achieve the same bounds up to $o(1)$.

