# OpenReview forum: "CLARITree: Cholesky and Lookahead Accelerations for Regression with Interpretable Piecewise Linear Trees"
_ICML.cc/2026/Conference — ICML 2026 regular_

### Official Review · Reviewer_8Yxh · 2026-03-10

**Soundness:** 3
**Presentation:** 3
**Significance:** 2
**Originality:** 3
**Overall Recommendation:** 4
**Confidence:** 5

**Summary:**

The authors propose CLARITree, an algorithm designed to train piecewise linear regression trees. It attempts to find a middle ground between fast but suboptimal greedy methods and highly accurate but computationally prohibitive optimal dynamic programming methods. The authors achieve this by combining a one-step lookahead split strategy with efficient rank-one Cholesky updates to incrementally maintain the regularized linear regressors in the leaf nodes. I.e., the split is made optimally for the current node assuming the subtrees below are constructed greedily.

**Compliance With Llm Reviewing Policy:**

Affirmed.

**Final Justification:**

All my concerns are resolved. I agree the proposed method is tractable for the classical tabular datasets targeted in this work, I still believe that scaling to high-dimensional continuous spaces remains a highly important challenge as such models can be used on dense (visual/text) embeddings for down-stream tasks. I keep my score.

**Key Questions For Authors:**

N/A

**Limitations:**

The runtime complexity $\mathcal{O}(d^2 n k^4 T)$ strictly limits the algorithm's applicability to low-dimensional feature spaces.

**Strengths And Weaknesses:**

Strengths:
- The application of rank-one Cholesky updates to incrementally maintain the regularized Gram matrix during continuous threshold scanning is a mathematically sound and clever alternative to recomputing the Gram matrix every time.
- The algorithm relies on a partly greedy lookahead approach incorporating linear leaves, which makes the resulting piecewise linear model significantly more flexible and expressive than standard constant-leaf trees.

Weaknesses:
- Although the method scales better than exact optimal solvers, it still cannot scaled for real-world large-scale problems. The training time complexity is $\mathcal{O}(d^2 n k^4 T)$. The $k^4$ term makes it prohibitively expensive for high-dimensional data. Moreover, dealing with real continuous values would make $T$ approach $n$, which makes the algorithm quadratic in the number of samples.
- The algorithm relies on a lookahead search coupled with an early-stopping mechanism. However, post-pruning techniques are generally considered superior to early stopping for mitigating the short-sightedness of splits ([1, chapter 9.2.2]).

Minor Weaknesses:
- In Equation (1), explicitly including the depth parameter $d$ in the prediction loss $\mathcal{R}(T_d, d, D)$ is mathematically redundant since the definition $T_d \in \mathcal{T}$ already constrains the tree to a maximum depth of $d$.
- Equation (1) outlines the global loss without a regularization term on the leaf nodes. However, Equation (2) explicitly has an $l_2$ penalty parameter $\kappa$ to fit the actual leaf models. The global objective is inconsistent with the local implementation.
- The structural complexity penalty $\lambda$ is listed as an input, but is never used in Algorithm 3 (Greedy).

References:
[1] Trevor J. Hastie and Robert J. Tibshirani and Jerome H. Friedman. "The Elements of Statistical Learning---Data Mining, Inference and Prediction" (2009)

---

> ### Author Rebuttal · Authors · 2026-03-31
>
> Thank you for your thoughtful review of our work.
>
> > W1: Although the method scales better than exact optimal solvers, it still cannot scaled for real-world large-scale problems.
>
>
> In our paper, we show some examples of our method handling large datasets; in particular, the California Housing data set (see Table 1 and Figure 2) has > 20,000 samples and more than 150 binary features; our method completes in under two minutes (~100 seconds). If there are any other large-scale datasets (especially in high stakes settings where interpretability may be important) that you think would be interesting, please let us know - we'd be excited to use them!
>
> One key factor explaining our scalability in these datasets is that the $k$ in the complexity analysis relates to the number of continuous features, not the number of binary features. Outside of genomics, we're not aware of many real-world datasets where there are a huge number of continuous features that are all necessary to incorporate; even, for example, one of the largest tabular datasets in [1], Higgs, has over a million samples but only 28 features. In these large-scale settings (or in settings where the number of continuous features can be narrowed down by feature selection), our method still scales quite well.
>
> The asymptotic scaling with the number of features is a limitation, but, in practice, is not prohibitive for many large datasets. For extremely large datasets, it is common to first reduce dimensionality using established techniques such as feature selection or knockoff-based filtering, which can substantially reduce the effective number of continuous variables without sacrificing predictive performance. Please also see our responses to Reviewer bGd1 (Q1). We will include a more detailed discussion of this limitation in the camera-ready version.
>
> > W2: The algorithm relies on a lookahead search coupled with an early-stopping mechanism. However, post-pruning techniques are generally considered superior to early stopping for mitigating the short-sightedness of splits.
>
> To evaluate this concern, we removed the early-stopping rule that prevents a split when it does not immediately improve over the parent node and reran the greedy baseline. In particular, we now first find the best split according to the greedy heuristic, complete the left and right subproblems greedily, and then check if the resulting tree is worse than a leaf; this now corresponds to a post-pruned greedy tree.
>
> ### Early Stop vs No Early Stop (ratio=no/early)
>
> |Dataset|TrainMSE|TestMSE|Time|Leaves|
> |---|---|---|---|---|
> |airfoil|0.998|0.998|1.087|1.036|
> |auction|1.000|1.000|0.617|1.024|
> |auto_mpg|0.993|1.003|0.865|1.062|
> |california_housing|1.002|1.001|1.970|1.073|
> |energe_c|0.996|0.988|1.440|1.009|
> |energe_h|0.973|0.964|0.813|1.058|
> |insurance|0.997|1.006|1.184|1.079|
> |optical_net|0.997|0.995|1.420|1.081|
> |real_estate|0.999|1.000|1.550|1.009|
> |seoul_bike|1.000|1.000|1.740|1.000|
> |servo|1.001|1.001|1.156|0.995|
> |synch|1.000|1.000|3.090|1.000|
> |temperature_max|1.002|1.004|1.290|1.067|
> |temperature_min|1.008|1.013|1.119|1.077|
> |walmart|1.000|1.000|2.110|1.059|
> |yacht|0.987|0.998|0.999|1.011|
>
> **Overall Mean**
> - **Train**: 0.997
> - **Test**: 0.998
> - **Time**: 1.404
> - **Leaves**: 1.040
>
> We observe that removing early stopping leads to only marginal changes in MSE,slightly increased training time, and slightly more leaves,consistent with standard expectations. Overall, the differences are negligible and do not affect the conclusions.
>
>
> > Minor Weakness
>
> Thank you for pointing out these issues. We agree and will clarify the redundancy in Eq. (1), ensure consistency between the global objective and the local implementation, and explicitly incorporate the structural penalty $\lambda$ into Algorithm 3. In particular, $\lambda$ is used to discourage further splitting when the reduction in loss is below $2\lambda$. These changes will be included in the camera-ready version.
>
> **Reference**
>
> [1] Grinsztajn, Léo, Edouard Oyallon, and Gaël Varoquaux. "Why do tree-based models still outperform deep learning on typical tabular data?." (2022)

---

> > ### Author Rebuttal · Reviewer_8Yxh · 2026-04-02
> >
> > I thank the authors for the detailed rebuttal and for taking the time to run the ablation experiment regarding early stopping versus post-pruning. The empirical results address my concern there.
> >
> > While I agree this is tractable for the classical tabular datasets targeted in this work, I still believe that scaling to high-dimensional continuous spaces remains a highly important challenge as such models can be used on dense (visual/text) embeddings for down-stream tasks.
> >
> > I will keep my positive score.

---

> > > ### Author Response · Authors · 2026-04-04
> > >
> > > Thank you for your positive feedback and support. We’re glad that our response addressed your concerns regarding early stopping. We agree that scaling to high-dimensional continuous spaces is an important direction. Our current work is primarily focused on classical interpretable tabular settings, rather than on dense representations such as visual or text embeddings. In such domains, interpretability is often approached through different model classes (e.g., prototype networks or concept-based networks), rather than tree-based models. We will clarify this intended scope in the camera-ready version to better position our contribution for readers.

---

### Official Review · Reviewer_DHpY · 2026-03-10

**Soundness:** 3
**Presentation:** 2
**Significance:** 3
**Originality:** 3
**Overall Recommendation:** 4
**Confidence:** 1

**Summary:**

This paper proposes CLARITree, a near-optimal and scalable algorithm for sparse piecewise linear regression trees that combines a one-step lookahead search strategy with efficient rank-one Cholesky updates of the regularized Gram matrix while keeping tree induction practical for large datasets and continuous features. The authors identify three core problems in existing methods: greedy induction deviates substantially from optimality with documented large performance gaps, optimal dynamic-programming approaches become computationally prohibitive once linear leaf models are required, and current linear tree methods either sacrifice expressiveness or fail to scale. CLARITree designs a lookahead mechanism that evaluates candidate splits with greedy completions, updates leaf regressors incrementally via rank-one Cholesky factorizations, and supports direct handling of continuous features without binarization. The authors conducted comprehensive evaluations on 16 real-world regression datasets across small, medium, and large scales using models with up to 21 features, significantly improving test R² while remaining within a 10-minute training budget, effectively alleviating the scalability bottleneck of exact solvers and the suboptimality of pure greedy baselines.

**Compliance With Llm Reviewing Policy:**

Affirmed.

**Key Questions For Authors:**

1. The authors only adopted the one-step lookahead strategy; why did they not attempt two-step or adaptive lookahead? Is there a better trade-off point worth exploring between accuracy improvement and computational overhead?
2. In the experiments, the maximum tree depth is only d=4; why were results for deeper trees (d=6 or 8) not reported? Does CLARITree still maintain its scalability advantage under more complex tree structures?
3. Compared to PILOT (Raymaekers et al., 2024), where does the main advantage of CLARITree lie?
4. The feature dimension k in the current experimental datasets is generally small (mostly <20). How does CLARITree perform and scale in high-dimensional scenarios (k>50 or even k>100)?

**Limitations:**

No, the authors did not discuss the limitations and potential negative societal impact of their work.

**Strengths And Weaknesses:**

**Strengths:**
1. The integration of one-step lookahead with rank-one Cholesky updates creates a clean “look one step, complete greedily” closed loop that achieves near-optimal accuracy with dramatically better scalability.
2. The direct continuous-feature handling and streaming split enumeration eliminate the need for binarization, delivering both theoretical complexity gains and practical speedups of several orders of magnitude.
3. The experiments span 16 datasets, multiple sparsity levels, and large-scale instances, delivering reusable infrastructure and strong empirical evidence for the entire piecewise linear tree community.

**Weaknesses:**
1. The method still relies heavily on greedy completions beyond the lookahead depth, which may limit further quality gains on certain distributions.
2. All results are obtained under a fixed 10-minute budget and depth-4 setting, leaving open questions about behavior at greater depths or longer runtimes.

---

> ### Author Rebuttal · Authors · 2026-03-31
>
> Thank you for your thoughtful review of our work.
> > Q1: The authors only adopted the one-step lookahead strategy; why did they not attempt two-step or adaptive lookahead? Is there a better trade-off point worth exploring between accuracy improvement and computational overhead?
>
> Our motivation for using one step lookahead is from [1], which shows that LicketySPLIT - which is a one step lookahead algorithm - finds classification trees that perform very close to optimal.
>
> We should note that one-step lookahead, in this context, still only makes a split when a greedy algorithm run to completion on that split is better than a greedy algorithm run to completion on any other choice of split. It serves as an effective evaluation function for a candidate feature, and further lookahead brings limited gains but significant overhead.
>
> > Q2: In the experiments, the maximum tree depth is only d=4; why were results for deeper trees (d=6 or 8) not reported? Does CLARITree still maintain its scalability advantage under more complex tree structures?
>
> We use depth 4 to align with prior work, STreeD, and ensure a fair and meaningful comparison. (1) For large datasets (i.e., the 'large scale' section in Table 2), even optimal solvers such as STreeD scale only to depth 2–3 (as reported in our paper). (2) For medium-scale datasets, we additionally evaluate deeper trees (d = 6, 8) under $\lambda\in$ {$1e-5, 1e-3, 1e-1$} with $\kappa = 0.01$, each entry reports the ratio (Test MSE / Time / #Leaves) relative to depth 4. Results show no consistent improvement in test MSE and often worse generalization due to overfitting, while training time increases only moderately ($\approx$ 2–5×), indicating that CLARITree remains computationally scalable, but the number of leaves grows dramatically (up to 50×), harming interpretability. (3) For large $\lambda$, all depths and datasets collapse to similar solutions, further diminishing the benefit of deeper trees (omitted due to space).
> |Dataset|D6, λ=1e-5|D6, λ=1e-3|D8, λ=1e-5|D8, λ=1e-3|
> |-|-|-|-|-|
> |airfoil|0.98/2.15/4.00|0.94/2.05/2.56|1.10/3.35/14.98|1.06/2.81/3.09|
> |auction|0.59/2.17/3.01|0.63/1.89/1.69|0.72/2.93/7.46|0.66/2.26/1.82|
> |auto_mpg|1.30/2.07/3.90|1.13/1.88/1.78|1.57/3.20/9.75|1.03/2.18/1.98|
> |energe_c|0.72/2.14/3.66|0.95/1.53/1.18|0.99/4.03/9.89|0.91/1.74/1.29|
> |energe_h|0.70/2.46/3.20|1.00/1.50/1.00|1.07/3.05/5.59|1.00/1.58/1.00|
> |insurance|1.34/2.57/3.94|1.09/1.77/1.85|1.99/3.72/12.24|1.13/1.98/2.14|
> |optical_net|1.13/1.90/3.94|1.05/2.25/2.03|1.64/2.83/9.74|1.08/2.03/2.21|
> |real_estate|1.35/2.04/3.84|1.42/1.94/2.60|1.51/3.13/9.55|1.09/3.02/2.86|
> |servo|1.00/1.23/1.00|1.00/1.02/1.00|1.00/1.23/1.00|1.00/1.00/1.00|
> |synch|5.90/3.96/39.41|1.00/0.99/1.00|7.36/5.72/55.80|1.00/0.99/1.00|
> |yacht|2.36/2.21/2.58|1.00/1.87/1.00|2.97/2.91/3.20|1.00/2.31/1.00|
> > Q3: Compared to PILOT (Raymaekers et al., 2024), where does the main advantage of CLARITree lie?
>
> PILOT operates on a completely different model class from ours. It fits residuals greedily with very small univariate models at each node and aggregates them along the path, i.e., it is closer to a boosting style tree. While CLARITree instead targets the classical piecewise-linear tree formulation and makes that harder objective tractable via lookahead optimization and Cholesky acceleration. Empirically, our added test R^2 results also show that PILOT is frequently even weaker than the greedy baseline:
>
> |Dataset|Greedy|PILOT|
> |-|-|-|
> |airfoil|0.86|0.53|
> |auction|0.96|0.88|
> |auto_mpg|0.84|0.81|
> |energe_c|0.97|0.90|
> |energe_h|1.00|0.93|
> |insurance|0.86|0.86|
> |optical_net|0.99|0.71|
> |real_estate|0.61|0.68|
> |servo|0.49|0.53|
> |synch|1.00|1.00|
> |yacht|0.99|0.87|
> |california_housing|0.73|0.65|
> |seoul_bike|0.71|0.54|
> |temperature_max|0.84|0.78|
> |temperature_min|0.88|0.84|
> |walmart|0.18|0.02|
> > Q4: The feature dimension $k$ in the current experimental datasets is generally small (mostly $< 20$). How does CLARITree perform and scale in high-dimensional scenarios ($k>50$ or even $k>100$)?
>
> We refer the reviewer to our responses to Reviewer 8Yxh (W1) for discussion on practical scalability, and Reviewer bGd1 (Q1) for mitigation strategies (e.g., feature/threshold selection, threshold guessing).
>
> As a brief empirical reference (5 thresholds per feature) on datasets from [2]:
>
> - nyc-taxi (581K samples, 9 features): Greedy 24.5s, CLARITree 518s;
>
> - superconduct (21K samples, 79 features): Greedy $\approx 1$ min, CLARITree $\approx 300$ min.
>
> While certainly the $k^4$ scaling is a limitation, our method still provides a way to find an interpretable model for high stakes tasks where optimal methods are intractable, but spending a few hours of compute to find a strong interpretable model is still warranted. We will discuss this more in the final version.
>
> **Reference**
>
> [1] Babbar et al. "Near optimal decision trees in a SPLIT second." (2025)
>
> [2] Grinsztajn et al "Why do tree-based models still outperform deep learning on typical tabular data?." (2022)

---

### Official Review · Reviewer_wjB2 · 2026-03-12

**Soundness:** 3
**Presentation:** 3
**Significance:** 3
**Originality:** 3
**Overall Recommendation:** 4
**Confidence:** 3

**Summary:**

This paper proposes CLARITree, an efficient algorithm for building sparse, piecewise linear regression trees that balances the computational cost with the performance limitations of standard greedy induction. Its main contribution is a lookahead search strategy accelerated by rank-one Cholesky updates. By incrementally updating regularized Gram matrices, CLARITree evaluates continuous feature splits without repeated least-squares refitting or full data binarization. Theoretically, the authors establish computational complexity bounds and show that CLARITree achieves strictly lower mean-squared error than greedy methods. Empirically, the proposed method demonstrates strong trade-offs among predictive accuracy, tree sparsity, and runtime, scaling to dataset sizes where exact dynamic programming approaches fail.

**Compliance With Llm Reviewing Policy:**

Affirmed.

**Final Justification:**

My concerns have been addressed, and I would like to raise my score.

**Key Questions For Authors:**

*Q1:* Sequential rank-one Cholesky updates easily accumulate floating-point errors, especially for deep trees with poorly conditioned data matrices. Can you provide a formal error bound for these updates, for example, may be based on tree depth and the root matrix's condition number? What safeguards prevent numerical failure?

*Q2:* Proving an MSE gap against greedy methods on parity problems is a weak baseline. Does CLARITree have a formal approximation guarantee (for example, $(1 + \epsilon)$-optimality) compared to exact optimal solvers like STreeD? If not, exactly when does this heuristic fail to find the optimal split?

*Q3:* The first and second contribution bullet points on Page 2 seem highly redundant, as both describe the lookahead search enabled by Cholesky updates. Is there a distinct mathematical or algorithmic difference in the second bullet that separates it from the first?

**Limitations:**

Yes

**Strengths And Weaknesses:**

*Soundness:*
The paper is, in general, technically sound. Using rank-one Cholesky updates reduces time complexity from $\mathcal{O}(k^3)$ to $\mathcal{O}(k^2)$.

*Presentation:*
The text is well-structured and clearly contrasts its approach with exact solvers. However, the first and second contribution bullets on Page 2 heavily overlap; both essentially describe the same algorithmic mechanism (lookahead search via Cholesky updates). Consolidating these would make the claims much more precise.

*Significance:*
The paper studies a major bottleneck concerning the relatively high computational cost of piecewise linear regression trees.

*Originality:*
The main novelty is applying the lookahead search from classification to linear regression. Combining this with Cholesky updates to scan continuous thresholds without binarization is a combination of existing methods. However, because the first two contribution bullets on Page 2 overlap heavily, the core novelty is narrower than initially claimed. The originality lies primarily in a clever engineering synthesis rather than in introducing fundamentally new learning theory.

*Overall:*
The paper provides a strong piece of engineering, bridging the computational gap between greedy induction and exact solvers (like STreeD) for piecewise linear regression trees. However, the theoretical result is overstated, redundant in its claims (Contributions 1 and 2), and lacks the strict error bounds required to prove its soundness. I am willing to raise my score if the authors can provide:

- A rigorous numerical error bound for the recursive Cholesky updates as a function of tree depth and initial matrix conditioning.
- A formal characterization of the algorithm's approximation ratio relative to exact DP, or a mathematical condition of the distributions where the lookahead succeeds or fails.

---

> ### Author Rebuttal · Authors · 2026-03-31
>
> Thank you for your thoughtful review of our work.
>
> >Q1: Sequential rank-one Cholesky updates easily accumulate floating-point errors, especially for deep trees with poorly conditioned data matrices. Can you provide a formal error bound for these updates, for example, may be based on tree depth and the root matrix's condition number? What safeguards prevent numerical failure?
>
> **Safeguards.**
> With $\kappa>0$ regularization for numerical stability and normalized $X$ (yielding well-controlled condition numbers). In practice, we observed no loss of positive definiteness or numerical instability in Cholesky updates/downdates, even at $\kappa=10^{-12}$.
>
> **Error Bound.**
> Let $A_t = A_{t-1} + x_t x_t^\top \succ 0$, and let $\tilde L_t = \mathrm{cholupdate}(\tilde L_{t-1}, x_t)$. Then
> $\tilde L_t \tilde L_t^\top = A_t + E_t$, where we can prove that
> $\\|E_t\\| \le (1+cu)\\|E_{t-1}\\| + cu(\\|A_{t-1}\\|+\\|x_t\\|^2)$,
> and hence $\\|E_t\\| = O\left(tu\\|A_t\\|\right)$, with $u$ is machine epsilon, $c$ is a constant, and $\\|\\|$ is l2 norm.
>
> **Proof sketch.**
> Each rank-one update is implemented via a sequence of Givens rotations [3], which are backward stable: $\mathrm{float}(QA) = (Q+\Delta Q)A$ with $\\|\Delta Q\\|=O(u)$.
>
> Applying this to $[\tilde L_{t-1}, x_t]$ gives
> $\tilde L_t = \hat L_t + \Delta L_t$ with $\\|\Delta L_t\\| \le cu(\\|\tilde L_{t-1}\\|+\\|x_t\\|)$, where $\hat L_t$ is the exact updated factor.
>
> Since $A_{t-1} = \tilde L_{t-1}\tilde L_{t-1}^\top - E_{t-1}$, the exact update satisfies $\hat L_t \hat L_t^\top = A_t + E_{t-1}$. Expanding $\tilde L_t \tilde L_t^\top$ yields $E_t = E_{t-1} + \hat L_t\Delta L_t^\top + \Delta L_t \hat L_t^\top + \Delta L_t \Delta L_t^\top$.
>
> Using triangle equality we obtain $\\|E_t\\| \le \\|E_{t-1}\\| + O(u)\big(\\|A_t+E_{t-1}\\|^{1/2}(\\|A_{t-1}\\|^{1/2}+\\|E_{t-1}\\|^{1/2}+\\|x_t\\|)\big)$.
>
> Using $A_t = A_{t-1}+x_tx_t^\top$, this simplifies to $\\|E_t\\| \le (1+cu)\\|E_{t-1}\\| + cu(\\|A_{t-1}\\|+\\|x_t\\|^2)$, which implies the recursion and the bound.
>
> **Insight.**
> Error scales **linearly** in $t$. Moreover, we have $\\|A_t^{-1}E_t\\|=O(tu\kappa_2(A_t))$. With normalized $X$, the condition number $\kappa_2(A_t)= O(k/\kappa)$. By [4, Thm 7.2],
> $\\|\hat\beta_t-\beta_t\\|/\|\\beta_t\\|=O(tu\kappa_2(A_{t}))=O(tuk/\kappa)$. Also, Cholesky is recomputed per node, so errors for coefficients do not accumulate along the tree and are bounded by $O(unk/\kappa)$.
>
> >Q2: Proving an MSE gap against greedy methods on parity problems is a weak baseline. Does CLARITree have a formal approximation guarantee (for example, -optimality) compared to exact optimal solvers like STreeD? If not, exactly when does this heuristic fail to find the optimal split?
>
> While an approximation ratio would certainly be ideal, it would also be an extraordinary contribution to the literature; we are not aware of any regularized decision tree optimization work with formal approximation guarantees for general inputs (aside from globally optimal methods). If the reviewer is aware of any work establishing such guarantees, we would greatly appreciate the reference and would be happy to incorporate and discuss it. It may be possible to bound our method's approximation in cases where data is perfectly separable and/or features are independent, but we do not consider those to be realistic settings for machine learning tasks where we want to deal with noisy data and limit overfitting.
>
> We are also not aware of any work that provides approximation ratio guarantees for this class of lookahead-based heuristics (e.g., rollout[1] or pilot[2] methods), even in standard combinatorial optimization settings.
>
> Our method will encounter difficulty for a parity function with 3 or more bits; this is common to many non-optimal methods. Intuitively, this implies that for a dataset where the degree of interaction between features is higher than 1 + the lookahead depth of CLARITree, it is not guaranteed to find the optimal tree.
>
> > Q3: The first and second contribution bullet points on Page 2 seem highly redundant, as both describe the lookahead search enabled by Cholesky updates. Is there a distinct mathematical or algorithmic difference in the second bullet that separates it from the first?
>
> We believe these contributions are distinct and will make that clearer in the final version. The first point introduces our algorithm's adoption of lookahead-style search for discovering piecewise linear regression trees on tabular data. The second point specifically highlights our algorithm's scalability in the continuous feature regime (e.g., in cases where binarization would yield similar thresholds - age > 35, age > 36…).
>
> **Reference**
>
> [1] Bertsekas et al. "Rollout algorithms for combinatorial optimization." (1997)
>
> [2] Voßs et al. "Looking ahead with the pilot method." (2005)
>
> [3] Seeger et al. "Low rank updates for the Cholesky decomposition." (2004)
>
> [4] Higham. "Accuracy and Stability of Numerical Algorithms." (2002)

---

> > ### Author Rebuttal · Reviewer_wjB2 · 2026-04-02
> >
> > I thank the authors for their effort on addressing these questions. My concerns have been adequately addressed and I raised my score to 4.

---

> > > ### Author Response · Authors · 2026-04-04
> > >
> > > Thank you for your positive feedback and for raising your score. We’re glad that our response addressed your concerns.

---

### Official Review · Reviewer_bGd1 · 2026-03-12

**Soundness:** 3
**Presentation:** 3
**Significance:** 3
**Originality:** 3
**Overall Recommendation:** 4
**Confidence:** 2

**Summary:**

The paper proposes CLARITree, an algorithm for learning piecewise linear regression trees. It addresses the computational bottleneck in evaluating splits for linear regression trees, where standard methods require $O(k^3)$ per candidate split. CLARITree introduces a lookahead search strategy to improve tree quality and utilizes rank-one Cholesky updates to reduce the cost of evaluating each split to $O(k^2)$. The authors demonstrate that this approach allows PLRTs to scale to datasets with tens of thousands of instances, achieving performance competitive with optimal decision tree solvers while being significantly faster.

**Compliance With Llm Reviewing Policy:**

Affirmed.

**Final Justification:**

I appreciate the author's rebuttal. I will maintain my positive score.

**Key Questions For Authors:**

1. Given the $k^4$ term in the complexity analysis, have you tested the performance on datasets with a higher number of features? Are there potential approximation methods to mitigate this bottleneck?

2. How does the lookahead depth ($d-d'-1$) impact the final results and the computational budget? Is there an optimal depth that balances performance gains and overhead?

3. In your experiments, how frequently did the Cholesky downdate result in non-positive definite matrices, and how much did the fallback mechanism affect final accuracy?

**Limitations:**

yes

**Strengths And Weaknesses:**

# Strengths
1. The algorithm fills a gap between fast but suboptimal greedy methods and accurate but computationally expensive optimal solvers. It enables piecewise linear regression trees to run on large-scale datasets where traditional optimal methods typically run slower.
2. The use of rank-one Cholesky updates/downdates effectively addresses the primary computational hurdle of PLRTs. Combining this with a lookahead search strategy and ensuring numerical stability represents a solid engineering and algorithmic contribution.
3. The paper provides a comprehensive comparison across multiple benchmark datasets, evaluating performance and runtime against both greedy baselines and state-of-the-art optimal tree solvers like STreeD.
# Weaknesses
1. According to the complexity analysis $O(d^2 n k^4 T)$, the algorithm’s dependence on the number of features $k$ is quartic. This may limit its applicability to high-dimensional tabular data.
2. The comparisons are largely confined to the decision tree family. While the paper focuses on interpretability, the lack of a direct accuracy comparison with high-performance black-box models makes it difficult to assess the accuracy-interpretability trade-off.
3. While numerical stability is discussed, more analysis could be provided regarding the robustness of the Cholesky downdating mechanism in extreme cases of feature collinearity on very large datasets.

---

> ### Author Rebuttal · Authors · 2026-03-31
>
> Thank you for your thoughtful review of our work.
>
> > Q1: Given the $k^4$ term in the complexity analysis, have you tested the performance on datasets with a higher number of features? Are there potential approximation methods to mitigate this bottleneck?
>
> The best approximation method would be to run feature selection or threshold selection based on the statistics in [1] as a part of a preprocessing method. Approximation methods available in our toolbox when further simplification is needed:
> 1. Reference ensemble threshold guessing [2],
> 2. Constant regression
>
> We also note that optimal decision tree methods have similar issues, but the runtime of those methods is exponential in the depth. We manage to mitigate these problems and achieve polynomial runtime, regardless of specified depth.
>
> For concerns regarding high dimensional datasets, we refer the reviewer to our response to Reviewer 8Yxh (W1) for a detailed discussion.
>
> > Q2: How does the lookahead depth ($d-d'-1$) impact the final results and the computational budget? Is there an optimal depth that balances performance gains and overhead?
>
> For the main paper, we focus on a lookahead of $1$, which is the most scalable while often already matching optimal performance; please also see our response to DHpY (Q1) for more reasons why $1$ is sufficient. Empirically, increasing the lookahead depth incurs significant computational overhead with only limited performance gains.
>
>
> > Q3: In your experiments, how frequently did the Cholesky downdate result in non-positive definite matrices, and how much did the fallback mechanism affect final accuracy?
>
> Because we have a nonzero kappa regularization term for numerical stability, the Cholesky downdate never resulted in non-positive definite matrices. We tried kappa values as low as 1e-12 without numerical issues.
>
> > W2: The comparisons are largely confined to the decision tree family. While the paper focuses on interpretability, the lack of a direct accuracy comparison with high-performance black-box models makes it difficult to assess the accuracy-interpretability trade-off.
>
> We have added a direct comparison with strong black-box models (same setup, 20 Optuna trials for all methods). Notably, CLARITree narrows the interpretability-accuracy gap while maintaining full interpretability.
>
> |method|airfoil|auction|california|
> |-|-|-|-|
> |**interpretable**||||
> |claritree|**0.892±0.019**|**0.973±0.003**|**0.784±0.009**|
> |greedy|0.875±0.019|0.969±0.008|0.752±0.008|
> |**black-box**||||
> |catboost|0.959±0.005|0.998±0.001|0.844±0.005|
> |gboost|0.947±0.007|0.996±0.002|0.840±0.008|
> |lightgbm|0.946±0.006|0.995±0.002|0.846±0.006|
> |rf|0.886±0.015|0.981±0.006|0.821±0.006|
> |xgboost|0.949±0.007|0.998±0.001|0.847±0.006|
>
>
> **Reference**
>
> [1] Zeileis, Achim, Torsten Hothorn, and Kurt Hornik. "Model-based recursive partitioning." (2008)
>
> [2] McTavish, Hayden, et al. "Fast sparse decision tree optimization via reference ensembles." (2022)

---

> > ### Author Rebuttal · Reviewer_bGd1 · 2026-04-04
> >
> > Thank you for the detailed response. Your response successfully addressed my main concerns. I will maintain my original positive score.

---

> > > ### Author Response · Authors · 2026-04-04
> > >
> > > Thank you for your positive feedback and support. We’re glad that our response addressed your concerns.

---

### Decision · Program_Chairs · 2026-04-30

**Decision:**

Accept (regular)

**Comment:**

This paper proposes a new scheme of constructing piecewise-linear interpretable regression trees (with axis-aligned split) in a non-greedy manner, combining the recent SPLIT method of lookahead in non-greedy classification tree construction with the observation that the ridge regression models for any two child nodes can be efficiently obtained and evaluated with a rank-one update to a Cholesky factorization, which is significantly more efficient than repeatedly solving two new ridge regression problems for each split evaluation. The computational overhead of this proposed scheme is theoretically bounded utilized standard decision tree analysis. The scheme is empirically compared against existing regression tree construction schemes with depth 4 decision trees, demonstrating extremely strong relative performance for time-bounded training.

There were some reviewer concerns regarding the (theoretical) scalability of the proposed scheme, and the accumulation of floating point errors (and numerical stability in general) in the rank-one updates to the Cholesky factorization. However, the authors adequately addressed these concerns, even providing new empirical results with deeper trees and larger datasets, highlighting the scalability of the proposed scheme.